# Learning Theory for Kernel Bilevel Optimization

**Fares El Khoury**[*]
Université Grenoble Alpes, Inria,
CNRS, Grenoble INP, LJK,
38000 Grenoble, France

**Edouard Pauwels**
Toulouse School of Economics,
Université Toulouse Capitole,
31080 Toulouse, France

**Samuel Vaiter**
CNRS & Université Côte d'Azur,
Laboratoire J. A. Dieudonné,
06108 Nice, France

**Michael Arbel**
Université Grenoble Alpes, Inria,
CNRS, Grenoble INP, LJK,
38000 Grenoble, France

## Abstract

Bilevel optimization has emerged as a technique for addressing a wide range of machine learning problems that involve an outer objective implicitly determined by the minimizer of an inner problem. While prior works have primarily focused on the *parametric* setting, a *learning-theoretic* foundation for bilevel optimization in the *nonparametric* case remains relatively unexplored. In this paper, we take a first step toward bridging this gap by studying *Kernel Bilevel Optimization* (KBO), where the inner objective is optimized over a reproducing kernel Hilbert space. This setting enables rich function approximation while providing a foundation for rigorous theoretical analysis. In this context, we derive novel *finite-sample generalization bounds* for KBO, leveraging tools from empirical process theory. These bounds further allow us to assess the statistical accuracy of gradient-based methods applied to the empirical discretization of KBO. We numerically illustrate our theoretical findings on a synthetic instrumental variable regression task.

## 1 Introduction

Bilevel optimization involves a nested structure where one optimization problem, called *outer-level*, is constrained by the solution of another one, called *inner-level* [19]. This formulation has found applications in a broad spectrum of machine learning fields, including hyperparameter tuning [43, 13, 28], meta-learning [14, 59], inverse problems [34], and reinforcement learning [35, 48], making it a powerful tool in theoretical and practical contexts. Its widespread use naturally raises fundamental questions about the *generalization properties* of models learned through this procedure as the number of data samples increases. Several existing works have studied the generalization and convergence of bilevel algorithms under the assumption that the inner-level problem is *strongly convex* and that its parameters lie in a *finite-dimensional* space. These include analyses of the convergence of stochastic bilevel optimization algorithms [3, 24, 29, 38] and approaches based on algorithmic stability [7, 78]. The strong convexity assumption ensures a *unique* inner-level solution, which is crucial for stability and convergence analysis in bilevel optimization. Moreover, restricting the inner-level parameters to a finite-dimensional space does not cover possibly *richer infinite-dimensional* spaces, as in kernel methods, where the parameter's dimension may grow with the sample size. This sample size dependence in *nonparametric* methods poses additional challenges as solutions at different sample sizes are not directly comparable. In contrast, in the finite-dimensional setting, generalization bounds can be derived by quantifying the convergence of inner-level finite-sample solutions toward the infinite samples *population solution* limit, within a fixed Euclidean space.

---

[*]Correspondence to: `fares.el-khoury@inria.fr`.

39th Conference on Neural Information Processing Systems (NeurIPS 2025).

Albeit theoretically convenient, strong convexity and finite-dimensionality limit models expressiveness, restricting them to linear functions. Moving beyond linear models requires either relaxing the strong convexity assumption to accommodate more expressive models, such as deep neural networks [30], or considering nonparametric bilevel problems, where the inner-level variable lies in an expressive infinite-dimensional function space, such as a *Reproducing Kernel Hilbert Space* (RKHS) [63]. Early works in bilevel optimization for machine learning followed the latter approach, developing methods for hyperparameter selection in kernel-based models [39, 41]. These works leverage the *representer theorem* [64] to transform the infinite-dimensional problem into a finite-dimensional one, with dimension depending on the sample size. However, they do not address how sample size impacts generalization. Another line of research instead focuses on relaxing the strong convexity assumption, proposing bilevel algorithms in the absence of convexity [4, 42, 66]. Yet, *non-convex* bilevel optimization is a *very hard* problem in general [47, 4, 17], and strong generalization guarantees in this setting remain out of reach due to instabilities of the inner-level solutions. Overall, learning theory for bilevel problems beyond the *strongly convex parametric* setting is essentially *lacking*.

In the present work, we take an initial step toward developing a *learning theory* that goes beyond the finite-dimensional setting. Specifically, we propose to study *Kernel Bilevel Optimization* (KBO) problems, where the inner objective $L_{in} : \mathbb{R}^d \times \mathcal{H} \to \mathbb{R}$ finds an optimal inner solution $h_\omega^\star$ in an RKHS $\mathcal{H}$ for a given parameter $\omega$ in $\mathbb{R}^d$, while the outer objective $L_{out} : \mathbb{R}^d \times \mathcal{H} \to \mathbb{R}$ optimizes the parameter $\omega$ over a closed subset $\mathcal{C}$ of $\mathbb{R}^d$, given the inner solution $h_\omega^\star$:

$$\min_{\omega \in \mathcal{C}} \mathcal{F}(\omega) := L_{out}(\omega, h_\omega^\star) \quad \text{s.t.} \quad h_\omega^\star = \arg\min_{h \in \mathcal{H}} L_{in}(\omega, h). \tag{KBO}$$

In particular, we focus on objectives that are expectations of point-wise losses, a common setting in learning theory. RKHS provides a natural framework to study learning-theoretic arguments, and has been instrumental for many fruitful results in pattern recognition and machine learning. They allow to describe very expressive non-linear models with simple and stable algorithms, while enabling a rich statistical analysis and featuring adaptivity to the regularity of the population problem [65, 63, 33]. Our choice is also motivated by the relevance of kernel methods, even in the deep learning era. They remain competitive for some prediction problems, such as those involving physics [26, 44]. Additionally, the mathematics of kernel methods are useful to describe the limiting behavior of deep network training for very large models [37, 11]. In this limit, the problem becomes (strongly) convex in an infinite-dimensional function space, simplifying the difficulties of non-convex model parameterizations, a major bottleneck in the analysis of such models. This point of view was leveraged by Petrulionytė et al. [58] who introduced *functional bilevel optimization*, and our setting can be seen as a special case for which the underlying function space is an RKHS. From a practical perspective, our setting is *amenable* to first-order methods using *implicit differentiation* techniques [31, 6, 15].

**Contributions.** We leverage empirical process theory and its extension to $U$-processes [67] to derive *uniform generalization bounds* for the value function of (KBO), quantifying the discrepancy between $\mathcal{F}$ and its plug-in estimator $\widehat{\mathcal{F}}$ in terms of both their values and gradients. Classical empirical process results [72] do not directly apply here, as our functional setting involves processes taking values in an *infinite-dimensional space* rather than real numbers. To address this, we exploit the RKHS structure to represent the resulting error terms as real-valued $U$-processes, enabling the use of results from Sherman [67]. The control in terms of gradients is crucial to study first-order optimization methods, since $\widehat{\mathcal{F}}$ is typically non-convex and iterative methods seek to find approximate critical points where $\|\nabla\widehat{\mathcal{F}}\|$ is small. Our result relies on an *equivalence* we establish between $\nabla\widehat{\mathcal{F}}$ and a plug-in statistical estimate of $\nabla\mathcal{F}$ that is *more amenable to a statistical analysis*. We then use our uniform bounds to provide generalization guarantees for *gradient descent* and *projected gradient descent* applied to $\widehat{\mathcal{F}}$. Under specific assumptions, we show *convergence rates* for *sub-optimality measures* that depend on the sample sizes and the number of algorithmic iterations. This illustrates the practical relevance of our generalization bounds on simple bilevel algorithms. For a large number of steps, gradient algorithms applied to the empirical (KBO) find approximate critical points of the population (KBO) up to a statistical error which we control.

**Organization of the paper.** In Section 2, we describe (KBO), give two application examples, and explain implicit differentiation in an RKHS. In Section 3, we present the empirical (KBO) and state our first main result on the gradient of its value function. Section 4 provides uniform generalization bounds for (KBO), with applications to bilevel gradient methods, as well as a sketch of the proof of our main result. Finally, in Section 5, we illustrate our theoretical findings with experiments on synthetic data for the instrumental variable regression problem.

## 2 Kernel bilevel optimization

### 2.1 Problem formulation

We consider the (KBO) problem with an RKHS $\mathcal{H}$, which is a space of real-valued functions defined on a *Borel* input space $\mathcal{X} \subset \mathbb{R}^p$ and associated with a *reproducing kernel* $K : \mathcal{X} \times \mathcal{X} \to \mathbb{R}$. We are interested, in particular, in (regularized) objectives expressed as expectations of point-wise loss functions, a formulation widely adopted in machine learning as it allows the loss functions to represent the average performance over some data distribution. Specifically, given two probability distributions $\mathbb{P}$ and $\mathbb{Q}$ supported on $\mathcal{X} \times \mathcal{Y}$ for some target space $\mathcal{Y} \subset \mathbb{R}^q$, we consider objectives of the form:

$$L_{out}(\omega, h) = \mathbb{E}_{\mathbb{Q}} \left[ \ell_{out}(\omega, h(x), y) \right], \quad L_{in}(\omega, h) = \mathbb{E}_{\mathbb{P}} \left[ \ell_{in}(\omega, h(x), y) \right] + \frac{\lambda}{2} \|h\|_{\mathcal{H}}^2,$$

where $\ell_{in}$ and $\ell_{out} : \mathbb{R}^d \times \mathbb{R} \times \mathbb{R}^q \to \mathbb{R}$ represent the inner and outer point-wise loss functions, $\lambda > 0$ is the regularization parameter which is fixed through this work, and $\| \cdot \|_{\mathcal{H}}$ denotes the norm in the RKHS $\mathcal{H}$. The regularization term in $L_{in}$ is often used in practice to prevent overfitting by penalizing overly complex models. In our setting, it ensures strong convexity of $h \mapsto L_{in}(\omega, h)$ under mild assumptions on $\ell_{in}$, which will be critical to leverage functional implicit differentiation.

**Assumptions.** Through the paper, we make the following five assumptions to derive generalization bounds while retaining a simple and modular presentation.

(**A**) (Measurability of $K$). $K$ is measurable on $\mathcal{X} \times \mathcal{X}$.

(**B**) (Boundedness of $K$). There exists a constant $\kappa > 0$ such that $K(x, x) \le \kappa$, for any $x \in \mathcal{X}$.

(**C**) (Compactness of $\mathcal{Y}$). The subset $\mathcal{Y}$ of $\mathbb{R}^q$ is compact.

(**D**) (Regularity of $\ell_{in}$ and $\ell_{out}$). The functions $\ell_{in}$ and $\ell_{out}$ are of class $C^3$ jointly in their first two arguments $(\omega, v)$, and their derivatives are jointly continuous in $(\omega, v, y)$.

(**E**) (Convexity of $\ell_{in}$). For any $(\omega, y) \in \mathbb{R}^d \times \mathbb{R}^q$, the map $v \mapsto \ell_{in}(\omega, v, y)$ is convex.

Assumptions (**A**) and (**B**) on $K$ hold for a wide class of kernels, such as the Gaussian, Laplacian, and Matérn [61] kernels. They also hold if $K$ is a *Mercer kernel* [50], *i.e.*, a continuous, positive-definite kernel on a compact domain $\mathcal{X}$. Specifically, a kernel built using neural network features, *e.g.*, *neural tangent kernel* [37], satisfies these assumptions on the space of images, which is compact since the pixel values have a bounded range. Assumption (**C**) on $\mathcal{Y}$ is a mild assumption that holds in most supervised learning applications, such as classification where $\mathcal{Y}$ is finite, or cases where $\mathcal{Y} = [0,1]^q$, enabling the representation of complex data, like images. Assumption (**D**) on the point-wise objectives is a mild regularity assumption, which is met for the most commonly used loss functions in practice, including the squared loss, logistic loss, cross-entropy loss, and KL divergence. Finally, Assumption (**E**) is essential to ensure the existence and uniqueness of a smooth minimizer $h_\omega^\star$. It is a relatively weak assumption that was recently considered in [58] in the context of functional bilevel optimization, and that holds in many cases of interest, as discussed in Section 2.2.

*Remark* 2.1. Assumptions (**B**) to (**D**) can be relaxed at the expense of weaker yet more technical assumptions, such as finite moment assumptions on $\mathbb{P}$ and $\mathbb{Q}$, and suitable polynomial growth of the kernel and some partial derivatives of $\ell_{in}$ and $\ell_{out}$. It is also sufficient to require that Assumption (**D**) holds on $\mathcal{U} \times \mathbb{R} \times \mathbb{R}^q$ where $\mathcal{U}$ is an open neighborhood of $\mathcal{C}$, and that Assumption (**E**) holds for any $\omega \in \mathcal{C}$ and $y \in \mathcal{Y}$. We prefer to keep these stronger yet simpler assumptions for clarity.

### 2.2 Examples of (KBO) in machine learning

To illustrate the relevance of (KBO), we consider two examples that highlight its applicability.

**Hyperparameter selection under distribution shift.** In this application, the aim is to select the best hyperparameters for a machine learning model, *e.g.*, regularization parameters, while accounting for distribution shift between the training and test data, *i.e.*, when the training and test data distributions $\mathcal{D}_{\text{train}}$ and $\mathcal{D}_{\text{test}}$ are different [57, 28]. This can be viewed as an instance of (KBO) when using models in an RKHS. At the inner-level, the model $h$ is trained to minimize the regularized training squared error loss, with the hyperparameter $\omega > 0$ representing the weight for the data fitting term. At the outer-level, the task is to select the hyperparameter $\omega$ that maximizes the model's performance on the

distribution-shifted test data. Both inner and outer objectives can thus be formulated as:

$$L_{out}(\omega, h) = \frac{1}{2}\mathbb{E}_{(x,y)\sim\mathcal{D}_{\text{test}}}\left[|h(x) - y|^2\right], \quad L_{in}(\omega, h) = \frac{\omega}{2}\mathbb{E}_{(x,y)\sim\mathcal{D}_{\text{train}}}\left[|h(x) - y|^2\right] + \frac{\lambda}{2}\|h\|_{\mathcal{H}}^2.$$

This formulation could be used for domain adaptation [12] or domain generalization [73] to choose hyperparameters that perform well on the distribution-shifted test data.

**Instrumental variable regression.** It is a technique used to address endogeneity in statistical modeling by leveraging instruments to estimate causal relationships [53]. The goal is to estimate a function $t \mapsto f_\omega(t)$ parameterized by a vector $\omega$, that satisfies $y = f_\omega(t) + \epsilon$, where $y \in \mathbb{R}$ is the observed outcome, $t$ is the treatment, and $\epsilon$ is the error term. The key issue is that $t$ is endogenous, which means that it is correlated with $\epsilon$, making direct regression inconsistent. Indeed, such correlation leads to biased estimates of $f_\omega(t)$ as the assumption of exogeneity, *i.e.*, independence of $t$ and $\epsilon$, is violated. To resolve this, one can use an instrumental variable $x$, uncorrelated with $\epsilon$ but correlated with $t$, to recover the relationship between $y$ and $t$ without being directly affected by the bias introduced by $\epsilon$, typically via two-stage least squares regression [68, 51]. As shown in [58], this approach can be naturally expressed as a bilevel problem with inner and outer objectives of the form:

$$L_{out}(\omega, h) = \frac{1}{2}\mathbb{E}_{x,y}\left[|h(x) - y|^2\right], \quad L_{in}(\omega, h) = \frac{1}{2}\mathbb{E}_{x,t}\left[|h(x) - f_\omega(t)|^2\right] + \frac{\lambda}{2}\|h\|_{\mathcal{H}}^2,$$

where $h$ can be chosen to be in an RKHS to allow flexibility in the estimation while retaining uniqueness of the solution $h_\omega^\star$, a key property in bilevel optimization.

## 2.3 Implicit differentiation in an RKHS

A stationarity measure in (KBO) is the gradient $\nabla\mathcal{F}(\omega)$ of the value function $\mathcal{F}$. Computing $\nabla\mathcal{F}(\omega)$, however, is challenging and will be addressed in this section. At a high level, our approach proceeds in two steps. First, we derive an abstract, *a priori* intractable, expression for $\nabla\mathcal{F}(\omega)$ using implicit differentiation in an RKHS, which is the main source of difficulty. Then, we leverage the structure of our problem to reformulate the gradient in Proposition 2.2 using the solution of a regression problem in the RKHS (the adjoint problem). This more concrete formulation can be approximated with finite samples and will later serve as the foundation of our statistical analysis. Formally, evaluating the gradient requires computing the Jacobian $\partial_\omega h_\omega^\star$, which can be viewed as a linear operator from $\mathcal{H}$ to $\mathbb{R}^d$. Indeed, $h_\omega^\star$ depends implicitly on $\omega$. A key ingredient for computing $\partial_\omega h_\omega^\star$ is the *implicit function theorem* [36], which guarantees the differentiability of the implicit function $\omega \mapsto h_\omega^\star$ and allows characterizing $\partial_\omega h_\omega^\star$ as the *unique* solution of a linear system of the form:

$$\partial_{\omega,h}^2 L_{in}(\omega, h_\omega^\star) + \partial_\omega h_\omega^\star \partial_h^2 L_{in}(\omega, h_\omega^\star) = 0, \tag{1}$$

where $\partial_h^2 L_{in}(\omega, h_\omega^\star)$ is an operator from $\mathcal{H}$ to itself representing the Hessian of $L_{in}$ w.r.t. $h$, while $\partial_{\omega,h}^2 L_{in}(\omega, h_\omega^\star)$ is an operator from $\mathcal{H}$ to $\mathbb{R}^d$ representing the cross derivatives of $L_{in}$ w.r.t. to $\omega$ and $h$. Applying such result requires $h \mapsto L_{in}(\omega, h)$ to be Fréchet differentiable with invertible Hessian operator and jointly Fréchet differentiable gradient map $(\omega, h) \mapsto \partial_h L_{in}(\omega, h)$. All these properties are satisfied in our setting under Assumptions (A) to (E) as shown in Propositions B.1 to B.3 of Appendix B.1. Furthermore, when $L_{out}$ is Fréchet differentiable, which is our case under Assumptions (A) to (D) (Proposition B.1 of Appendix B.1), then by composition with $\omega \mapsto (\omega, h_\omega^\star)$, the map $\omega \mapsto \mathcal{F}(\omega)$ must also be differentiable with gradient obtained using the chain rule:

$$\nabla\mathcal{F}(\omega) = \partial_\omega L_{out}(\omega, h_\omega^\star) + \partial_\omega h_\omega^\star \partial_h L_{out}(\omega, h_\omega^\star).$$

The above expression for the gradient is intractable as it involves abstract operators, namely the derivatives $\partial_h$, $\partial_h^2$, and $\partial_{\omega,h}^2$, the last two of which arise when replacing $\partial_\omega h_\omega^\star$ by its expression in Equation (1). In Proposition 2.2 below, we derive an explicit expression for $\nabla\mathcal{F}(\omega)$ which exploits the particular structure of the objectives $L_{in}$ and $L_{out}$ as expectations of point-wise losses.

**Proposition 2.2** (Expression of the total gradient). *Under Assumptions (A) to (E), $\mathcal{F}$ is differentiable on $\mathbb{R}^d$, with gradient $\nabla\mathcal{F}(\omega)$, for any $\omega \in \mathbb{R}^d$, given by:*

$$\nabla\mathcal{F}(\omega) = \mathbb{E}_\mathbb{Q}\left[\partial_\omega \ell_{out}(\omega, h_\omega^\star(x), y)\right] + \mathbb{E}_\mathbb{P}\left[\partial_{\omega,v}^2 \ell_{in}(\omega, h_\omega^\star(x), y)a_\omega^\star(x)\right], \tag{2}$$

*where the adjoint function $a_\omega^\star \in \mathcal{H}$ is the unique minimizer of a strongly convex quadratic objective $a \mapsto L_{adj}(\omega, a)$ defined on $\mathcal{H}$ as:*

$$L_{adj}(\omega, a) := \frac{1}{2}\mathbb{E}_\mathbb{P}\left[\partial_v^2 \ell_{in}(\omega, h_\omega^\star(x), y)a^2(x)\right] + \mathbb{E}_\mathbb{Q}\left[\partial_v \ell_{out}(\omega, h_\omega^\star(x), y)a(x)\right] + \frac{\lambda}{2}\|a\|_{\mathcal{H}}^2, \tag{3}$$

where $\partial_\omega \ell_{out}$ and $\partial_v \ell_{out}$ are the first-order partial derivatives of $\ell_{out}$ w.r.t. $\omega$ and $v$, while $\partial^2_{\omega,v}\ell_{in}$ and $\partial^2_v \ell_{in}$ denote the second-order partial derivatives of $\ell_{in}$ w.r.t. $\omega$ and $v$.

Proposition 2.2 is proved in Appendix B and relies essentially on proving Bochner's integrability [25, Definition 1, Chapter 2] of some suitable operators on $\mathcal{H}$, and then applying Lebesgue's dominated convergence theorem for Bochner's integral [25, Theorem 3, Chapter 2] to interchange derivatives and expectations. The expression in Proposition 2.2 provides a natural way for approximating $\nabla\mathcal{F}(\omega)$ by estimating all expectations using finite-sample averages, as we further discuss in Section 3.

## 3  Finite-sample approximation of (KBO)

In this section, we consider an approximation of (KBO) when only a *finite* number of *i.i.d.* samples $(x_i, y_i)_{1 \le i \le n}$ and $(\tilde{x}_j, \tilde{y}_j)_{1 \le j \le m}$ from $\mathbb{P}$ and $\mathbb{Q}$ are available. This setting is ubiquitous in machine learning as it allows finding tractable approximate solutions to the original problem. As we are interested in approximately solving (KBO) using gradient methods, our focus here is to derive estimators for both the value function $\mathcal{F}(\omega)$ and its gradient $\nabla\mathcal{F}(\omega)$, whose generalization properties will be studied in Section 4.

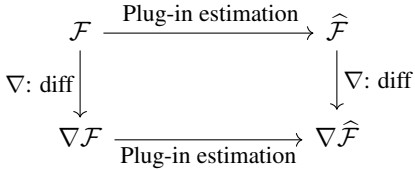

Figure 1: A commutative diagram illustrating that plug-in statistical estimation and differentiation can be interchanged for $\mathcal{F}$ and $\widehat{\mathcal{F}}$ resulting in a single gradient estimator.

In Section 3.1, we follow a commonly used approach of first deriving a plug-in estimator $\widehat{\mathcal{F}}$ of the value function, then considering its gradient $\nabla\widehat{\mathcal{F}}(\omega)$ as an approximation to $\nabla\mathcal{F}(\omega)$. In Section 3.2, we show that this approximation is *equivalent* to a second estimator, more amenable to a statistical analysis, obtained by directly computing a plug-in estimator of $\nabla\mathcal{F}$ based on its expression in Equation (2). Figure 1 summarizes such equivalence.

### 3.1  Value function: plug-in estimator and its gradient

A natural approach for finding approximate solutions to (KBO) is to consider an approximate problem obtained after replacing the objectives $L_{in}$ and $L_{out}$ by their empirical approximations $\widehat{L}_{in}$ and $\widehat{L}_{out}$:

$$\widehat{L}_{out}(\omega, h) = \frac{1}{m}\sum_{j=1}^m \ell_{out}(\omega, h(\tilde{x}_j), \tilde{y}_j), \quad \widehat{L}_{in}(\omega, h) = \frac{1}{n}\sum_{i=1}^n \ell_{in}(\omega, h(x_i), y_i) + \frac{\lambda}{2}\|h\|_{\mathcal{H}}^2.$$

A plug-in estimator $\omega \mapsto \widehat{\mathcal{F}}(\omega)$ is then obtained by first finding a solution $\hat{h}_\omega$ minimizing $h \mapsto \widehat{L}_{in}(\omega, h)$, that is meant to approximate the optimal inner solution $h_\omega^\star$, and subsequently plugging it into $\widehat{L}_{out}$. This procedure results in the following empirical version of (KBO):

$$\min_{\omega \in \mathcal{C}} \widehat{\mathcal{F}}(\omega) \coloneqq \widehat{L}_{out}(\omega, \hat{h}_\omega) \quad \text{s.t.} \quad \hat{h}_\omega = \arg\min_{h \in \mathcal{H}} \widehat{L}_{in}(\omega, h).$$

The inner problem still requires optimizing over a, potentially infinite-dimensional, RKHS. However, its finite-sum structure allows equivalently expressing it as a finite-dimensional bilevel optimization, by application of the so-called representer theorem [64]:

$$\min_{\omega \in \mathcal{C}} \widehat{\mathcal{F}}(\omega) \coloneqq \frac{1}{m}\sum_{j=1}^m \ell_{out}(\omega, (\overline{\mathbf{K}}\,\hat{\boldsymbol{\gamma}}_\omega)_j, \tilde{y}_j)$$

$$\text{s.t.} \quad \hat{\boldsymbol{\gamma}}_\omega = \arg\min_{\boldsymbol{\gamma} \in \mathbb{R}^n} \frac{1}{n}\sum_{i=1}^n \ell_{in}(\omega, (\mathbf{K}\,\boldsymbol{\gamma})_i, y_i) + \frac{\lambda}{2}\,\boldsymbol{\gamma}^\top \mathbf{K}\,\boldsymbol{\gamma}. \tag{$\widehat{\text{KBO}}$}$$

Here, $\mathbf{K} \in \mathbb{R}^{n \times n}$ and $\overline{\mathbf{K}} \in \mathbb{R}^{m \times n}$ are matrices containing the pairwise kernel similarities between the data points, *i.e.*, $\mathbf{K}_{ij} \coloneqq K(x_i, x_j)$ and $\overline{\mathbf{K}}_{ij} \coloneqq K(\tilde{x}_i, x_j)$, while $\boldsymbol{\gamma}$ is a parameter vector in $\mathbb{R}^n$ representing the inner-level variables. The optimal solution $\hat{\boldsymbol{\gamma}}_\omega$ enables recovering the prediction function $\hat{h}_\omega$ by linearly combining kernel evaluations at inner-level samples, *i.e.*, $\hat{h}_\omega = \sum_{i=1}^n (\hat{\boldsymbol{\gamma}}_\omega)_i K(x_i, \cdot)$. The formulation in ($\widehat{\text{KBO}}$) enables deriving an expression for the gradient $\nabla\widehat{\mathcal{F}}(\omega)$ in terms of

the Jacobian $\partial_\omega \hat{\gamma}_\omega$ by direct application of the chain rule. Unlike $\partial_\omega h_\omega^\star$, which requires solving the *infinite-dimensional* linear system in Equation (1), $\partial_\omega \hat{\gamma}_\omega$ can be obtained by solving a *finite-dimensional* linear system using the implicit function theorem (see Proposition C.1 of Appendix C). Hence, ($\widehat{\text{KBO}}$) falls into a class of optimization problems for which a rich body of literature has proposed practical and scalable algorithms, leveraging the expression of $\nabla \widehat{\mathcal{F}}(\omega)$ [38, 3, 24]. Solving ($\widehat{\text{KBO}}$) thus provides a practical way to approximate the solution of the original population problem (KBO), as proposed by several prior works on bilevel optimization involving kernel methods [39, 41].

**Non-applicability of existing results.** Despite its practical advantages, the above approach yields algorithms that are *not* directly amenable to a statistical analysis. The key challenge is to be able to control the approximation error between the true gradient $\nabla \mathcal{F}(\omega)$ and its approximation $\nabla \widehat{\mathcal{F}}(\omega)$ as the sample sizes $n$ and $m$ increase. Existing statistical analyses for bilevel optimization, such as [7, 78], consider objectives that are expectations or finite sums of point-wise losses, as we do here. While these results can be applied to our setting for each *fixed $n$*, they do not capture the generalization behavior as $n$ grows. In particular, they require both the inner- and outer-level parameters to lie in spaces of fixed dimensions, that are independent of $n$ and $m$. That is because these parameters are expected to converge to some fixed vectors as $n, m \to +\infty$. In contrast, in our setting, the inner-level parameter $\gamma$ lies in $\mathbb{R}^n$, so its dimension grows with $n$ and is not expected to converge to any well-defined object. Existing *non-kernel* generalization bounds are discussed in Appendix A.

**Relation to instrumental variable regression.** Our formulation is related to instrumental variable regression, which is a special case, but differs in that we study *regularized* inner problems with a fixed $\lambda$, independent of the sample sizes. In contrast, the instrumental variable regression literature typically considers *un-regularized* population problems ($\lambda = 0$), for which a *closed-form* expression of the inner minimizer is available [32, 68, 76, 45]. Our contribution lies in an orthogonal direction: we handle *more general* inner objectives, for which even the analysis of regularized problems raises new obstacles that had not been tackled before. Moreover, some prior works have provided convergence rates in the instrumental variable regression setting [1, 2, 22, 23]. Yet, these studies focus on *asymptotic* results with sieve estimators, meanwhile we leverage the RKHS structure to provide *finite-sample* bounds. More recently, Meunier et al. [51] established *minimax optimal rates* under *source assumptions* by exploiting bounds for vector-valued kernel ridge regression [46] via a *spectral filtering* technique [52]. However, this approach is not applicable to our case, as it requires the losses to be quadratic in $\omega$, as further discussed in Appendix A.

Next, we provide an equivalent expression for $\nabla \widehat{\mathcal{F}}(\omega)$, essential for our analysis in Section 4.

### 3.2 Plug-in estimator of the total gradient

We now consider an *a priori* different approach for approximating the total gradient $\nabla \mathcal{F}(\omega)$ based on direct plug-in estimation from Equation (2), and show that it recovers the previously introduced estimator $\nabla \widehat{\mathcal{F}}(\omega)$. Such approach consists in replacing all expectations in Equation (2) by empirical averages, then replacing $h_\omega^\star$ and $a_\omega^\star$ by their finite-sample estimates $\hat{h}_\omega$ and $\hat{a}_\omega$. This yields the following estimator of the total gradient:

$$\widehat{\nabla \mathcal{F}}(\omega) = \frac{1}{m} \sum_{j=1}^m \partial_\omega \ell_{out}(\omega, \hat{h}_\omega(\tilde{x}_j), \tilde{y}_j) + \frac{1}{n} \sum_{i=1}^n \partial_{\omega,v}^2 \ell_{in}(\omega, \hat{h}_\omega(x_i), y_i) \hat{a}_\omega(x_i). \qquad (4)$$

Just as in Section 3.1, $h_\omega^\star$ can be estimated by $\hat{h}_\omega$, the minimizer of the empirical objective $h \mapsto \widehat{L}_{in}(\omega, h)$. Similarly, $a_\omega^\star$ can be approximated by $\hat{a}_\omega$, the minimizer of $a \mapsto \widehat{L}_{adj}(\omega, a)$ defined as:

$$\widehat{L}_{adj}(\omega, a) = \frac{1}{2n} \sum_{i=1}^n \partial_v^2 \ell_{in}(\omega, \hat{h}_\omega(x_i), y_i) a^2(x_i) + \frac{1}{m} \sum_{j=1}^m \partial_v \ell_{out}(\omega, \hat{h}_\omega(\tilde{x}_j), \tilde{y}_j) a(\tilde{x}_j) + \frac{\lambda}{2} \|a\|_{\mathcal{H}}^2,$$
$$\qquad (5)$$

which serves as the empirical counterpart of the adjoint objective $L_{adj}$ given in Equation (3). Both functions $\hat{h}_\omega$ and $\hat{a}_\omega$ can be expressed as linear combinations of kernel evaluations with some given parameter vectors whose dimensions *increase* with the sample size $n$ (see Proposition C.1 for $\hat{h}_\omega$ and Lemma C.2 for $\hat{a}_\omega$, both in Appendix C). However, these parameters are not required to compute the plug-in estimator $\widehat{\nabla \mathcal{F}}(\omega)$ in Equation (4), since only the function values of $\hat{h}_\omega$ and $\hat{a}_\omega$ are needed.

This property is precisely what makes $\widehat{\nabla \mathcal{F}}(\omega)$ *suitable for a statistical analysis*. Indeed, its estimation error depends on the approximation errors of $\hat{h}_\omega$ and $\hat{a}_\omega$, which always belong to the same space $\mathcal{H}$ regardless of the sample size, and are expected to approach their population counterparts. This *contrasts* with $\nabla \widehat{\mathcal{F}}(\omega)$ obtained by implicit differentiation, whose analysis would need controlling the behavior of the vector $\hat{\gamma}_\omega$ that resides in a growing-dimensional space as $n \to +\infty$.

The next proposition establishes a link between practical applications and theoretical analysis by demonstrating that, surprisingly, both estimators $\nabla \widehat{\mathcal{F}}(\omega)$ and $\widehat{\nabla \mathcal{F}}(\omega)$ are precisely *equal*.

**Proposition 3.1.** *Under Assumptions (A) to (E), the gradient $\nabla \widehat{\mathcal{F}}(\omega)$ of the plug-in estimator $\widehat{\mathcal{F}}(\omega)$ of $\mathcal{F}(\omega)$ defined in* (KBO) *is equal to the plug-in estimator $\widehat{\nabla \mathcal{F}}(\omega)$ of the total gradient $\nabla \mathcal{F}(\omega)$ introduced in Equation* (4).

Proposition 3.1 is proved in Appendix C and relies on an application of the representer theorem [64] to provide explicit expressions for both estimators in terms of $\hat{\gamma}_\omega$, kernel matrices $\mathbf{K}$ and $\overline{\mathbf{K}}$ and partial derivatives of the point-wise objectives $\ell_{in}$ and $\ell_{out}$. Both expressions are then shown to be equal using optimality conditions on the parameters defining $\hat{a}_\omega$. The result in Proposition 3.1 precisely says that the operations of differentiation and plug-in estimation commute in the case of (KBO). Such a commutativity property does not necessarily hold anymore if one considers spaces other than an RKHS, such as $L_2$ [58, Appendix F]. The main difficulty arises from the argmin constraint and the use of implicit differentiation, which may introduce non-linear dependencies between inner- and outer-level variables, making the exchange of differentiation and discretization nontrivial. Next, we leverage the expression of the plug-in estimator $\widehat{\nabla \mathcal{F}}(\omega)$ to provide generalization bounds.

# 4 Generalization bounds for (KBO)

In this section, we present our main result: a *maximal inequality* that controls how well both $\mathcal{F}$ and $\nabla \mathcal{F}$ are approximated by their empirical counterparts, uniformly over a compact subset $\Omega$ of $\mathbb{R}^d$.

## 4.1 Maximal inequalities for (KBO)

The following theorem provides *finite-sample bounds* on the uniform approximation errors on the objective and its gradient in expectation over both inner- and outer-level samples.

**Theorem 4.1** (Maximal inequalities)**.** *Fix any compact subset $\Omega$ of $\mathbb{R}^d$. Under Assumptions (A) to (E), the following maximal inequalities hold:*

$$\mathbb{E}\left[\sup_{\omega \in \Omega}\left|\mathcal{F}(\omega) - \widehat{\mathcal{F}}(\omega)\right|\right] \leq C\left(\frac{1}{\sqrt{m}} + \frac{1}{\sqrt{n}}\right), \mathbb{E}\left[\sup_{\omega \in \Omega}\left\|\nabla \mathcal{F}(\omega) - \widehat{\nabla \mathcal{F}}(\omega)\right\|\right] \leq C\left(\frac{1}{\sqrt{m}} + \frac{1}{\sqrt{n}}\right)$$

*where the expectation is taken over the finite samples, and $C$ is a constant that depends only on $\Omega$, the dimension $d$, the regularization parameter $\lambda$, $\kappa$, and local upper bounds on $\ell_{in}$, $\ell_{out}$, and their partial derivatives over suitable compact sets.*

Theorem 4.1 states that the estimation error can be decomposed into two contributions each resulting from finite-sample approximation of $L_{in}$ and $L_{out}$ with a *parametric rate* of $1/\sqrt{m}$ and $1/\sqrt{n}$ up to a constant factor $C$. We provide a detailed expression for the constant in Theorem E.7 of Appendix E. The restriction to a compact subset $\Omega$ instead of the whole space $\mathbb{R}^d$ allows controlling the complexity of some function classes indexed by the parameter $\omega$. Without further assumptions on the objectives, we obtain a constant $C$ that can grow with the diameter of the subset $\Omega$.

**Role of $\lambda$.** The regularization parameter $\lambda$ simultaneously controls the strong convexity of the inner objective (see Proposition B.3 of Appendix B.1), the boundedness and Lipschitz continuity of the inner solutions (see Appendix D.1), the smoothness of the outer objective (see Proposition D.4 of Appendix D.2), the modulus of continuity of $L_{in}$, $\widehat{L}_{in}$, $L_{out}$, $\widehat{L}_{out}$ and their partial derivatives (see Appendix E.1), and maximal inequalities for certain processes (see Appendix E.2). Larger values of $\lambda$ yield smoother problems that are faster to optimize with larger step sizes, but introduce a larger statistical bias, while smaller values of $\lambda$ reduce bias but make optimization and generalization more delicate, with the error tending to $+\infty$ as $\lambda \to 0$. Selecting $\lambda$ therefore involves a trade-off between *bias* (regularized vs un-regularized problems), *variance* (finite sample vs population, as

done in our study), and *optimization efficiency* (step size). Under our assumptions, the dependence of the constants on $\lambda$ cannot be quantified, precluding generalization guarantees as $\lambda \to 0$. Stronger assumptions are required to obtain a quantitative control on $\lambda$.

**Probabilistic and variance bounds.** The most difficult quantity to control is the expectation of the maximal differences, which we have established. Once this expectation is bounded, a high-probability bound on the maximal differences can be derived via Markov's inequality. Moreover, since these differences are bounded (see Proposition E.4), we can also bound their variance. Indeed, let $Z \in [0, z]$ be a random variable representing the maximal difference between $\mathcal{F}$ or $\nabla \mathcal{F}$ and their respective plug-in estimators. Then, $\mathrm{Var}(Z/z) \leq \mathbb{E}[(Z/z)^2] \leq \mathbb{E}[Z/z]$, which implies that $\mathrm{Var}(Z) \leq z\mathbb{E}[Z]$.

We outline the general proof strategy for Theorem 4.1 in the following section, with a full proof provided in Appendix E.

## 4.2 General proof strategy for Theorem 4.1

The main strategy behind the proof of Theorem 4.1 in Appendix E consists of three steps: (step 1) obtaining a point-wise error decomposition of the errors into manageable error terms that holds almost surely for any $\omega \in \Omega$, then applying maximal inequalities to suitable empirical processes (step 2) and some degenerate $U$-processes (step 3) to control each of these terms. The final error bounds are obtained by combining all these bounds as shown in Appendix E.3.

**Step 1: point-wise error decomposition.** A main challenge in controlling the errors in Theorem 4.1 is the non-linear dependence of both estimators $\widehat{\mathcal{F}}(\omega)$ and $\widehat{\nabla \mathcal{F}}(\omega)$ on the empirical distributions, as they are obtained via a plug-in procedure. We address this by breaking down the errors into components based on the discrepancies between expected values and their empirical counterparts of individual point-wise losses and their derivatives, all evaluated at the optimal solution $h_\omega^\star$. Specifically, we denote by $\delta_\omega^{out}$ and $\delta_\omega^{in}$ the errors on the objectives defined as $\delta_\omega^{out} := |L_{out}(\omega, h_\omega^\star) - \widehat{L}_{out}(\omega, h_\omega^\star)|$ and $\delta_\omega^{in} := |L_{in}(\omega, h_\omega^\star) - \widehat{L}_{in}(\omega, h_\omega^\star)|$. Moreover, we quantify the errors between the partial derivatives of these objectives and their empirical counterparts. To simplify our proof outline, we slightly abuse notation by denoting $\partial_h \delta_\omega^{out}$, $\partial_h \delta_\omega^{in}$, $\partial_\omega \delta_\omega^{out}$, $\partial_{\omega,h}^2 \delta_\omega^{in}$, and $\partial_h^2 \delta_\omega^{in}$ to refer to these errors in terms of partial derivatives. For instance, $\partial_h \delta_\omega^{out}$ is defined as $\|\partial_h L_{out}(\omega, h_\omega^\star) - \partial_h \widehat{L}_{out}(\omega, h_\omega^\star)\|_{\mathcal{H}}$, with similar definitions for the other terms (see Appendix E.1). We get $|\mathcal{F}(\omega) - \widehat{\mathcal{F}}(\omega)| \leq C(\delta_\omega^{out} + \partial_h \delta_\omega^{in})$ and $\|\nabla \mathcal{F}(\omega) - \widehat{\nabla \mathcal{F}}(\omega)\| \leq C(\partial_\omega \delta_\omega^{out} + \partial_h \delta_\omega^{out} + \partial_h^2 \delta_\omega^{in} + \partial_{\omega,h}^2 \delta_\omega^{in} + \partial_h \delta_\omega^{in})$. Proposition E.4 formalizes this step and includes the exact constants. The error terms in both decompositions are amenable to a statistical analysis using empirical process theory as we discuss next.

**Step 2: maximal inequalities for empirical processes.** Some of the error terms, namely $\delta_\omega^{out}$ and $\partial_\omega \delta_\omega^{out}$, can be controlled directly using empirical process theory. For example, $\delta_\omega^{out}$ is associated to the family of random functions $\sqrt{m}(L_{out}(\omega, h_\omega^\star) - \widehat{L}_{out}(\omega, h_\omega^\star))_{\omega \in \Omega}$, which defines an empirical process, a scaled and centered empirical average of real-valued functions indexed by the parameter $\omega$. Thus, provided that suitable estimates of the class complexity are available (as measured by its packing number in Proposition F.1 of Appendix F), which are easy to obtain in our setting, we show in Proposition E.5 of Appendix E.2 that a maximal inequality of the following form follows from classical results on empirical processes: $\mathbb{E}_{\mathbb{Q}}[\sup_{\omega \in \Omega} \delta_\omega^{out}] \leq C/\sqrt{m}$.

**Step 3: maximal inequalities for degenerate $U$-processes.** Step 2 cannot be readily applied to the remaining terms involving partial derivatives w.r.t. $h$ $(\partial_h \delta_\omega^{out}, \partial_h \delta_\omega^{in}, \partial_{\omega,h}^2 \delta_\omega^{in}, \partial_h^2 \delta_\omega^{in}) =: \mathbf{D}_h$. These are associated to processes that are not real-valued anymore, but take values in an infinite-dimensional space. In fact, one could apply step 2 to get an error per dimension, but then summing the errors yields a divergent sum. While the recent work in [56] develops an empirical process theory for functions taking values in a vector space, the provided complexity estimates would result in an unfavorable dependence on the sample size. Instead, we leverage the structure of the RKHS to control these errors using maximal inequalities for suitable *degenerate $U$-processes of order* 2 indexed by the parameter $\omega$ and for which such inequalities were provided in the seminal works of Sherman [67], Nolan and Pollard [54]. $U$-processes of order 2 are generalization of empirical processes and involve empirical averages of real-valued functions which depend on *pairs* of samples, instead of a single one as in empirical processes. In our case, these functions arise when taking the square of any term in $\mathbf{D}_h$ and exploiting the reproducing property of the RKHS. This approach, presented in Proposition E.6 of Appendix E.2, allows us to obtain maximal inequalities for the terms in $\mathbf{D}_h$. For example, it is of the

following form for $\partial_h \delta_\omega^{out}$: $\mathbb{E}_\mathbb{Q}\left[\sup_{\omega \in \Omega} \partial_h \delta_\omega^{out}\right] \leq C/\sqrt{m}$. Combining the maximal inequalities from steps 2 and 3 with the error decomposition from step 1 allows to obtain the result of Theorem 4.1.

**Discussion.** Alternative approaches to $U$-processes could be used to derive generalization bounds, although these would result in a degraded sample dependence. Specifically, one could employ a variational formulation of the RKHS norm appearing in some of the error terms, such as $\partial_h \delta_\omega^{out}$, to express them as the error of some real-valued empirical process to which standard results could be applied. However, this comes at the cost of considering processes indexed not only by the finite-dimensional parameter $\omega$, but also by functions in the unit RKHS ball. As a result, these families have much larger complexities as measured by their covering/packing numbers [77, Lemma D.2], which directly impacts the generalization rate. In contrast, our proposed approach *bypasses* this challenge by using *real-valued U-processes indexed by finite-dimensional parameters*, at the expense of employing a more general empirical process theory for degenerate $U$-processes [67].

To illustrate the implications of Theorem 4.1, we next provide convergence results for bilevel gradient methods.

### 4.3 Applications to empirical bilevel gradient methods

A typical strategy to solve (KBO) is to obtain empirical samples and apply a bilevel optimization algorithm to ($\widehat{\text{KBO}}$), for which our results offer statistical guarantees. Below, we present the generalization error for bilevel gradient descent. Generalization results for the projected bilevel gradient descent are directed to Appendix E.4.

**Bilevel gradient descent.** It is the simplest gradient-based method for solving the unconstrained (KBO) problem, *i.e.*, when $\mathcal{C} = \mathbb{R}^d$. It performs the update $\omega_{t+1} = \omega_t - \eta \nabla \widehat{\mathcal{F}}(\omega_t)$ for all $t \geq 0$, where $\eta > 0$ is the step size. The algorithm requires access to the strongly convex inner-level solution and its derivative, which can be obtained using implicit differentiation.

**Corollary 4.2** (Generalization for bilevel gradient descent). *Consider Assumptions (A) to (E) and fix $\lambda > 0$. Assume further that $\mathbf{K}$ in ($\widehat{\text{KBO}}$) is almost surely definite, and that there exists $c > 0$ such that $\inf_{\omega,v,y} \ell_{out}(\omega,v,y) - c\|\omega\|^2 > -\infty$. Fix $\omega_0 \in \mathbb{R}^d$ and let $\omega_{t+1} = \omega_t - \eta \nabla \widehat{\mathcal{F}}(\omega_t)$ for all $t \geq 0$, where $\eta > 0$ is the step size. Then, there exist constants $\bar{\eta} > 0$ and $\bar{c} > 0$ such that for any $0 < \eta < \bar{\eta}$ any $t > 0$, the following holds:*

$$\mathbb{E}\left[\min_{i=0,\ldots,t} \|\nabla \mathcal{F}(\omega_i)\|\right] \leq \bar{c}\left(\frac{1}{\sqrt{m}} + \frac{1}{\sqrt{n}} + \frac{1}{\sqrt{t+1}}\right), \mathbb{E}\left[\limsup_{i \to \infty} \|\nabla \mathcal{F}(\omega_i)\|\right] \leq \bar{c}\left(\frac{1}{\sqrt{m}} + \frac{1}{\sqrt{n}}\right)$$

The additional assumption on $\ell_{out}$ serves as a device to ensure the almost sure boundedness of the sequence *a priori*. It is rather mild, as it can be enforced by a small perturbation of the form $(\omega,v,y) \mapsto \ell_{out}(\omega,v,y) + c\|\omega\|^2$, assuming $\ell_{out} \geq 0$, which is typical in applications. Any other device that ensures *a priori* boundedness could also be considered. The assumption on $\mathbf{K}$ is satisfied almost surely for most commonly used kernels. The proof of Corollary 4.2 follows from Theorem 4.1 and can be found in Appendix E.4. Corollary 4.2 also highlights a *key algorithmic insight*: the convergence of the bilevel method requires striking a balance between *data availability* (sample sizes $n$ and $m$) and *computational budget* (number of gradient steps). This trade-off arises because the total convergence error combines a *statistical component*, due to approximating the population gradient from finite samples, and an *optimization component*, due to performing only a limited number of (projected) gradient steps. Ensuring the right balance when designing practical algorithms prevents either insufficient data or limited computation from dominating the overall error.

## 5 Numerical experiments

**Setup.** To empirically validate our theoretical results, we consider the instrumental variable regression problem discussed in Section 2.2, in which we assume a linear dependence on $\omega$ for the function $f_\omega$ of the form $f_\omega(t) = \omega^\top \phi(t)$, where $\phi(t) \in \mathbb{R}^d$ denotes the feature map. We chose this particular problem because it allows us to derive closed-form expressions of the exact value function and its gradient, as well as their plug-in estimators, which are detailed in Appendix I.2. We use the Gaussian kernel and follow the experimental setup of Singh et al. [68], generating synthetic data that remain fixed across all runs. We vary $n$ between 100 and 5,000, setting $m = n$. The case

$n \neq m$ is analyzed separately in Appendix I.5. For the instrumental variable $x$, we consider two distributions: a $p$-dimensional standard Gaussian and a $p$-dimensional Student's $t$-distribution with degrees of freedom $\nu \in \{2.1, 2.5, 2.9\}$. Further details on the experimental setup are provided in Appendix I.4. We optimize the outer loss in ($\widehat{\text{KBO}}$) using gradient descent, where the step size is selected using backtracking line search and $\omega_0$ is randomly drawn from $\mathcal{U}(0, 1)^d$. The stopping criterion is when $\|\nabla\widehat{\mathcal{F}}(\omega_i)\| \leq 10^{-5}$, where $i$ is the iteration index. Our code is available at https://github.com/fareselkhoury/KBO.

**Scalable approximations for $\mathcal{F}$ and $\nabla\mathcal{F}$.** Since the expressions of $\mathcal{F}$ and $\nabla\mathcal{F}$ involve expectations, they are intractable to compute exactly. We approximate them accurately using their plug-in estimators $\widehat{\mathcal{F}}$ and $\widehat{\nabla\mathcal{F}}$, derived in Appendix I.2, and evaluate them using a large number of samples. To make this computation scalable, we approximate the kernel using a *random Fourier features* approximation [60], as detailed in Appendix I.3. Specifically, we use 1,000,000 samples and 26,000 random features, and handle memory constraints via a block decomposition strategy with a block size of 1,000.

**Results.** The plots in Figure 2 show the generalization behavior as a function of the number of inner samples $n$. (a) and (b) display the generalization error at initialization for the value function and the gradient, respectively. (c) presents the generalization bound for the gradient norm at the final iteration, while (d) shows the bound for the minimum gradient norm across all iterations. These results align with our theoretical findings, as all curves closely follow the expected theoretical slope. Additionally, Figure 3 of Appendix I.5 shows that balanced sample sizes lead to improved optimization behavior.

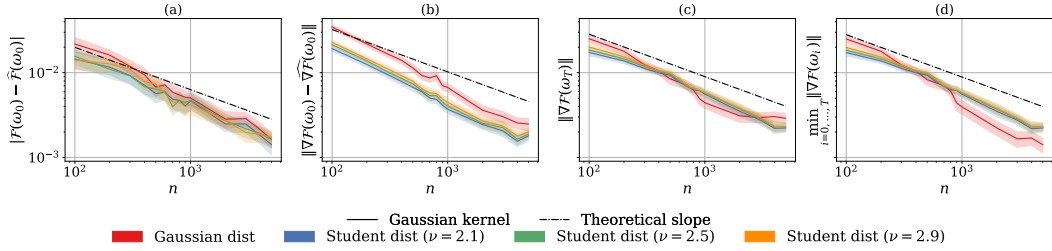

Figure 2: Illustration of gradient descent on ($\widehat{\text{KBO}}$) for the instrumental variable regression task using synthetic data. The plots are averaged over 50 runs and displayed on a log-log scale. The line represents the mean across all runs, and the shaded region indicates the 95% confidence interval.

# 6 Conclusion and perspectives

**Summary.** In this work, we established the first generalization bounds for (KBO). These results are crucial for understanding the generalization properties of algorithms for solving (KBO). They offer rigorous guarantees on the algorithm's performance on unseen data—a fundamental criterion for any algorithmic design—and help control overfitting. Given that our bounds are of order $\mathcal{O}(1/\sqrt{m} + 1/\sqrt{n})$, this highlights the equal importance of both outer- and inner-level sample sizes to the overall generalization error. Our findings can impact current practices, particularly in hyperparameter optimization, where the validation dataset is typically much smaller than the training set.

**Limitations and future work.** This paper takes a first step toward providing generalization results for bilevel gradient-based methods in a nonparametric setting. While our theoretical analysis focused on a *full-batch* bilevel setting with *exact gradients*, extending this framework to *stochastic* variants, such as those in [3, 29, 21, 24], remains an open challenge. A promising direction would be to consider *approximate kernel representations*, such as random Fourier features or neural tangent kernels, which enable scalable learning using kernel methods while preserving useful theoretical properties. Furthermore, the constants in our bounds are likely *conservative*; we did not investigate their tightness or potential for improvement. A deeper analysis of their optimality could provide valuable insights and constitutes an avenue worth exploring. Additionally, providing generalization guarantees for the un-regularized problem, possibly in the form of minimax optimal rates for (KBO), is a worthwhile future direction. This requires controlling the constants as $\lambda \to 0$, provided additional source assumptions are made, as discussed in [69, 70]. Finally, broadening our framework to cover *non-smooth losses*, such as the hinge loss in SVMs, is an interesting direction for future work.

## Acknowledgments and Disclosure of Funding

This work was supported by the ANR project BONSAI (grant ANR-23-CE23-0012-01). EP acknowledges funding from the AI Interdisciplinary Institute ANITI through the French Investments for the Future – PIA3 program under grant agreement ANR-19-PI3A-0004; the Air Force Office of Scientific Research, Air Force Materiel Command, USAF under grant number FA8655-22-1-7012; TSE-P; the ANR project Chess (grant ANR-17-EURE-0010); the ANR project Regulia; and the Institut Universitaire de France (IUF). EP and SV acknowledge support from the ANR project MAD. SV also thanks the PEPR PDE-AI program (ANR-23-PEIA-0004) and the 3IA Côte d'Azur Chair BOGL.

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

# Appendices

**Roadmap.** In Appendix A, we review existing non-kernel generalization bounds in the bilevel optimization literature and explain why minimax optimal rates cannot be obtained in our setting. We begin the theoretical appendices by presenting and establishing regularity properties of the objective functions in Appendix B. In Appendix C, we introduce the gradient estimators. Appendix D is dedicated to proving the boundedness and Lipschitz continuity of $h_\omega^\star$ and $\hat{h}_\omega$, along with local boundedness and Lipschitz properties of $\ell_{in}$, $\ell_{out}$, and their derivatives. The generalization results are provided in Appendix E. In Appendix F, we establish maximal inequalities for bounded and Lipschitz families of functions. Differentiability properties of the objectives are studied in Appendix G. Appendix H contains auxiliary technical lemmas used throughout the proofs. Finally, further details on the experiments and additional numerical results are provided in Appendix I.

**Notations.** $\|\cdot\|$ denotes the Euclidean norm in $\mathbb{R}^d$, $\|\cdot\|_\mathcal{H}$ denotes the norm in the RKHS $\mathcal{H}$, $\|\cdot\|_{\mathrm{op}}$ denotes the operator norm, and $\|\cdot\|_{\mathrm{HS}}$ denotes the Hilbert-Schmidt norm. $\langle\cdot,\cdot\rangle_\mathcal{H}$ denotes the inner product on $\mathcal{H}$, and $\langle\cdot,\cdot\rangle_{\mathrm{HS}}$ denotes the Hilbert-Schmidt inner product. $K(x,\cdot)$ denotes the feature map, for any $x \in \mathcal{X}$. For any two normed spaces $E$ and $F$, $\mathcal{L}(E,F)$ denotes the space of continuous linear operators from $E$ to $F$. For any two probability distributions $\mathcal{P}$ and $\mathcal{Q}$, $\mathcal{P} \otimes \mathcal{Q}$ denotes the product measure of $\mathcal{P}$ and $\mathcal{Q}$. Given two Hilbert spaces $(H_1, \langle\cdot,\cdot\rangle_{H_1})$ and $(H_2, \langle\cdot,\cdot\rangle_{H_2})$, the tensor product of $u \in H_1$ and $v \in H_2$, denoted by $u \otimes v$, is an operator from $H_2$ to $H_1$ defined, for any $e \in H_2$, as $(u \otimes v)e = u\langle v,e\rangle_{H_2}$. For any $v_1,\ldots,v_n \in \mathbb{R}$, $\mathbf{diag}(v_1,\ldots,v_n) \in \mathbb{R}^{n\times n}$ denotes a diagonal matrix of size $n \times n$, where the diagonal entries are $v_1,\ldots,v_n$ and all the off-diagonal entries are 0. $\mathbb{1}_m$ denotes a vector of size $m$ where all entries are 1. $\mathbb{1}_{n\times n}$ denotes the identity matrix of size $n$. For any vector space $V$ over $\mathbb{R}$, $\mathrm{Id}_V$ denotes the identity operator on $V$. Given a compact set $\mathcal{K}$, $\mathrm{diam}(\mathcal{K})$ denotes its diameter. $v^\top$ denotes the transpose of either a vector or a matrix, depending on the context.

# Contents

## A   Further Discussion

**Existing generalization bounds for the non-kernel case.** Bao et al. [7] laid foundational work towards understanding generalization in bilevel optimization by analyzing uniform stability in full-batch bilevel optimization. Their generalization criterion compares the population outer loss evaluated at the output of a randomized algorithm to the empirical outer loss evaluated using the same algorithm. Given a number $\kappa$ between 0 and 1, they obtain a decay at a rate of $\mathcal{O}(T^\kappa/m)$ for unrolled optimization, which decreases as $1/m$ in outer sample size, but increases with the number of outer iterations $T$ made. This criterion differs from ours, which instead compares the population outer objective at the theoretically optimal inner solution $h_\omega^\star$ to the empirical loss evaluated at the empirical solution $\hat{h}_\omega$. Complementing this upper bound, Wang et al. [74] established lower bounds on the uniform stability of gradient-based bilevel algorithms, demonstrating a rate of $\Omega(1/m)$. Building on [7], Zhang et al. [78] extended the analysis to stochastic bilevel optimization, establishing on-average stability bounds and deriving a generalization rate of $\mathcal{O}(1/\sqrt{m})$ due to the presence of stochastic gradients. In a related context, Oymak et al. [55] studied generalization in neural architecture search using a bilevel formulation, showing that approximate inner solutions and Lipschitz continuity of the outer loss yield a generalization bound of $\mathcal{O}(1/\sqrt{m} + 1/\sqrt{n})$. Arora et al. [5] investigated representation learning for imitation learning via bilevel optimization, offering generalization bounds of order $\mathcal{O}(1/\sqrt{m})$ that depend both on the size of the dataset and the stability of learned representations.

**Difficulty of obtaining minimax rates in our setting.** Although spectral filtering yields minimax rates [51] in the kernel instrumental variable regression setting [68], it fundamentally relies on a linear operator representation of the inner minimizer, typically characterized through the spectral decomposition of a compact, self-adjoint covariance operator (see, *e.g.*, [20, 8]). This formulation allows one to apply functional calculus on the spectrum, with filter functions (such as Tikhonov regularization and truncated SVD) controlling the contribution of small eigenvalues [27, 49]. Under suitable source conditions, this enables the derivation of minimax optimal convergence rates for kernel ridge regression and other problems involving quadratic losses. In our framework, however, the inner objective is generally not quadratic in $\omega$, and the mapping $\omega \mapsto h_\omega^\star$ is nonlinear. Moreover, we consider a fixed $\lambda$, in contrast to the vanishing-regularization regimes where spectral filtering is most effective for minimax analysis. As a result, the source condition assumption and the spectral filtering tools that underpin minimax guarantees in the quadratic case do not apply directly in our setting.

## B   Regularity and Differentiability Results

### B.1   Regularity of the objectives

The following propositions establish differentiability of considered objectives. We defer their proof to Appendix G.

**Proposition B.1** (Differentiability of $L_{in}$ and $L_{out}$)**.** *Under Assumptions (A) to (D), for any $(\omega, h) \in \mathbb{R}^d \times \mathcal{H}$, the functions $L_{in}$ and $L_{out}$ admit finite values at $(\omega, h)$, are jointly differentiable in $(\omega, h)$, with gradients given by:*

$$\partial_\omega L_{out}(\omega, h) = \mathbb{E}_\mathbb{Q}\left[\partial_\omega \ell_{out}(\omega, h(x), y)\right] \in \mathbb{R}^d, \partial_h L_{out}(\omega, h) = \mathbb{E}_\mathbb{Q}\left[\partial_v \ell_{out}(\omega, h(x), y)K(x, \cdot)\right] \in \mathcal{H},$$

$$\partial_\omega L_{in}(\omega, h) = \mathbb{E}_\mathbb{P}\left[\partial_\omega \ell_{in}(\omega, h(x), y)\right] \in \mathbb{R}^d, \partial_h L_{in}(\omega, h) = \mathbb{E}_\mathbb{P}\left[\partial_v \ell_{in}(\omega, h(x), y)K(x, \cdot)\right] + \lambda h \in \mathcal{H}.$$

Similarly, the empirical estimates $\widehat{L}_{in}$ and $\widehat{L}_{out}$ admit finite values, and are differentiable with gradients admitting similar expressions as above with $\mathbb{P}$ and $\mathbb{Q}$ replaced by their empirical estimates $\widehat{\mathbb{P}}_n$ and $\widehat{\mathbb{Q}}_m$.

**Proposition B.2** (Differentiability of $\partial_h L_{in}$). *Under Assumptions (A) to (D), for any $(\omega, h) \in \mathbb{R}^d \times \mathcal{H}$, the function $(\omega, h) \mapsto \partial_h L_{in}(\omega, h)$ is differentiable with partial derivatives given by:*

$$\partial^2_{\omega, h} L_{in}(\omega, h) = \mathbb{E}_{\mathbb{P}} \left[ \partial^2_{\omega, v} \ell_{in}(\omega, h(x), y) K(x, \cdot) \right] \in \mathcal{L}(\mathcal{H}, \mathbb{R}^d),$$

$$\partial^2_h L_{in}(\omega, h) = \mathbb{E}_{\mathbb{P}} \left[ \partial^2_v \ell_{in}(\omega, h(x), y) K(x, \cdot) \otimes K(x, \cdot) \right] + \lambda \operatorname{Id}_{\mathcal{H}} \in \mathcal{L}(\mathcal{H}, \mathcal{H}).$$

*Moreover, for any $\omega \in \mathbb{R}^d$ and $h \in \mathcal{H}$, the operators $\partial^2_{\omega, h} L_{in}(\omega, h)$ and $\partial^2_h L_{in}(\omega, h) - \lambda \operatorname{Id}_{\mathcal{H}}$ are Hilbert-Schmidt, i.e., bounded operators with finite Hilbert-Schmidt norm. The same conclusions hold for the empirical estimate $(\omega, h) \mapsto \partial_h \widehat{L}_{in}(\omega, h)$ with partial derivatives admitting similar expressions as above with $\mathbb{P}$ replaced by its empirical estimate $\widehat{\mathbb{P}}_n$.*

**Proposition B.3** (Strong convexity of the inner objective in its second variable and invertibility of the Hessians). *Under Assumptions (A) to (E), $h \mapsto L_{in}(\omega, h)$ and $h \mapsto \widehat{L}_{in}(\omega, h)$ are $\lambda$-strongly convex for any $\omega \in \mathbb{R}^d$. Moreover, for any $\omega \in \mathbb{R}^d$ and $h \in \mathcal{H}$, the Hessian operators $\partial^2_h L_{in}(\omega, h)$ and $\partial^2_h \widehat{L}_{in}(\omega, h)$ are invertible with their operator norm bounded by $\frac{1}{\lambda}$.*

*Proof.* By Assumption (E), we know that $v \mapsto \ell_{in}(\omega, v, y)$ is convex for any $\omega \in \mathbb{R}^d$ and $y \in \mathcal{Y}$. Moreover, by Proposition B.1, $(x, y) \mapsto \ell_{in}(\omega, h(x), y)$ is integrable for any $\omega \in \mathbb{R}^d$ and $h \in \mathcal{H}$. Consequently, by integration, we directly deduce that $h \mapsto \mathbb{E}_{\mathbb{P}} [\ell_{in}(\omega, h(x), y)]$ is convex for any $\omega \in \mathbb{R}^d$. Finally, $h \mapsto L_{in}(\omega, h) := \mathbb{E}_{\mathbb{P}} [\ell_{in}(\omega, h(x), y)] + \frac{\lambda}{2} \|h\|^2_{\mathcal{H}}$ must be $\lambda$-strongly convex, for any $\omega \in \mathbb{R}^d$, as a sum of a convex function and a $\lambda$-strongly convex function. Similarly, we deduce that $h \mapsto \widehat{L}_{in}(\omega, h)$ is $\lambda$-strongly convex, for any $\omega \in \mathbb{R}^d$. Invertibility follows from the expression of the Hessian operator in Proposition B.2 $\qquad \square$

## B.2 Differentiability of the value function

**Proposition B.4** (Total functional gradient $\nabla \mathcal{F}$). *Assume Assumptions (A), (B), (D) and (E) hold. For any $\omega \in \mathbb{R}^d$, the total functional gradient $\nabla \mathcal{F}(\omega)$ satisfies:*

$$\nabla \mathcal{F}(\omega) = \partial_\omega L_{out}(\omega, h^\star_\omega) + \partial^2_{\omega, h} L_{in}(\omega, h^\star_\omega) a^\star_\omega \in \mathbb{R}^d, \tag{6}$$

*where $a^\star_\omega$ is the unique minimizer of the following quadratic objective:*

$$L_{adj}(\omega, a) := \frac{1}{2} \langle a, H_\omega a \rangle_{\mathcal{H}} + \langle a, d_\omega \rangle_{\mathcal{H}}, \quad \text{for any } a \in \mathcal{H}, \tag{7}$$

*with $H_\omega := \partial^2_h L_{in}(\omega, h^\star_\omega) : \mathcal{H} \to \mathcal{H}$ being the Hessian operator and $d_\omega := \partial_h L_{out}(\omega, h^\star_\omega) \in \mathcal{H}$.*

*Proof.* By applying Propositions B.1 and B.3, we know that $h \mapsto L_{in}(\omega, h)$ has finite values, is $\lambda$-strongly convex and Fréchet differentiable. Moreover, by Proposition B.2, $\partial_h L_{in}$ is Fréchet differentiable on $\mathbb{R}^d \times \mathcal{H}$, and, a fortiori, Hadamard differentiable. Therefore, by the functional implicit differentiation theorem [36, 58, Theorem 2.1], we deduce that the map $\omega \mapsto h^\star_\omega$ is uniquely defined and is Fréchet differentiable with Jacobian $\partial_\omega h^\star_\omega$ solving the following linear system for any $\omega \in \mathbb{R}^d$:

$$\partial^2_{\omega, h} L_{in}(\omega, h^\star_\omega) + \partial_\omega h^\star_\omega \partial^2_h L_{in}(\omega, h^\star_\omega) = 0.$$

Using that $\partial^2_h L_{in}(\omega, h^\star_\omega)$ is invertible by Proposition B.3, we can express $\partial_\omega h^\star_\omega$ as:

$$\partial_\omega h^\star_\omega = -\partial^2_{\omega, h} L_{in}(\omega, h^\star_\omega) \left( \partial^2_h L_{in}(\omega, h^\star_\omega) \right)^{-1}.$$

Furthermore, $L_{out}$ is jointly Fréchet differentiable by application of Proposition B.1, so that $\omega \mapsto \mathcal{F}(\omega)$ is also differentiable by composition of the functions $(\omega, h) \mapsto L_{out}(\omega, h)$ and $\omega \mapsto (\omega, h^\star_\omega)$. For a given $\omega \in \mathbb{R}^d$, the gradient of $\mathcal{F}$ is then given by the chain rule:

$$\nabla \mathcal{F}(\omega) = \partial_\omega L_{out}(\omega, h^\star_\omega) + \partial_\omega h^\star_\omega \partial_h L_{out}(\omega, h^\star_\omega). \tag{8}$$

Substituting the expression of $\partial_\omega h^\star_\omega$ into Equation (8) yields:

$$\nabla \mathcal{F}(\omega) = \partial_\omega L_{out}(\omega, h^\star_\omega) - \partial^2_{\omega, h} L_{in}(\omega, h^\star_\omega) \left( \partial^2_h L_{in}(\omega, h^\star_\omega) \right)^{-1} \partial_h L_{out}(\omega, h^\star_\omega).$$

To conclude, it suffices to notice that the function $a_\omega^\star$ appearing in Equation (6) must be equal to $-H_\omega^{-1} d_\omega$. Indeed, $a_\omega^\star$ is defined as the minimizer of the quadratic objective $L_{adj}(\omega, a)$ in Equation (7) which is strongly convex since the Hessian operator is lower-bounded by $\lambda \operatorname{Id}_{\mathcal{H}}$. Consequently, the minimizer $a_\omega^\star$ exists and is uniquely characterized by the optimality condition:

$$H_\omega a_\omega^\star + d_\omega = 0.$$

The above equation is a linear system in $\mathcal{H}$ whose solution is given by $a_\omega^\star := -H_\omega^{-1} d_\omega$. $\qquad\square$

## C  Gradient Estimators

**Proposition C.1** (Expression of $\nabla\widehat{\mathcal{F}}(\omega)$ by implicit differentiation)**.** *Under Assumptions (B) to (E), for any $\omega \in \mathbb{R}^d$, the gradient $\nabla\widehat{\mathcal{F}}(\omega)$ of the discretized kernel bilevel optimization problem (KBO) is given by:*

$$\nabla\widehat{\mathcal{F}}(\omega) = \frac{1}{m}\, \mathbf{D}_\omega^{\mathbf{out}}\, \mathbb{1}_m - \frac{1}{m}\, \mathbf{D}_{\omega,\mathbf{v}}^{\mathbf{in}}\, \mathbf{M}^{-1}\, \mathbf{u} \in \mathbb{R}^d,$$

*where $\mathbf{K}$ and $\overline{\mathbf{K}}$ are the Gram matrices in $\mathbb{R}^{n\times n}$ and $\mathbb{R}^{m\times n}$ with entries given by $\mathbf{K}_{ij} := K(x_i, x_j)$ and $\overline{\mathbf{K}}_{ij} := K(\tilde{x}_i, x_j)$, and $\mathbf{M} \in \mathbb{R}^{n\times n}$, $\mathbf{u} \in \mathbb{R}^n$, $\mathbf{D}_{\mathbf{v}}^{\mathbf{out}} \in \mathbb{R}^m$, $\mathbf{D}_\omega^{\mathbf{out}} \in \mathbb{R}^{d\times m}$, $\mathbf{D}_{\mathbf{v},\mathbf{v}}^{\mathbf{in}} \in \mathbb{R}^{n\times n}$, and $\mathbf{D}_{\omega,\mathbf{v}}^{\mathbf{in}} \in \mathbb{R}^{d\times n}$ are defined as:*

$$\mathbf{M} := \mathbf{K}\, \mathbf{D}_{\mathbf{v},\mathbf{v}}^{\mathbf{in}} + n\lambda \mathbb{1}_{n\times n}, \qquad\qquad \mathbf{u} := \overline{\mathbf{K}}^\top \mathbf{D}_{\mathbf{v}}^{\mathbf{out}},$$

$$\mathbf{D}_{\mathbf{v}}^{\mathbf{out}} := \left(\partial_v \ell_{out}\left(\omega, \hat{h}_\omega(\tilde{x}_j), \tilde{y}_j\right)\right)_{1\le j\le m}, \qquad \mathbf{D}_\omega^{\mathbf{out}} := \left(\partial_\omega \ell_{out}(\omega, \hat{h}_\omega(\tilde{x}_j), \tilde{y}_j)\right)_{1\le j\le m},$$

$$\mathbf{D}_{\mathbf{v},\mathbf{v}}^{\mathbf{in}} := \mathbf{diag}\left(\left(\partial_v^2 \ell_{in}(\omega, \hat{h}_\omega(x_i), y_i)\right)_{1\le i\le n}\right), \qquad \mathbf{D}_{\omega,\mathbf{v}}^{\mathbf{in}} := \left(\partial_{\omega,v}^2 \ell_{in}(\omega, \hat{h}_\omega(x_i), y_i)\right)_{1\le i\le n}.$$

*Proof.* Let $\omega \in \mathbb{R}^d$. Recall the expression of $\widehat{\mathcal{F}}(\omega)$:

$$\widehat{\mathcal{F}}(\omega) := \frac{1}{m}\sum_{j=1}^m \ell_{out}\left(\omega, \hat{h}_\omega(\tilde{x}_j), \tilde{y}_j\right)$$

$$\text{s.t.} \quad \hat{h}_\omega = \operatorname*{arg\,min}_{h\in\mathcal{H}} \widehat{L}_{in}(\omega, h) := \frac{1}{n}\sum_{i=1}^n \ell_{in}\left(\omega, h(x_i), y_i\right) + \frac{\lambda}{2}\|h\|_{\mathcal{H}}^2.$$

By the representer theorem, it is easy to see that $\hat{h}_\omega$ must be a linear combination of $K(x_1, \cdot), \ldots, K(x_n, \cdot)$:

$$\hat{h}_\omega = \sum_{i=1}^n (\hat{\gamma}_\omega)_i K(x_i, \cdot). \tag{9}$$

Hence, finding $\hat{h}_\omega$ amounts to minimizing $\widehat{L}_{in}(\omega, h)$ over the span of $(K(x_1, \cdot), \ldots, K(x_n, \cdot))$, *i.e.*, over functions $h^\gamma$ of the form $h^\gamma = \sum_{i=1}^n (\gamma)_i K(x_i, \cdot)$ for $\gamma \in \mathbb{R}^n$. Restricting the objective to such functions results in the following inner optimization problem which is finite-dimensional:

$$\hat{\gamma}_\omega := \operatorname*{arg\,min}_{\gamma\in\mathbb{R}^n} \frac{1}{n}\sum_{i=1}^n \ell_{in}\left(\omega, (\mathbf{K}\,\gamma)_i, y_i\right) + \frac{\lambda}{2}\,\gamma^\top \mathbf{K}\,\gamma,$$

where we used that $(h^\gamma(x_i))_{1\le i\le n} = \mathbf{K}\,\gamma$ and $\|h^\gamma\|_{\mathcal{H}}^2 = \gamma^\top \mathbf{K}\,\gamma$. Similarly, using that $(h^\gamma(\tilde{x}_j))_{1\le j\le m} = \overline{\mathbf{K}}\,\gamma$, we can express $\widehat{\mathcal{F}}(\omega)$ as follows:

$$\widehat{\mathcal{F}}(\omega) = \frac{1}{m}\sum_{j=1}^m \ell_{out}\left(\omega, (\overline{\mathbf{K}}\,\hat{\gamma}_\omega)_j, \tilde{y}_j\right).$$

Differentiating the above expression w.r.t. $\omega$ and applying the chain rule result in:

$$\nabla\widehat{\mathcal{F}}(\omega) = \frac{1}{m}\sum_{j=1}^m \partial_\omega \ell_{out}\left(\omega, (\overline{\mathbf{K}}\,\hat{\gamma}_\omega)_j, \tilde{y}_j\right) + \frac{1}{m}\sum_{j=1}^m (\partial_\omega \hat{\gamma}_\omega\, \overline{\mathbf{K}}^\top)_j \partial_v \ell_{out}\left(\omega, (\overline{\mathbf{K}}\,\hat{\gamma}_\omega)_j, \tilde{y}_j\right),$$

where $\partial_\omega \hat{\gamma}_\omega$ denotes the Jacobian of $\hat{\gamma}_\omega$. We can further express the above equation in matrix form to get:

$$\nabla \widehat{\mathcal{F}}(\omega) = \frac{1}{m} \mathbf{D}_\omega^{\mathbf{out}} \mathbb{1}_m + \frac{1}{m} \partial_\omega \hat{\gamma}_\omega \overline{\mathbf{K}}^\top \mathbf{D}_\mathbf{v}^{\mathbf{out}}. \tag{10}$$

Moreover, an application of the implicit function theorem[2] allows to directly express the Jacobian $\partial_\omega \hat{\gamma}_\omega$ as a solution of the following linear system obtained by differentiating the optimality condition for $\hat{\gamma}_\omega$ w.r.t. $\omega$:

$$\mathbf{D}_{\omega,\mathbf{v}}^{\mathbf{in}} \mathbf{K} + (\partial_\omega \hat{\gamma}_\omega) \underbrace{\left( \mathbf{K} \mathbf{D}_{\mathbf{v},\mathbf{v}}^{\mathbf{in}} + n\lambda \mathbb{1}_{n \times n} \right) \mathbf{K}}_{\mathbf{M}} = 0.$$

A solution of the form $\partial_\omega \hat{\gamma}_\omega = -\mathbf{D}_{\omega,\mathbf{v}}^{\mathbf{in}} \mathbf{M}^{-1}$ always exists by invertibility of the matrix $\mathbf{M}$. The result follows after replacing $\partial_\omega \hat{\gamma}_\omega$ by $-\mathbf{D}_{\omega,\mathbf{v}}^{\mathbf{in}} \mathbf{M}^{-1}$ in Equation (10). $\qquad\square$

**Lemma C.2** (Estimator of the total functional gradient). *Let $\omega \in \mathbb{R}^d$. Consider the following functional estimator:*

$$\widehat{\nabla \mathcal{F}}(\omega) = \frac{1}{m} \sum_{j=1}^{m} \partial_\omega \ell_{out}(\omega, \hat{h}_\omega(\tilde{x}_j), \tilde{y}_j) + \frac{1}{n} \sum_{i=1}^{n} \partial_{\omega,v}^2 \ell_{in}(\omega, \hat{h}_\omega(x_i), y_i) \hat{a}_\omega(x_i).$$

*Then, under Assumptions (A) to (E), $\widehat{\nabla \mathcal{F}}(\omega)$ admits the following expression:*

$$\widehat{\nabla \mathcal{F}}(\omega) = \frac{1}{m} \mathbf{D}_\omega^{\mathbf{out}} \mathbb{1}_m + \frac{1}{n} \mathbf{D}_{\omega,\mathbf{v}}^{\mathbf{in}} [\mathbf{K} \quad \mathbf{u}] \begin{bmatrix} \hat{\boldsymbol{\alpha}}_\omega \\ \hat{\beta}_\omega \end{bmatrix} \in \mathbb{R}^d,$$

*where $\mathbf{D}_\mathbf{v}^{\mathbf{out}}, \mathbf{D}_{\mathbf{v},\mathbf{v}}^{\mathbf{in}}, \mathbf{D}_\omega^{\mathbf{out}}$, and $\mathbf{D}_{\omega,\mathbf{v}}^{\mathbf{in}}$ are the same matrices given in Proposition C.1, while $\hat{\boldsymbol{\alpha}}_\omega \in \mathbb{R}^n$ and $\hat{\beta}_\omega \in \mathbb{R}$ are solutions to the linear system:*

$$\begin{bmatrix} \mathbf{M} \mathbf{K} & \mathbf{M} \mathbf{u} \\ \mathbf{u}^\top \mathbf{M} & p \end{bmatrix} \begin{bmatrix} \hat{\boldsymbol{\alpha}}_\omega \\ \hat{\beta}_\omega \end{bmatrix} = -\frac{n}{m} \begin{bmatrix} \mathbf{u} \\ v \end{bmatrix}, \tag{11}$$

*where the vector $\mathbf{u}$ and matrix $\mathbf{M}$ are the same as in Proposition C.1, while $p$ and $v$ are non-negative scalars.*

*Proof.* Let $\omega \in \mathbb{R}^d$. We start by providing an expression of $\hat{a}_\omega$ as a linear combination of the kernel evaluated at the inner training points $x_i$, *i.e.*, $K(x_i, \cdot)$, and some element $\xi \in \mathcal{H}$ that we will characterize shortly. From it, we will obtain the expression of $\widehat{\nabla \mathcal{F}}(\omega)$.

**Expression of $\hat{a}_\omega$.** Recall that $\hat{a}_\omega$ is the unique minimizer of $\widehat{L}_{adj}$ in Equation (5), which admits, for any $a \in \mathcal{H}$, the following simple expression by the reproducing property:

$$\widehat{L}_{adj}(\omega, a) = \frac{1}{2n} \sum_{i=1}^{n} \partial_v^2 \ell_{in}(\omega, \hat{h}_\omega(x_i), y_i) a^2(x_i)$$

$$+ \frac{1}{m} \left\langle a, \overbrace{\sum_{j=1}^{m} \partial_v \ell_{out}(\omega, \hat{h}_\omega(\tilde{x}_j), \tilde{y}_j) K(\tilde{x}_j, \cdot)}^{\xi} \right\rangle_{\mathcal{H}} + \frac{\lambda}{2} \|a\|_\mathcal{H}^2.$$

Hence, by application of the representer theorem, it follows that $\hat{a}_\omega$ admits an expression of the form:

$$\hat{a}_\omega = \sum_{i=1}^{n} (\hat{\boldsymbol{\alpha}}_\omega)_i K(x_i, \cdot) + \hat{\beta}_\omega \xi. \tag{12}$$

Therefore, it is possible to recover $\hat{a}_\omega$ by minimizing $a \mapsto L_{adj}(\omega, a)$ over the span of $(\xi, K(x_1, \cdot), \ldots, K(x_n, \cdot))$. Hence, to find the optimal coefficients $\hat{\boldsymbol{\alpha}}_\omega := ((\hat{\boldsymbol{\alpha}}_\omega)_i)_{1 \leq i \leq n}$ and $\hat{\beta}_\omega$,

---

[2]In the case where the matrix $\mathbf{K}$ is non-invertible, one needs to restrict $\boldsymbol{\gamma}$ to the orthogonal complement of the null space of $\mathbf{K}$. Such a restriction is valid since the resulting solution $\hat{h}_\omega$ will not depend on the component belonging to the null space of $\mathbf{K}$.

we first need to express the objective $L_{adj}$ in terms of the coefficients $\boldsymbol{\alpha} \in \mathbb{R}^n$ and $\beta \in \mathbb{R}$ for a given $a^{\boldsymbol{\alpha},\beta} \in \mathcal{H}$ of the form $a^{\boldsymbol{\alpha},\beta} = \sum_{i=1}^{n} (\boldsymbol{\alpha})_i K(x_i,\cdot) + \beta\xi$. To this end, note that the vector $(\xi(x_1),\ldots,\xi(x_n))$ is exactly equal to $\mathbf{u} = \overline{\mathbf{K}}^{\top} \mathbf{D_v^{out}}$ as defined in Proposition C.1. Moreover, using the reproducing property, we directly have:

$$(a^{\boldsymbol{\alpha},\beta}(x_i))_{1\leq i \leq n} = \begin{bmatrix}\mathbf{K} & \mathbf{u}\end{bmatrix} \begin{bmatrix}\boldsymbol{\alpha}\\\beta\end{bmatrix}, \quad \langle a^{\boldsymbol{\alpha},\beta},\xi\rangle_{\mathcal{H}} = \begin{bmatrix}\mathbf{u}^{\top} & \|\xi\|_{\mathcal{H}}^2\end{bmatrix}\begin{bmatrix}\boldsymbol{\alpha}\\\beta\end{bmatrix},$$

$$\left\|a^{\boldsymbol{\alpha},\beta}\right\|_{\mathcal{H}}^2 = \begin{bmatrix}\boldsymbol{\alpha}^{\top} & \beta\end{bmatrix}\begin{bmatrix}\mathbf{K} & \mathbf{u}\\\mathbf{u}^{\top} & \|\xi\|_{\mathcal{H}}^2\end{bmatrix}\begin{bmatrix}\boldsymbol{\alpha}\\\beta\end{bmatrix}.$$

We can therefore express the objective $\widehat{L}_{adj}$ as follows:

$$\widehat{L}_{adj}(\omega, a^{\boldsymbol{\alpha},\beta}) = \frac{1}{2n}\begin{bmatrix}\boldsymbol{\alpha}^{\top} & \beta\end{bmatrix}\begin{bmatrix}\mathbf{K}\\\mathbf{u}^{\top}\end{bmatrix}\mathbf{D_{v,v}^{in}}\begin{bmatrix}\mathbf{K} & \mathbf{u}\end{bmatrix}\begin{bmatrix}\boldsymbol{\alpha}\\\beta\end{bmatrix}$$
$$+ \frac{1}{m}\begin{bmatrix}\mathbf{u}^{\top} & \|\xi\|_{\mathcal{H}}^2\end{bmatrix}\begin{bmatrix}\boldsymbol{\alpha}\\\beta\end{bmatrix} + \frac{\lambda}{2}\begin{bmatrix}\boldsymbol{\alpha}^{\top} & \beta\end{bmatrix}\begin{bmatrix}\mathbf{K} & \mathbf{u}\\\mathbf{u}^{\top} & \|\xi\|_{\mathcal{H}}^2\end{bmatrix}\begin{bmatrix}\boldsymbol{\alpha}\\\beta\end{bmatrix}.$$

Hence, the optimal coefficients $\hat{\boldsymbol{\alpha}}_{\omega} := ((\hat{\boldsymbol{\alpha}}_{\omega})_i)_{1\leq i \leq n}$ and $\hat{\beta}_{\omega}$ are those minimizing the above quadratic form and are characterized by the following optimality condition:

$$\begin{bmatrix}\overbrace{(\mathbf{K}\,\mathbf{D_{v,v}^{in}} +n\lambda\mathbb{1}_{n\times n})\,\mathbf{K}}^{\mathbf{M}} & \overbrace{(\mathbf{K}\,\mathbf{D_{v,v}^{in}} +n\lambda\mathbb{1}_{n\times n})\,\mathbf{u}}^{\mathbf{M}}\\ \underbrace{\mathbf{u}^{\top}\,(\mathbf{K}\,\mathbf{D_{v,v}^{in}} +n\lambda\mathbb{1}_{n\times n})}_{\mathbf{M}} & \underbrace{\mathbf{u}^{\top}\,\mathbf{D_{v,v}^{in}}\,\mathbf{u} + n\lambda\,\|\xi\|_{\mathcal{H}}^2}_{p\geq 0}\end{bmatrix}\begin{bmatrix}\hat{\boldsymbol{\alpha}}_{\omega}\\\hat{\beta}_{\omega}\end{bmatrix} = -\frac{n}{m}\begin{bmatrix}\mathbf{u}\\\underbrace{\|\xi\|_{\mathcal{H}}^2}_{v\geq 0}\end{bmatrix}.$$

**Expression of $\widehat{\nabla\mathcal{F}}(\omega)$.** The result follows directly after expressing $\widehat{\nabla\mathcal{F}}(\omega)$ in vector form using the notations $\mathbf{D_\omega^{out}}$ and $\mathbf{D_{\omega,v}^{in}}$ from Proposition C.1 and recalling that $(\hat{a}_\omega(x_i))_{1\leq i \leq n} = \begin{bmatrix}\mathbf{K} & \mathbf{u}\end{bmatrix}\begin{bmatrix}\hat{\boldsymbol{\alpha}}_{\omega}\\\hat{\beta}_{\omega}\end{bmatrix}$.
$\square$

*Proof of Proposition 3.1.* Let $\omega \in \mathbb{R}^d$. Define

$$\widehat{\nabla\mathcal{F}}(\omega) = \frac{1}{m}\sum_{j=1}^{m}\partial_\omega\ell_{out}(\omega,\hat{h}_\omega(\tilde{x}_j),\tilde{y}_j) + \frac{1}{n}\sum_{i=1}^{n}\partial^2_{\omega,v}\ell_{in}(\omega,\hat{h}_\omega(x_i),y_i)\hat{a}_\omega(x_i),$$

where $\hat{h}_\omega$ and $\hat{a}_\omega$ are given by Equations (9) and (12). We will show that $\widehat{\nabla\mathcal{F}}(\omega) = \nabla\widehat{\mathcal{F}}(\omega)$. By Proposition C.1 and Lemma C.2, we know that $\nabla\widehat{\mathcal{F}}(\omega)$ and $\widehat{\nabla\mathcal{F}}(\omega)$ admit the following expressions:

$$\nabla\widehat{\mathcal{F}}(\omega) = \frac{1}{m}\mathbf{D_\omega^{out}}\,\mathbb{1}_m - \frac{1}{m}\mathbf{D_{\omega,v}^{in}}\,\mathbf{M}^{-1}\,\mathbf{u}$$
$$\widehat{\nabla\mathcal{F}}(\omega) = \frac{1}{m}\mathbf{D_\omega^{out}}\,\mathbb{1}_m + \frac{1}{n}\mathbf{D_{\omega,v}^{in}}\begin{bmatrix}\mathbf{K} & \mathbf{u}\end{bmatrix}\begin{bmatrix}\hat{\boldsymbol{\alpha}}_{\omega}\\\hat{\beta}_{\omega}\end{bmatrix}.$$

Taking the difference of the two estimators yields:

$$\widehat{\nabla\mathcal{F}}(\omega) - \nabla\widehat{\mathcal{F}}(\omega) = \frac{1}{m}\mathbf{D_{\omega,v}^{in}}\left(\mathbf{M}^{-1}\mathbf{u} + \frac{m}{n}\begin{bmatrix}\mathbf{K} & \mathbf{u}\end{bmatrix}\begin{bmatrix}\hat{\boldsymbol{\alpha}}_{\omega}\\\hat{\beta}_{\omega}\end{bmatrix}\right)$$
$$= \frac{1}{m}\mathbf{D_{\omega,v}^{in}}\,\mathbf{M}^{-1}\underbrace{\left(\mathbf{u} + \frac{m}{n}\mathbf{M}\begin{bmatrix}\mathbf{K} & \mathbf{u}\end{bmatrix}\begin{bmatrix}\hat{\boldsymbol{\alpha}}_{\omega}\\\hat{\beta}_{\omega}\end{bmatrix}\right)}_{=0},$$

where the term $\mathbf{u} + \frac{m}{n}(\mathbf{M}\,\mathbf{K}\,\hat{\boldsymbol{\alpha}}_{\omega} + \hat{\beta}_{\omega}\,\mathbf{M}\,\mathbf{u})$ is equal to 0 by definition of $\hat{\boldsymbol{\alpha}}_{\omega}$ and $\hat{\beta}_{\omega}$ as solutions of the linear system (11) of Lemma C.2.
$\square$

## D Preliminary Results

In this section, $\Omega$ is an arbitrary compact subset of $\mathbb{R}^d$ with hull($\Omega$) denoting its convex hull, which is also compact. We also consider an arbitrary fixed positive value $\Lambda$ such that $\lambda \leq \Lambda$ as this would allow us to simplify the dependence on $\lambda$ of the boundedness and Lipschitz constants.

## D.1 Boundedness and Lipschitz continuity of $h_\omega^\star$ and $\hat{h}_\omega$

**Proposition D.1** (Boundedness of $h_\omega^\star$ and $\hat{h}_\omega$). *Under Assumptions (A) to (E), the functions $\omega \mapsto \|h_\omega^\star\|_{\mathcal{H}}$ and $\omega \mapsto \|\hat{h}_\omega\|_{\mathcal{H}}$ are bounded over $\text{hull}(\Omega)$ by $\frac{B\sqrt{\kappa}}{\lambda}$, where $B :=$ $\sup_{\omega \in \text{hull}(\Omega), y \in \mathcal{Y}} |\partial_v \ell_{in}(\omega, 0, y)| > 0$. Moreover, for all $\omega \in \text{hull}(\Omega)$ and $x \in \mathcal{X}$, $h_\omega^\star(x)$ and $\hat{h}_\omega(x)$ take value in the compact interval $\mathcal{V} := \left[-\frac{B\kappa}{\lambda}, \frac{B\kappa}{\lambda}\right] \subset \mathbb{R}$.*

*Proof.* **Boundedness of $\|h_\omega^\star\|_{\mathcal{H}}$ and $\left\|\hat{h}_\omega\right\|_{\mathcal{H}}$.** Let $\omega \in \text{hull}(\Omega)$. Using Lemma H.2, we know, for any $h \in \mathcal{H}$, that:

$$\|h - h_\omega^\star\|_{\mathcal{H}} \leq \frac{1}{\lambda} \|\partial_h L_{in}(\omega, h)\|_{\mathcal{H}}.$$

This is particularly valid for $h = 0$. Thus,

$$\|h_\omega^\star\|_{\mathcal{H}} \leq \frac{1}{\lambda} \|\partial_h L_{in}(\omega, 0)\|_{\mathcal{H}}.$$

Using the expression of the partial derivative $\partial_h L_{in}$ established in Proposition B.1, we obtain:

$$\|h_\omega^\star\|_{\mathcal{H}} \leq \frac{1}{\lambda} \big\|\mathbb{E}_{\mathbb{P}}\left[\partial_v \ell_{in}(\omega, 0, y) K(x, \cdot)\right]\big\|_{\mathcal{H}}.$$

By Assumption (B), $K$ is bounded by $\kappa$. Hence, Jensen's inequality yields:

$$\|h_\omega^\star\|_{\mathcal{H}} \leq \frac{1}{\lambda} \mathbb{E}_{\mathbb{P}}\Big[|\partial_v \ell_{in}(\omega, 0, y)| \, \|K(x, \cdot)\|_{\mathcal{H}}\Big] \leq \frac{\sqrt{\kappa}}{\lambda} \mathbb{E}_{\mathbb{P}}\Big[|\partial_v \ell_{in}(\omega, 0, y)|\Big].$$

By Assumption (C), $\mathcal{Y}$ is compact, which implies that $\text{hull}(\Omega) \times \mathcal{Y}$ is compact. From Assumption (D), we know that the function $(\omega, y) \mapsto \partial_v \ell_{in}(\omega, 0, y)$ is continuous. Given that every continuous function on a compact space is bounded, we obtain:

$$\|h_\omega^\star\|_{\mathcal{H}} \leq \frac{B\sqrt{\kappa}}{\lambda} < +\infty, \quad \text{where} \quad B := \sup_{\omega \in \text{hull}(\Omega), y \in \mathcal{Y}} |\partial_v \ell_{in}(\omega, 0, y)| > 0.$$

To prove that $\left\|\hat{h}_\omega\right\|_{\mathcal{H}} \leq \frac{B\sqrt{\kappa}}{\lambda}$, we follow a similar approach to that of $\|h_\omega^\star\|_{\mathcal{H}} \leq \frac{B\sqrt{\kappa}}{\lambda}$. More precisely, we investigate the case where the expectation is with respect to the empirical estimate $\widehat{\mathbb{P}}_n$ of $\mathbb{P}$.

$h_\omega^\star(x)$ **and** $\hat{h}_\omega(x)$ **belong to** $\mathcal{V}$**.** Let $\omega \in \text{hull}(\Omega)$ and $x \in \mathcal{X}$. By the reproducing property, the Cauchy-Schwarz inequality, and Assumption (B), we have:

$$|h_\omega^\star(x)| \leq \sqrt{\kappa} \|h_\omega^\star\| \quad \text{and} \quad \left|\hat{h}_\omega(x)\right| \leq \sqrt{\kappa} \left\|\hat{h}_\omega\right\|.$$

Using the bound on $\|h_\omega^\star\|_{\mathcal{H}}$ and $\left\|\hat{h}_\omega\right\|_{\mathcal{H}}$ already proved in the first part of this proof, we get:

$$|h_\omega^\star(x)| \leq \frac{B\kappa}{\lambda} \quad \text{and} \quad \left|\hat{h}_\omega(x)\right| \leq \frac{B\kappa}{\lambda}.$$

This concludes the proof. $\qquad \square$

**Proposition D.2** (Lipschitz continuity of $\omega \mapsto h_\omega^\star$). *Under Assumptions (A) to (E), the function $\omega \mapsto h_\omega^\star$ is $\frac{L\sqrt{\kappa}}{\lambda}$-Lipschitz continuous on $\text{hull}(\Omega)$, where $L := \sup_{\omega \in \text{hull}(\Omega), v \in \mathcal{V}, y \in \mathcal{Y}} \left\|\partial_{\omega,v}^2 \ell_{in}(\omega, v, y)\right\| > 0$, and $\mathcal{V}$ is the compact interval introduced in Proposition D.1.*

*Proof.* To prove this proposition, we adopt the strategy of finding an upper bound for the Jacobian, which serves as the Lipschitz constant.

Let $\omega \in \text{hull}(\Omega)$. Using Propositions B.1 and B.3, we know that $h \mapsto L_{in}(\omega, h)$ is $\lambda$-strongly convex and Fréchet differentiable. Also, by Proposition B.2, $\partial_h L_{in}$ is Fréchet differentiable on $\mathbb{R}^d \times \mathcal{H}$, and,

a fortiori, Hadamard differentiable. Then, by the functional implicit differentiation theorem [36, 58, Theorem 2.1], the Jacobian $\partial_\omega h_\omega^\star : \mathcal{H} \to \mathbb{R}^d$ can be expressed as:

$$\partial_\omega h_\omega^\star = -\partial_{\omega,h}^2 L_{in}(\omega, h_\omega^\star) \left(\partial_h^2 L_{in}(\omega, h_\omega^\star)\right)^{-1}.$$

We have:

$$
\begin{aligned}
\|\partial_\omega h_\omega^\star\|_{\mathrm{op}} &\leq \left\|\partial_{\omega,h}^2 L_{in}(\omega, h_\omega^\star)\right\|_{\mathrm{op}} \left\|\left(\partial_h^2 L_{in}(\omega, h_\omega^\star)\right)^{-1}\right\|_{\mathrm{op}} \\
&\leq \frac{\left\|\partial_{\omega,h}^2 L_{in}(\omega, h_\omega^\star)\right\|_{\mathrm{op}}}{\lambda} \\
&= \frac{\left\|\mathbb{E}_\mathbb{P}\left[\partial_{\omega,v}^2 \ell_{in}(\omega, h_\omega^\star(x), y) K(x, \cdot)\right]\right\|_{\mathrm{op}}}{\lambda} \\
&\leq \frac{\mathbb{E}_\mathbb{P}\left[\left\|\partial_{\omega,v}^2 \ell_{in}(\omega, h_\omega^\star(x), y)\right\| \|K(x, \cdot)\|_\mathcal{H}\right]}{\lambda} \\
&\leq \frac{\sqrt{\kappa}\,\mathbb{E}_\mathbb{P}\left[\left\|\partial_{\omega,v}^2 \ell_{in}(\omega, h_\omega^\star(x), y)\right\|\right]}{\lambda},
\end{aligned}
\tag{13}
$$

where the first line uses the sub-multiplicative property of the operator norm $\|\cdot\|_{\mathrm{op}}$, the second line stems from the fact that $h \mapsto L_{in}(\omega, h)$ is $\lambda$-strongly convex, for any $\omega \in \mathbb{R}^d$, as proved in Proposition B.3, the third line follows from Proposition B.2, the fourth line uses Jensen's inequality, and the last line is a direct consequence of the boundedness of $K$ by $\kappa$ (Assumption (B)). According to Proposition D.1, $h_\omega^\star(x) \in \mathcal{V} := \left[-\frac{B\kappa}{\lambda}, \frac{B\kappa}{\lambda}\right]$, which is a compact interval of $\mathbb{R}$, where $B := \sup_{\omega \in \mathrm{hull}(\Omega), y \in \mathcal{Y}} |\partial_v \ell_{in}(\omega, 0, y)| > 0$. By Assumption (C), $\mathcal{Y}$ is a compact set, hence $\mathrm{hull}(\Omega) \times \mathcal{V} \times \mathcal{Y}$ is compact. Besides, by Assumption (D), $(\omega, v, y) \mapsto \partial_v \ell_{in}(\omega, v, y)$ is continuous over the domain $\mathrm{hull}(\Omega) \times \mathcal{V} \times \mathcal{Y}$. Since every continuous function on a compact set is bounded, this leads to:

$$\mathbb{E}_\mathbb{P}\left[\left\|\partial_{\omega,v}^2 \ell_{in}(\omega, h_\omega^\star(x), y)\right\|\right] \leq L := \sup_{\omega \in \mathrm{hull}(\Omega), v \in \mathcal{V}, y \in \mathcal{Y}} \left\|\partial_{\omega,v}^2 \ell_{in}(\omega, v, y)\right\| < +\infty.$$

Substituting this bound into Equation (13) means that $\frac{L\sqrt{\kappa}}{\lambda}$ is an upper bound on $\|\partial_\omega h_\omega^\star\|_{\mathrm{op}}$. Thus, the result follows as desired. $\qquad\square$

### D.2 Local boundedness and Lipschitz properties of $\ell_{in}$, $\ell_{out}$, and their derivatives

**Proposition D.3** (Local boundedness). *Under Assumptions (A) to (E), the functions $(\omega, x, y) \mapsto \ell_{out}(\omega, h_\omega^\star(x), y)$, $(\omega, x, y) \mapsto \partial_\omega \ell_{out}(\omega, h_\omega^\star(x), y)$, and $(\omega, x, y) \mapsto \partial_v \ell_{out}(\omega, h_\omega^\star(x), y)$ are bounded over $\mathrm{hull}(\Omega) \times \mathcal{X} \times \mathcal{Y}$ by some positive constant $M_{out}$. Similarly, the functions $(\omega, x, y) \mapsto \partial_v \ell_{in}(\omega, h_\omega^\star(x), y)$, $(\omega, x, y) \mapsto \partial_v^2 \ell_{in}(\omega, h_\omega^\star(x), y)$, and $(\omega, x, y) \mapsto \partial_{\omega,v}^2 \ell_{in}(\omega, h_\omega^\star(x), y)$ are bounded over $\mathrm{hull}(\Omega) \times \mathcal{X} \times \mathcal{Y}$ by some positive constant $M_{in}$. The constants $M_{out}$ and $M_{in}$ are defined as:*

$$M_{out} := \sup_{\omega \in \mathrm{hull}(\Omega), v \in \mathcal{V}, y \in \mathcal{Y}} \max\left(|\ell_{out}(\omega, v, y)|, \|\partial_\omega \ell_{out}(\omega, v, y)\|, |\partial_v \ell_{out}(\omega, v, y)|\right) > 0,$$

$$M_{in} := \sup_{\omega \in \mathrm{hull}(\Omega), v \in \mathcal{V}, y \in \mathcal{Y}} \max\left(|\partial_v \ell_{in}(\omega, v, y)|, \left|\partial_v^2 \ell_{in}(\omega, v, y)\right|, \left\|\partial_{\omega,v}^2 \ell_{in}(\omega, v, y)\right\|\right) > 0,$$

*where $\mathcal{V} \subset \mathbb{R}$ is the compact interval defined in Proposition D.1.*

*Proof.* By Proposition D.1, we have that $h_\omega^\star(x) \in \mathcal{V} := \left[-\frac{B\kappa}{\lambda}, \frac{B\kappa}{\lambda}\right] \subset \mathbb{R}$, for any $x \in \mathcal{X}$. From Assumption (D), we know that $\ell_{in}$, $\ell_{out}$, and their partial derivatives are all continuous on $\mathrm{hull}(\Omega) \times \mathcal{V} \times \mathcal{Y}$. Also, $\mathcal{Y}$ is compact by Assumption (C). Thus, $\mathrm{hull}(\Omega) \times \mathcal{V} \times \mathcal{Y}$ is compact. As every continuous function defined over a compact space is bounded, we obtain that:

$$\sup_{\omega \in \mathrm{hull}(\Omega), x \in \mathcal{X}, y \in \mathcal{Y}} |\ell_{out}(\omega, h_\omega^\star(x), y)| \leq \sup_{\omega \in \mathrm{hull}(\Omega), v \in \mathcal{V}, y \in \mathcal{Y}} |\ell_{out}(\omega, v, y)| < +\infty,$$

$$\sup_{\omega \in \mathrm{hull}(\Omega), x \in \mathcal{X}, y \in \mathcal{Y}} \|\partial_\bullet \ell_\circ(\omega, h_\omega^\star(x), y)\| \leq \sup_{\omega \in \mathrm{hull}(\Omega), v \in \mathcal{V}, y \in \mathcal{Y}} \|\partial_\bullet \ell_\circ(\omega, v, y)\| < +\infty,$$

where $\bullet \in \{\{v\}, \{w\}, \{w, v\}\}$ and $\circ \in \{in, out\}$. This implies the desired result. $\qquad\square$

**Proposition D.4** (Local Lipschitz continuity)**.** *Under Assumptions (A) to (E), there exists a positive constant* $\mathrm{Lip}_{out}$ *so that for any* $(x,y)$ *in* $\mathcal{X} \times \mathcal{Y}$, *the functions* $\omega \mapsto \ell_{out}(\omega, h_\omega^\star(x), y)$, $\omega \mapsto \partial_\omega \ell_{out}(\omega, h_\omega^\star(x), y)$, *and* $\omega \mapsto \partial_v \ell_{out}(\omega, h_\omega^\star(x), y)$ *are locally* $\frac{\mathrm{Lip}_{out}}{\lambda}$-*Lipschitz continuous over* $\mathrm{hull}(\Omega)$. *Similarly, there exists a positive constant* $\mathrm{Lip}_{in}$ *so that for any* $(x,y)$ *in* $\mathcal{X} \times \mathcal{Y}$, *the functions* $\omega \mapsto \partial_v \ell_{in}(\omega, h_\omega^\star(x), y)$, $\omega \mapsto \partial_v^2 \ell_{in}(\omega, h_\omega^\star(x), y)$, *and* $\omega \mapsto \partial_{\omega,v}^2 \ell_{in}(\omega, h_\omega^\star(x), y)$ *are locally* $\frac{\mathrm{Lip}_{in}}{\lambda}$-*Lipschitz continuous over* $\mathrm{hull}(\Omega)$. *The constants* $\mathrm{Lip}_{out}$ *and* $\mathrm{Lip}_{in}$ *are defined, for any* $0 < \lambda \le \Lambda$, *as:*

$$\mathrm{Lip}_{out} := (\Lambda + M_{in}\kappa) \max \left( M_{out}, \bar{M}_{out} \right) > 0$$
$$\mathrm{Lip}_{in} := (\Lambda + M_{in}\kappa) \max \left( M_{in}, \bar{M}_{in} \right) > 0,$$

*where:*

$$\bar{M}_{out} := \sup_{\omega \in \mathrm{hull}(\Omega), v \in \mathcal{V}, y \in \mathcal{Y}} \max \left( \left\| \partial_\omega^2 \ell_{out}(\omega, v, y) \right\|_{\mathrm{op}}, \left\| \partial_{\omega,v}^2 \ell_{out}(\omega, v, y) \right\|, \left| \partial_v^2 \ell_{out}(\omega, v, y) \right| \right) > 0,$$

$$\bar{M}_{in} := \sup_{\omega \in \mathrm{hull}(\Omega), v \in \mathcal{V}, y \in \mathcal{Y}} \max \left( \left\| \partial_\omega \partial_v^2 \ell_{in}(\omega, v, y) \right\|, \left| \partial_v^3 \ell_{in}(\omega, v, y) \right|, \left\| \partial_\omega \partial_{\omega,v}^2 \ell_{in}(\omega, v, y) \right\| \right) > 0,$$

*with* $M_{out}$ *and* $M_{in}$ *being the positive constants defined in Proposition D.3, and* $\mathcal{V} \subset \mathbb{R}$ *is the compact interval defined in Proposition D.1.*

*Proof.* For any $(\omega, x, y) \in \mathrm{hull}(\Omega) \times \mathcal{X} \times \mathcal{Y}$, we have:

$$\|\nabla_\omega \ell_{out}(\omega, h_\omega^\star(x), y)\| = \|\partial_\omega \ell_{out}(\omega, h_\omega^\star(x), y) + \partial_v \ell_{out}(\omega, h_\omega^\star(x), y) \partial_\omega h_\omega^\star(x)\|$$
$$\le \|\partial_\omega \ell_{out}(\omega, h_\omega^\star(x), y)\| + |\partial_v \ell_{out}(\omega, h_\omega^\star(x), y)| \, \|\partial_\omega h_\omega^\star\|_{\mathrm{op}} \, \|K(x, \cdot)\|_{\mathcal{H}}$$
$$\le M_{out} \left( 1 + \frac{M_{in}\kappa}{\lambda} \right)$$
$$\le \frac{M_{out} (\Lambda + M_{in}\kappa)}{\lambda},$$

where the first line uses the chain rule, the second line applies the triangle inequality and the reproducing property of the RKHS $\mathcal{H}$, the third line follows from Proposition D.3 to bound the derivatives of $\ell_{out}$, from Proposition D.2, which states that the function $\omega \mapsto h_\omega^\star$ is $\frac{L\sqrt{\kappa}}{\lambda}$-Lipschitz continuous with $L := \sup_{\omega \in \mathrm{hull}(\Omega), v \in \mathcal{V}, y \in \mathcal{Y}} \left\| \partial_{\omega,v}^2 \ell_{in}(\omega, v, y) \right\| < M_{in}$, to bound $\|\partial_\omega h_\omega^\star\|_{\mathrm{op}}$, and from Assumption (B) to bound $\|K(x, \cdot)\|_{\mathcal{H}}$, and the last line is a direct consequence of $0 < \lambda \le \Lambda$. In a similar way, we obtain:

$$\|\nabla_\omega \partial_\omega \ell_{out}(\omega, h_\omega^\star(x), y)\|_{\mathrm{op}} \le \frac{\bar{M}_{out} (\Lambda + M_{in}\kappa)}{\lambda}, \|\nabla_\omega \partial_v \ell_{out}(\omega, h_\omega^\star(x), y)\| \le \frac{\bar{M}_{out} (\Lambda + M_{in}\kappa)}{\lambda},$$

$$\|\nabla_\omega \partial_v \ell_{in}(\omega, h_\omega^\star(x), y)\| \le \frac{M_{in} (\Lambda + M_{in}\kappa)}{\lambda}, \|\nabla_\omega \partial_v^2 \ell_{in}(\omega, h_\omega^\star(x), y)\| \le \frac{\bar{M}_{in} (\Lambda + M_{in}\kappa)}{\lambda},$$

$$\|\nabla_\omega \partial_{\omega,v}^2 \ell_{in}(\omega, h_\omega^\star(x), y)\|_{\mathrm{op}} \le \frac{\bar{M}_{in} (\Lambda + M_{in}\kappa)}{\lambda}.$$

Combining all these bounds concludes the proof. $\qquad\square$

## E  Generalization Properties

As before, let $\Omega$ be an arbitrary compact subset of $\mathbb{R}^d$.

### E.1  Point-wise estimates

We present a point-wise upper bound on the value error $\left| \mathcal{F}(\omega) - \widehat{\mathcal{F}}(\omega) \right|$ and gradient error $\left\| \nabla \mathcal{F}(\omega) - \widehat{\nabla \mathcal{F}}(\omega) \right\|$. To this end, we introduce the following notation for the error between the inner and outer objectives and their empirical approximations evaluated at the optimal inner solution $h_\omega^\star$:

$$\delta_\omega^{out} := \left| L_{out}(\omega, h_\omega^\star) - \widehat{L}_{out}(\omega, h_\omega^\star) \right|, \qquad \delta_\omega^{in} := \left| L_{in}(\omega, h_\omega^\star) - \widehat{L}_{in}(\omega, h_\omega^\star) \right|.$$

By abuse of notation, we introduce the following errors between partial derivatives of $L_{in}$ and $\widehat{L}_{in}$ (resp. $L_{out}$ and $\widehat{L}_{out}$), evaluated at $(\omega, h_\omega^\star)$, *i.e.*,

$$\partial_h \delta_\omega^{out} := \left\| \partial_h L_{out}(\omega, h_\omega^\star) - \partial_h \widehat{L}_{out}(\omega, h_\omega^\star) \right\|_{\mathcal{H}}, \partial_\omega \delta_\omega^{out} := \left\| \partial_\omega L_{out}(\omega, h_\omega^\star) - \partial_\omega \widehat{L}_{out}(\omega, h_\omega^\star) \right\|,$$

$$\partial_h \delta_\omega^{in} := \left\| \partial_h L_{in}(\omega, h_\omega^\star) - \partial_h \widehat{L}_{in}(\omega, h_\omega^\star) \right\|_{\mathcal{H}}, \partial_\omega \delta_\omega^{in} := \left\| \partial_\omega L_{in}(\omega, h_\omega^\star) - \partial_\omega \widehat{L}_{in}(\omega, h_\omega^\star) \right\|,$$

$$\partial_h^2 \delta_\omega^{in} := \left\| \partial_h^2 L_{in}(\omega, h_\omega^\star) - \partial_h^2 \widehat{L}_{in}(\omega, h_\omega^\star) \right\|_{op}, \partial_{\omega,h}^2 \delta_\omega^{in} := \left\| \partial_{\omega,h}^2 L_{in}(\omega, h_\omega^\star) - \partial_{\omega,h}^2 \widehat{L}_{in}(\omega, h_\omega^\star) \right\|_{op}.$$

**Proposition E.1.** *Under Assumptions (A) to (E), the following holds for any $\omega \in \Omega$:*

$$\left\| h_\omega^\star - \hat{h}_\omega \right\|_{\mathcal{H}} \leq \frac{1}{\lambda} \left\| \partial_h \widehat{L}_{in}(\omega, h_\omega^\star) \right\|_{\mathcal{H}} = \frac{1}{\lambda} \partial_h \delta_\omega^{in}.$$

*Proof.* Let $\omega \in \Omega$. The function $h \mapsto \widehat{L}_{in}(\omega, h)$ is $\lambda$-strongly convex and Fréchet differentiable by Propositions B.1 and B.3. Moreover, $\hat{h}_\omega$ is the minimizer of $h \mapsto \widehat{L}_{in}(\omega, h)$ by definition. Therefore, using Lemma H.2, we obtain a control on the distance in $\mathcal{H}$ to the optimum $\hat{h}_\omega$ of $h \mapsto \widehat{L}_{in}(\omega, h)$ in terms of the gradient $\partial_h \widehat{L}_{in}(\omega, h)$:

$$\left\| h - \hat{h}_\omega \right\|_{\mathcal{H}} \leq \frac{1}{\lambda} \left\| \partial_h \widehat{L}_{in}(\omega, h) \right\|_{\mathcal{H}}, \qquad \forall h \in \mathcal{H}.$$

In particular, choosing $h = h_\omega^\star$ yields the inequality. The fact that $\left\| \partial_h \widehat{L}_{in}(\omega, h_\omega^\star) \right\|_{\mathcal{H}} = \partial_h \delta_\omega^{in}$ follows from the optimality of $h_\omega^\star$ which implies that $\partial_h L_{in}(\omega, h_\omega^\star) = 0$. $\qquad\square$

**Proposition E.2.** *Under Assumptions (A) to (E), the following inequalities hold for any $\omega \in \Omega$:*

$$E_\omega^{out} := \left| \widehat{L}_{out}(\omega, h_\omega^\star) - \widehat{L}_{out}(\omega, \hat{h}_\omega) \right| \leq C_{out} \left\| h_\omega^\star - \hat{h}_\omega \right\|_{\mathcal{H}},$$

$$\partial_h E_\omega^{out} := \left\| \partial_h \widehat{L}_{out}(\omega, h_\omega^\star) - \partial_h \widehat{L}_{out}(\omega, \hat{h}_\omega) \right\|_{\mathcal{H}} \leq C_{out} \left\| h_\omega^\star - \hat{h}_\omega \right\|_{\mathcal{H}},$$

$$\partial_\omega E_\omega^{out} := \left\| \partial_\omega \widehat{L}_{out}(\omega, h_\omega^\star) - \partial_\omega \widehat{L}_{out}(\omega, \hat{h}_\omega) \right\| \leq C_{out} \left\| h_\omega^\star - \hat{h}_\omega \right\|_{\mathcal{H}},$$

$$\partial_h^2 E_\omega^{in} := \left\| \partial_h^2 \widehat{L}_{in}(\omega, h_\omega^\star) - \partial_h^2 \widehat{L}_{in}(\omega, \hat{h}_\omega) \right\|_{op} \leq C_{in} \left\| h_\omega^\star - \hat{h}_\omega \right\|_{\mathcal{H}},$$

$$\partial_{\omega,h}^2 E_\omega^{in} := \left\| \partial_{\omega,h}^2 \widehat{L}_{in}(\omega, h_\omega^\star) - \partial_{\omega,h}^2 \widehat{L}_{in}(\omega, \hat{h}_\omega) \right\|_{op} \leq C_{in} \left\| h_\omega^\star - \hat{h}_\omega \right\|_{\mathcal{H}}.$$

*The positive constants $C_{out}$ and $C_{in}$ are defined as:*

$$C_{out} := \max \left( M_{out}\sqrt{\kappa}, \bar{M}_{out}\kappa, \bar{M}_{out}\sqrt{\kappa} \right) > 0,$$

$$C_{in} := \max \left( \bar{M}_{in}\kappa\sqrt{\kappa}, \bar{M}_{in}\kappa, M_{in}\sqrt{d\kappa} \right) > 0,$$

*where $M_{out}$, $\bar{M}_{out}$, and $\bar{M}_{in}$ are the positive constants defined in Propositions D.3 and D.4.*

*Proof.* **Lipschitz continuity of some functions of interest.** Let $\omega \in \Omega$. According to Proposition D.1, both $h_\omega^\star(x)$ and $\hat{h}_\omega(x)$ lie in the compact interval $\mathcal{V} := \left[ -\frac{B\kappa}{\lambda}, \frac{B\kappa}{\lambda} \right] \subset \mathbb{R}$, for any $x \in \mathcal{X}$, where $B := \sup_{\omega \in \text{hull}(\Omega), y \in \mathcal{Y}} |\partial_v \ell_{in}(\omega, 0, y)| > 0$. By Assumption (C), $\mathcal{Y}$ is a compact set. Hence, $\Omega \times \mathcal{V} \times \mathcal{Y}$ is a compact set as well. Furthermore, by Assumption (D), $(\omega, v, y) \mapsto \ell_{in}(\omega, v, y)$, $(\omega, v, y) \mapsto \ell_{out}(\omega, v, y)$, and their derivatives are all continuous over the compact domain $\Omega \times \mathcal{V} \times \mathcal{Y}$. Therefore, these functions and their derivatives are bounded on this domain. In particular, this also holds when $v$ takes the specific values $h_\omega^\star(x)$ or $\hat{h}_\omega(x)$. Let $\bar{v}$ be either $h_\omega^\star(x)$ or $\hat{h}_\omega(x)$, for any

$x \in \mathcal{X}$. For any $\omega \in \Omega$ and $y \in \mathcal{Y}$, we have:

$$|\partial_v \ell_{out}(\omega, \bar{v}, y)| \leq \sup_{\omega \in \Omega, v \in \mathcal{V}, y \in \mathcal{Y}} |\partial_v \ell_{out}(\omega, v, y)| \leq M_{out} < +\infty,$$

$$\left|\partial_v^2 \ell_{out}(\omega, \bar{v}, y)\right| \leq \sup_{\omega \in \Omega, v \in \mathcal{V}, y \in \mathcal{Y}} \left|\partial_v^2 \ell_{out}(\omega, v, y)\right| \leq \bar{M}_{out} < +\infty,$$

$$\left\|\partial_{\omega,v}^2 \ell_{out}(\omega, \bar{v}, y)\right\| \leq \sup_{\omega \in \Omega, v \in \mathcal{V}, y \in \mathcal{Y}} \left\|\partial_{\omega,v}^2 \ell_{out}(\omega, v, y)\right\| \leq \bar{M}_{out} < +\infty,$$

$$\left|\partial_v^3 \ell_{in}(\omega, \bar{v}, y)\right| \leq \sup_{\omega \in \Omega, v \in \mathcal{V}, y \in \mathcal{Y}} \left|\partial_v^3 \ell_{in}(\omega, v, y)\right| \leq \bar{M}_{in} < +\infty,$$

$$\left\|\partial_v \partial_{\omega,v}^2 \ell_{in}(\omega, \bar{v}, y)\right\|_{op} \leq \sup_{\omega \in \Omega, v \in \mathcal{V}, y \in \mathcal{Y}} \left\|\partial_\omega \partial_v^2 \ell_{in}(\omega, v, y)\right\| \leq \bar{M}_{in} < +\infty.$$

This means that $v \in \mathcal{V} \mapsto \ell_{out}(\omega, v, y)$, $v \in \mathcal{V} \mapsto \partial_v \ell_{out}(\omega, v, y)$, $v \in \mathcal{V} \mapsto \partial_\omega \ell_{out}(\omega, v, y)$, $v \in \mathcal{V} \mapsto \partial_v^2 \ell_{in}(\omega, v, y)$, and $v \in \mathcal{V} \mapsto \partial_{\omega,v}^2 \ell_{in}(\omega, v, y)$ are Lipschitz continuous, with Lipschitz constants $M_{out}, \bar{M}_{out}, \bar{M}_{out}, \bar{M}_{in}$, and $\bar{M}_{in}$, respectively, for any $\omega \in \Omega$ and $y \in \mathcal{Y}$.

**Upper bounds.** We have:

$$E_\omega^{out} := \left|\widehat{L}_{out}(\omega, h_\omega^\star) - \widehat{L}_{out}(\omega, \hat{h}_\omega)\right| = \left|\frac{1}{m}\sum_{j=1}^m \ell_{out}(\omega, h_\omega^\star(\tilde{x}_j), \tilde{y}_j) - \frac{1}{m}\sum_{j=1}^m \ell_{out}(\omega, \hat{h}_\omega(\tilde{x}_j), \tilde{y}_j)\right|$$

$$\leq \frac{1}{m}\sum_{j=1}^m \left|\ell_{out}(\omega, h_\omega^\star(\tilde{x}_j), \tilde{y}_j) - \ell_{out}(\omega, \hat{h}_\omega(\tilde{x}_j), \tilde{y}_j)\right|$$

$$\leq \frac{M_{out}}{m}\sum_{j=1}^m \left|h_\omega^\star(\tilde{x}_j) - \hat{h}_\omega(\tilde{x}_j)\right|$$

$$\leq M_{out}\sqrt{\kappa}\left\|h_\omega^\star - \hat{h}_\omega\right\|_{\mathcal{H}},$$

where the first line uses the definition of $(\omega, h) \mapsto \widehat{L}_{out}(\omega, h)$, the second line applies the triangle inequality, the third line leverages the fact that $v \mapsto \ell_{out}(\omega, v, y)$ is $M_{out}$-Lipschitz continuous, for any $\omega \in \Omega$ and $y \in \mathcal{Y}$, and the last line follows from the reproducing property of the RKHS $\mathcal{H}$, Cauchy-Schwarz's inequality, and Assumption (B) to bound $\|K(x, \cdot)\|_{\mathcal{H}}$ by $\sqrt{\kappa}$. Similarly, we obtain:

$$\partial_h E_\omega^{out} \leq \bar{M}_{out}\kappa \left\|h_\omega^\star - \hat{h}_\omega\right\|_{\mathcal{H}}, \quad \partial_\omega E_\omega^{out} \leq \bar{M}_{out}\sqrt{\kappa}\left\|h_\omega^\star - \hat{h}_\omega\right\|_{\mathcal{H}},$$

$$\partial_h^2 E_\omega^{in} \leq \bar{M}_{in}\kappa\sqrt{\kappa}\left\|h_\omega^\star - \hat{h}_\omega\right\|_{\mathcal{H}}, \quad \partial_{\omega,h}^2 E_\omega^{in} \leq \bar{M}_{in}\kappa\left\|h_\omega^\star - \hat{h}_\omega\right\|_{\mathcal{H}}.$$

Combining all the bounds finishes the proof. $\square$

**Proposition E.3.** *Under Assumptions (A) to (E), the following inequalities hold for any $\omega \in \Omega$:*

$$\|\partial_h L_{out}(\omega, h_\omega^\star)\|_{\mathcal{H}} \leq C_{out}, \quad \left\|\partial_{\omega,h}^2 L_{in}(\omega, h_\omega^\star)\right\|_{op} \leq C_{in}, \quad \left\|\partial_{\omega,h}^2 \widehat{L}_{in}(\omega, \hat{h}_\omega)\right\|_{op} \leq C_{in},$$

*where $C_{out}$ and $C_{in}$ are the positive constants defined in Proposition E.2.*

*Proof.* Let $\omega \in \Omega$.

**Upper bound on** $\|\partial_h L_{out}(\omega, h_\omega^\star)\|_{\mathcal{H}}$**.** We have:

$$\|\partial_h L_{out}(\omega, h_\omega^\star)\|_{\mathcal{H}} = \left\|\mathbb{E}_{\mathbb{Q}}\left[\partial_v \ell_{out}(\omega, h_\omega^\star(x), y)K(x, \cdot)\right]\right\|_{\mathcal{H}}$$

$$\leq \mathbb{E}_{\mathbb{Q}}\Big[|\partial_v \ell_{out}(\omega, h_\omega^\star(x), y)| \, \|K(x, \cdot)\|_{\mathcal{H}}\Big]$$

$$\leq \sqrt{\kappa}\mathbb{E}_{\mathbb{Q}}\Big[|\partial_v \ell_{out}(\omega, h_\omega^\star(x), y)|\Big],$$

where the first line follows from Proposition B.1, the second line results from the triangle inequality, and the last line uses Assumption (B) to bound $\|K(x, \cdot)\|_{\mathcal{H}}$ by $\sqrt{\kappa}$. Furthermore, we know by

Proposition D.1 that $(\omega, h_\omega^\star(x), y)$ belongs to the compact subset $\Omega \times \mathcal{V} \times \mathcal{Y}$, and by Proposition D.3 that $\partial_v \ell_{out}(\omega, h_\omega^\star(x), y)$ is bounded by a constant $M_{out}$ on $\mathrm{hull}(\Omega) \times \mathcal{V} \times \mathcal{Y}$. Hence, it follows that:

$$\|\partial_h L_{out}(\omega, h_\omega^\star)\|_\mathcal{H} \leq \sqrt{\kappa} M_{out} \leq C_{out},$$

where $C_{out}$ is defined in Proposition E.2.

**Upper bound on** $\left\|\partial_{\omega,h}^2 L_{in}(\omega, h_\omega^\star)\right\|_{\mathrm{op}}$. According to Proposition B.2, $\partial_{\omega,h}^2 L_{in}(\omega, h_\omega^\star)$ is a Hilbert-Schmidt operator, which points to:

$$\left\|\partial_{\omega,h}^2 L_{in}(\omega, h_\omega^\star)\right\|_{\mathrm{op}} \leq \left\|\partial_{\omega,h}^2 L_{in}(\omega, h_\omega^\star)\right\|_{\mathrm{HS}} = \sqrt{\sum_{l=1}^d \left\|\partial_{\omega_l,h}^2 L_{in}(\omega, h_\omega^\star)\right\|_\mathcal{H}^2}. \tag{14}$$

This means that to find an upper bound on $\left\|\partial_{\omega,h}^2 L_{in}(\omega, h_\omega^\star)\right\|_{\mathrm{op}}$, it suffices to establish an upper bound on $\left\|\partial_{\omega_l,h}^2 L_{in}(\omega, h_\omega^\star)\right\|_\mathcal{H}^2$ for any $l \in \{1, \ldots, d\}$. For a fixed $l \in \{1, \ldots, d\}$, we have:

$$\begin{aligned}
\left\|\partial_{\omega_l,h}^2 L_{in}(\omega, h_\omega^\star)\right\|_\mathcal{H}^2 &= \left\|\mathbb{E}_\mathbb{P}\left[\partial_{\omega_l,v}^2 \ell_{in}(\omega, h_\omega^\star(x), y) K(x, \cdot)\right]\right\|_\mathcal{H}^2 \\
&\leq \mathbb{E}_\mathbb{P}\left[\left|\partial_{\omega_l,v}^2 \ell_{in}(\omega, h_\omega^\star(x), y)\right|^2 \|K(x, \cdot)\|_\mathcal{H}^2\right] \\
&\leq \mathbb{E}_\mathbb{P}\left[\left\|\partial_{\omega,v}^2 \ell_{in}(\omega, h_\omega^\star(x), y)\right\|^2\right] \kappa,
\end{aligned}$$

where the first line follows from Proposition B.2, the second line is a consequence of Jensen's inequality applied on the convex function $\|\cdot\|^2$, and the last line applies Assumption (B) to bound $\|K(x, \cdot)\|_\mathcal{H}^2$ by $\kappa$. Incorporating this upper bound into Equation (14) yields:

$$\left\|\partial_{\omega,h}^2 L_{in}(\omega, h_\omega^\star)\right\|_{\mathrm{op}} \leq \sqrt{\mathbb{E}_\mathbb{P}\left[\left\|\partial_{\omega,v}^2 \ell_{in}(\omega, h_\omega^\star(x), y)\right\|^2\right] d\kappa} \leq M_{in}\sqrt{d\kappa} \leq C_{in},$$

where we used Proposition D.3 to bound $\partial_{\omega,v}^2 \ell_{in}(\omega, h_\omega^\star(x), y)$ by the constant $M_{in}$.

**Upper bound on** $\left\|\partial_{\omega,h}^2 \widehat{L}_{in}(\omega, \hat{h}_\omega)\right\|_{\mathrm{op}}$. The derivation of this upper bound follows the same steps as the previous one, with the only differences being the use of $\widehat{L}_{in}$ instead of $L_{in}$, and $\hat{h}_\omega$ instead of $h_\omega^\star$.

Note that in the last step of each of the three upper bounds, we used the fact that the functions we are dealing with are continuous by Assumption (D) on $\Omega \times \mathcal{V} \times \mathcal{Y}$, which is compact because $\Omega$ is compact, $\mathcal{Y}$ is compact by Assumption (C), and $\mathcal{V}$ is a compact interval of $\mathbb{R}$ defined in Proposition D.1. Hence, those functions are bounded. $\qquad\square$

**Proposition E.4** (Approximation bounds). *Under Assumptions (A) to (E), the following holds for any $\omega \in \Omega$:*

$$\left|\mathcal{F}(\omega) - \widehat{\mathcal{F}}(\omega)\right| \leq \delta_\omega^{out} + \frac{C_{out}}{\lambda} \partial_h \delta_\omega^{in},$$

$$\begin{aligned}
\left\|\nabla\mathcal{F}(\omega) - \widehat{\nabla\mathcal{F}}(\omega)\right\| \leq &\partial_\omega \delta_\omega^{out} + \frac{C_{in}}{\lambda} \partial_h \delta_\omega^{out} + \frac{C_{out}C_{in}}{\lambda^2} \partial_h^2 \delta_\omega^{in} \\
&+ \frac{C_{out}}{\lambda} \partial_{\omega,h}^2 \delta_\omega^{in} + \frac{C_{out}}{\lambda}\left(1 + 2\frac{C_{in}}{\lambda} + \frac{C_{in}^2}{\lambda^2}\right) \partial_h \delta_\omega^{in},
\end{aligned}$$

*where the constants $C_{in}$ and $C_{out}$ are given in Proposition E.2.*

*Proof.* In all what follows, we fix a value for $\omega$ in $\Omega$. We start by controlling the value function, then its gradient.

**Control on the value function.** By the triangle inequality, we have:

$$\left|\mathcal{F}(\omega) - \widehat{\mathcal{F}}(\omega)\right| \leq \underbrace{\left|L_{out}(\omega, h_\omega^\star) - \widehat{L}_{out}(\omega, h_\omega^\star)\right|}_{\delta_\omega^{out}} + \underbrace{\left|\widehat{L}_{out}(\omega, h_\omega^\star) - \widehat{L}_{out}(\omega, \hat{h}_\omega)\right|}_{E_\omega^{out}}. \tag{15}$$

According to Proposition E.2, the error term $E_\omega^{out}$ is controlled by the norm of the difference $h_\omega^\star - \hat{h}_\omega$, i.e., $E_\omega^{out} \leq C_{out} \left\| h_\omega^\star - \hat{h}_\omega \right\|_{\mathcal{H}}$. Moreover, by Proposition E.1, we know that $\left\| h_\omega^\star - \hat{h}_\omega \right\|_{\mathcal{H}} \leq \frac{1}{\lambda} \partial_h \delta_\omega^{in}$. Therefore, combining both bounds yields $E_\omega^{out} \leq \frac{C_{out}}{\lambda} \partial_h \delta_\omega^{in}$. The upper bound on the value function follows by substituting the previous inequality into Equation (15).

**Control on the gradient.** By Proposition B.4, we have the following expression for the total gradient $\nabla \mathcal{F}$:

$$\nabla \mathcal{F}(\omega) = \partial_\omega L_{out}(\omega, h_\omega^\star) - \partial_{\omega,h}^2 L_{in}(\omega, h_\omega^\star) \left( \partial_h^2 L_{in}(\omega, h_\omega^\star) \right)^{-1} \partial_h L_{out}(\omega, h_\omega^\star).$$

Similarly, the gradient estimator $\widehat{\nabla \mathcal{F}}$ is defined by replacing $L_{out}$ and $L_{in}$ by their empirical versions $\widehat{L}_{out}$ and $\widehat{L}_{in}$, and $h_\omega^\star$ by $\hat{h}_\omega := \arg\min_{h \in \mathcal{H}} \widehat{L}_{in}(\omega, h)$ in the above expression, i.e.,

$$\widehat{\nabla \mathcal{F}}(\omega) = \partial_\omega \widehat{L}_{out}(\omega, \hat{h}_\omega) - \partial_{\omega,h}^2 \widehat{L}_{in}(\omega, \hat{h}_\omega) \left( \partial_h^2 \widehat{L}_{in}(\omega, \hat{h}_\omega) \right)^{-1} \partial_h \widehat{L}_{out}(\omega, \hat{h}_\omega).$$

To simplify notations, for any $h \in \mathcal{H}$, we introduce the following operators $R(h), \widehat{R}(h) : \mathcal{H} \to \Omega$:

$$R(h) = \partial_{\omega,h}^2 L_{in}(\omega, h) \left( \partial_h^2 L_{in}(\omega, h) \right)^{-1} \quad \text{and} \quad \widehat{R}(h) = \partial_{\omega,h}^2 \widehat{L}_{in}(\omega, h) \left( \partial_h^2 \widehat{L}_{in}(\omega, h) \right)^{-1}.$$

The difference $\nabla \mathcal{F}(\omega) - \widehat{\nabla \mathcal{F}}(\omega)$ can be decomposed as:

$$\begin{aligned}
& \nabla \mathcal{F}(\omega) - \widehat{\nabla \mathcal{F}}(\omega) \\
&= \left( \partial_\omega L_{out}(\omega, h_\omega^\star) - \partial_\omega \widehat{L}_{out}(\omega, h_\omega^\star) \right) + \left( \partial_\omega \widehat{L}_{out}(\omega, h_\omega^\star) - \partial_\omega \widehat{L}_{out}(\omega, \hat{h}_\omega) \right) \\
&\quad - \widehat{R}(\hat{h}_\omega) \left( \left( \partial_h L_{out}(\omega, h_\omega^\star) - \partial_h \widehat{L}_{out}(\omega, h_\omega^\star) \right) + \left( \partial_h \widehat{L}_{out}(\omega, h_\omega^\star) - \partial_h \widehat{L}_{out}(\omega, \hat{h}_\omega) \right) \right) \\
&\quad - \left( R(h_\omega^\star) - \widehat{R}(\hat{h}_\omega) \right) \partial_h L_{out}(\omega, h_\omega^\star).
\end{aligned}$$

By taking the norm of the above equality and using the triangle inequality, we obtain the following upper bound:

$$\left\| \nabla \mathcal{F}(\omega) - \widehat{\nabla \mathcal{F}}(\omega) \right\|$$

$$\leq \underbrace{\left\| \partial_\omega L_{out}(\omega, h_\omega^\star) - \partial_\omega \widehat{L}_{out}(\omega, h_\omega^\star) \right\|}_{\partial_\omega \delta_\omega^{out}} + \underbrace{\left\| \partial_\omega \widehat{L}_{out}(\omega, h_\omega^\star) - \partial_\omega \widehat{L}_{out}(\omega, \hat{h}_\omega) \right\|}_{\partial_\omega E_\omega^{out}}$$

$$+ \left\| \widehat{R}(\hat{h}_\omega) \right\|_{op} \left( \underbrace{\left\| \partial_h L_{out}(\omega, h_\omega^\star) - \partial_h \widehat{L}_{out}(\omega, h_\omega^\star) \right\|_{\mathcal{H}}}_{\partial_h \delta_\omega^{out}} + \underbrace{\left\| \partial_h \widehat{L}_{out}(\omega, h_\omega^\star) - \partial_h \widehat{L}_{out}(\omega, \hat{h}_\omega) \right\|_{\mathcal{H}}}_{\partial_h E_\omega^{out}} \right)$$

$$+ \left\| R(h_\omega^\star) - \widehat{R}(\hat{h}_\omega) \right\|_{op} \left\| \partial_h L_{out}(\omega, h_\omega^\star) \right\|_{\mathcal{H}}. \tag{16}$$

Next, we provide upper bounds on $\left\| R(h_\omega^\star) - \widehat{R}(\hat{h}_\omega) \right\|_{op}$ and $\left\| \widehat{R}(\hat{h}_\omega) \right\|_{op}$ in terms of derivatives of $L_{in}$ and $\widehat{L}_{in}$.

**Upper bounds on** $\left\| R(h_\omega^\star) - \widehat{R}(\hat{h}_\omega) \right\|_{op}$ **and** $\left\| \widehat{R}(\hat{h}_\omega) \right\|_{op}$. By application of Propositions B.2 and B.3, we deduce that $\partial_{\omega,h}^2 L_{in}(\omega, h_\omega^\star)$, $\partial_h^2 L_{in}(\omega, h_\omega^\star)$, $\partial_{\omega,h}^2 \widehat{L}_{in}(\omega, \hat{h}_\omega)$, and $\partial_h^2 \widehat{L}_{in}(\omega, \hat{h}_\omega)$ are all bounded operators. Moreover, since $L_{in}$ and $\widehat{L}_{in}$ are $\lambda$-strongly convex in their second argument by Proposition B.3, it follows that $\partial_h^2 L_{in}(\omega, h_\omega^\star) \geq \lambda \operatorname{Id}_{\mathcal{H}}$ and $\partial_h^2 \widehat{L}_{in}(\omega, \hat{h}_\omega) \geq \lambda \operatorname{Id}_{\mathcal{H}}$. We can therefore apply Lemma H.1 which yields the following inequalities:

$$\begin{aligned}
\left\| R(h_\omega^\star) - \widehat{R}(\hat{h}_\omega) \right\|_{op} \leq & \frac{1}{\lambda^2} \left\| \partial_{\omega,h}^2 L_{in}(\omega, h_\omega^\star) \right\|_{op} \left\| \partial_h^2 L_{in}(\omega, h_\omega^\star) - \partial_h^2 \widehat{L}_{in}(\omega, \hat{h}_\omega) \right\|_{op} \\
& + \frac{1}{\lambda} \left\| \partial_{\omega,h}^2 L_{in}(\omega, h_\omega^\star) - \partial_{\omega,h}^2 \widehat{L}_{in}(\omega, \hat{h}_\omega) \right\|_{op},
\end{aligned}$$

$$\left\| \widehat{R}(\hat{h}_\omega) \right\|_{op} \leq \frac{1}{\lambda} \left\| \partial_{\omega,h}^2 \widehat{L}_{in}(\omega, \hat{h}_\omega) \right\|_{op}.$$

By applying the triangle inequality to both terms of the first inequality above, we obtain:

$$\left\|R(h_\omega^\star) - \widehat{R}(\hat{h}_\omega)\right\|_{\text{op}}$$

$$\leq \frac{1}{\lambda^2} \left\|\partial_{\omega,h}^2 L_{in}(\omega, h_\omega^\star)\right\|_{\text{op}} \left(\underbrace{\left\|\partial_h^2 L_{in}(\omega, h_\omega^\star) - \partial_h^2 \widehat{L}_{in}(\omega, h_\omega^\star)\right\|_{\text{op}}}_{\partial_h^2 \delta_\omega^{in}} + \underbrace{\left\|\partial_h^2 \widehat{L}_{in}(\omega, h_\omega^\star) - \partial_h^2 \widehat{L}_{in}(\omega, \hat{h}_\omega)\right\|_{\text{op}}}_{\partial_h^2 E_\omega^{in}}\right)$$

$$+ \frac{1}{\lambda} \left(\underbrace{\left\|\partial_{\omega,h}^2 L_{in}(\omega, h_\omega^\star) - \partial_{\omega,h}^2 \widehat{L}_{in}(\omega, h_\omega^\star)\right\|_{\text{op}}}_{\partial_{\omega,h}^2 \delta_\omega^{in}} + \underbrace{\left\|\partial_{\omega,h}^2 \widehat{L}_{in}(\omega, h_\omega^\star) - \partial_{\omega,h}^2 \widehat{L}_{in}(\omega, \hat{h}_\omega)\right\|_{\text{op}}}_{\partial_{\omega,h}^2 E_\omega^{in}}\right).$$

**Final bound.** We can now substitute the above bounds on $\left\|R(h_\omega^\star) - \widehat{R}(\hat{h}_\omega)\right\|_{\text{op}}$ and $\left\|\widehat{R}(\hat{h}_\omega)\right\|_{\text{op}}$ into Equation (16) to obtain the following upper bound on the gradient error:

$$\left\|\nabla\mathcal{F}(\omega) - \widehat{\nabla\mathcal{F}}(\omega)\right\|$$

$$\leq \partial_\omega \delta_\omega^{out} + \partial_\omega E_\omega^{out} + \frac{1}{\lambda}\left\|\partial_{\omega,h}^2 \widehat{L}_{in}(\omega, \hat{h}_\omega)\right\|_{\text{op}} \left(\partial_h \delta_\omega^{out} + \partial_h E_\omega^{out}\right)$$

$$+ \|\partial_h L_{out}(\omega, h_\omega^\star)\|_{\mathcal{H}} \left(\frac{1}{\lambda^2}\left\|\partial_{\omega,h}^2 L_{in}(\omega, h_\omega^\star)\right\|_{\text{op}} \left(\partial_h^2 \delta_\omega^{in} + \partial_h^2 E_\omega^{in}\right) + \frac{1}{\lambda}\left(\partial_{\omega,h}^2 \delta_\omega^{in} + \partial_{\omega,h}^2 E_\omega^{in}\right)\right).$$
$$\tag{17}$$

Furthermore, by Proposition E.3, we have the following upper bounds on the derivatives of $L_{in}$ and $L_{out}$:

$$\|\partial_h L_{out}(\omega, h_\omega^\star)\|_{\mathcal{H}} \leq C_{out}, \quad \left\|\partial_{\omega,h}^2 L_{in}(\omega, h_\omega^\star)\right\|_{\text{op}} \leq C_{in}, \quad \left\|\partial_{\omega,h}^2 \widehat{L}_{in}(\omega, \hat{h}_\omega)\right\|_{\text{op}} \leq C_{in}.$$

Incorporating the above bounds into Equation (17), we further get:

$$\left\|\nabla\mathcal{F}(\omega) - \widehat{\nabla\mathcal{F}}(\omega)\right\| \leq \partial_\omega \delta_\omega^{out} + \partial_\omega E_\omega^{out} + \frac{C_{in}}{\lambda}\left(\partial_h \delta_\omega^{out} + \partial_h E_\omega^{out}\right)$$

$$+ C_{out}\left(\frac{C_{in}}{\lambda^2}\left(\partial_h^2 \delta_\omega^{in} + \partial_h^2 E_\omega^{in}\right) + \frac{1}{\lambda}\left(\partial_{\omega,h}^2 \delta_\omega^{in} + \partial_{\omega,h}^2 E_\omega^{in}\right)\right).$$

By Proposition E.2, we can upper-bound the error terms $\partial_\omega E_\omega^{out}$ and $\partial_h E_\omega^{out}$ by $C_{out}\left\|h_\omega^\star - \hat{h}_\omega\right\|_{\mathcal{H}}$, and $\partial_h^2 E_\omega^{in}$ and $\partial_{\omega,h}^2 E_\omega^{in}$ by $C_{in}\left\|h_\omega^\star - \hat{h}_\omega\right\|_{\mathcal{H}}$. Furthermore, since $\left\|h_\omega^\star - \hat{h}_\omega\right\|_{\mathcal{H}} \leq \frac{1}{\lambda}\partial_h \delta_\omega^{in}$ by Proposition E.1, we can further show that the gradient error satisfies the desired bound:

$$\left\|\nabla\mathcal{F}(\omega) - \widehat{\nabla\mathcal{F}}(\omega)\right\| \leq \partial_\omega \delta_\omega^{out} + \frac{C_{in}}{\lambda}\partial_h \delta_\omega^{out} + C_{out}\left(\frac{C_{in}}{\lambda^2}\partial_h^2 \delta_\omega^{in} + \frac{1}{\lambda}\partial_{\omega,h}^2 \delta_\omega^{in}\right)$$

$$+ \frac{C_{out}}{\lambda}\left(1 + 2\frac{C_{in}}{\lambda} + \frac{C_{in}^2}{\lambda^2}\right)\partial_h \delta_\omega^{in}.$$

$$\square$$

### E.2  Maximal inequalities

**Proposition E.5** (Maximal inequalities for empirical processes)**.** *Let $\Lambda$ be a positive constant. Under Assumptions (A) to (E), the following maximal inequalities hold for any $0 < \lambda \leq \Lambda$:*

$$\mathbb{E}_{\mathbb{Q}}\left[\sup_{\omega\in\Omega} \delta_\omega^{out}\right] \leq \sqrt{\frac{1}{\lambda^2 m}}c(\Omega)\max(M_{out}\text{Lip}_{out}\text{diam}(\Omega), \Lambda M_{out}^2),$$

$$\mathbb{E}_{\mathbb{Q}}\left[\sup_{\omega\in\Omega} \partial_\omega \delta_\omega^{out}\right] \leq \sqrt{\frac{d}{\lambda^2 m}}c(\Omega)\max(M_{out}\,\text{Lip}_{out}\text{diam}(\Omega), \Lambda M_{out}^2),$$

*where $c(\Omega)$ is a positive constant greater than 1 that depends only on $\Omega$ and $d$, while $\text{Lip}_{out}$ and $M_{out}$ are positive constants defined in Propositions D.3 and D.4.*

*Proof.* We will apply the result of Proposition F.3 which provides maximal inequalities for real-valued empirical processes that are uniformly bounded and Lipschitz in their parameter. To this end, consider the parametric families:

$$\mathcal{T}_l^{out} \coloneqq \{\mathcal{X} \times \mathcal{Y} \ni (x,y) \mapsto \partial_{w_l}\ell_{out}(\omega, h_\omega^\star(x), y) \mid \omega \in \Omega\}, \qquad 1 \le l \le d$$

$$\mathcal{T}_0^{out} \coloneqq \{\mathcal{X} \times \mathcal{Y} \ni (x,y) \mapsto \ell_{out}(\omega, h_\omega^\star(x), y) \mid \omega \in \Omega\}.$$

For any $0 \le l \le d$, these real-valued functions are uniformly bounded by a positive constant $M_{out}$, thanks to Proposition D.3. Moreover, by Proposition D.4, the functions $\omega \mapsto \partial_{\omega_l}\ell_{out}(\omega, h_\omega^\star(x), y)$ and $\omega \mapsto \ell_{out}(\omega, h_\omega^\star(x), y)$ are all $\lambda^{-1}\,\mathrm{Lip}_{out}$-Lipschitz for any $(x,y) \in \mathcal{X} \times \mathcal{Y}$. Hence, Proposition F.3 is applicable to each of these families, with $\mathbb{D}$ set to $\mathbb{Q}$ and $\mathcal{Z}$ set to $\mathcal{X} \times \mathcal{Y}$. We treat both $\delta_\omega^{out}$ and $\partial_\omega \delta_\omega^{out}$ separately.

**A maximal inequality for $\delta_\omega^{out}$.** For $l = 0$, we readily apply Proposition F.3 with $p = 1$ to get the following maximal inequality for $\delta_\omega^{out}$:

$$\mathbb{E}_{\mathbb{Q}}\left[\sup_{\omega \in \Omega} \delta_\omega^{out}\right] \coloneqq \mathbb{E}_{\mathbb{Q}}\left[\sup_{\omega \in \Omega} \left| \mathbb{E}_{(x,y)\sim\mathbb{Q}}[\ell_{out}(\omega, h_\omega^\star(x), y)] - \frac{1}{m}\sum_{j=1}^m \ell_{out}(\omega, h_\omega^\star(\tilde{x}_j), \tilde{y}_j)\right|\right]$$

$$\le \sqrt{\frac{1}{\lambda^2 m} c(\Omega) \max(M_{out}\mathrm{Lip}_{out}\,\mathrm{diam}(\Omega), \Lambda M_{out}^2)}.$$

**A maximal inequality for $\partial_\omega \delta_\omega^{out}$.** We now turn to $\partial_\omega \delta_\omega^{out}$, which involves vector-valued processes (as an error between the gradient and its estimate). While the maximal inequalities in Proposition F.3 hold for real-valued processes, we will first obtain maximal inequalities for each component appearing in $\partial_\omega \delta_\omega^{out}$ and then sum these to control $\partial_\omega \delta_\omega^{out}$. To this end, we first use the Cauchy-Schwarz inequality which implies that $\mathbb{E}_{\mathbb{Q}}[\sup_{\omega\in\Omega}\partial_\omega\delta_\omega^{out}] \le \mathbb{E}_{\mathbb{Q}}\left[\sup_{\omega\in\Omega}(\partial_\omega\delta_\omega^{out})^2\right]^{\frac{1}{2}}$. Thus we only need to control $\mathbb{E}_{\mathbb{Q}}\left[\sup_{\omega\in\Omega}(\partial_\omega\delta_\omega^{out})^2\right]$. Simple calculations show that:

$$\mathbb{E}_{\mathbb{Q}}\left[\sup_{\omega\in\Omega}\partial_\omega\delta_\omega^{out}\right]^2$$

$$\le \mathbb{E}_{\mathbb{Q}}\left[\sup_{\omega\in\Omega}(\partial_\omega\delta_\omega^{out})^2\right]$$

$$\le \sum_{l=1}^d \mathbb{E}_{\mathbb{Q}}\left[\sup_{\omega\in\Omega}\left|\mathbb{E}_{(x,y)\sim\mathbb{Q}}[\partial_{w_l}\ell_{out}(\omega, h_\omega^\star(x), y)] - \frac{1}{m}\sum_{j=1}^m \partial_{\omega_l}\ell_{out}(\omega, h_\omega^\star(\tilde{x}_j), \tilde{y}_j)\right|^2\right]$$

$$\le \left(\sqrt{\frac{d}{\lambda^2 m} c(\Omega) \max(M_{out}\,\mathrm{Lip}_{out}\,\mathrm{diam}(\Omega), \Lambda M_{out}^2)}\right)^2,$$

where the last inequality follows by application of Proposition F.3 with $p = 2$ to each term in the right-hand side of the first inequality for $1 \le l \le d$. We get the desired bound on $\mathbb{E}_{\mathbb{Q}}[\sup_{\omega\in\Omega}\partial_\omega\delta_\omega^{out}]$ by taking the square root of the above inequality. $\qquad\square$

**Proposition E.6** (Maximal inequalities for RKHS-valued empirical processes). *Let $\Lambda$ be a positive constant. Under Assumptions (A) to (E), the following maximal inequalities hold for any $0 < \lambda \le \Lambda$:*

$$\mathbb{E}_{\mathbb{Q}}\left[\sup_{\omega\in\Omega}\partial_h\delta_\omega^{out}\right] \le \lambda^{-\frac{1}{4}}m^{-\frac{1}{2}}\left(c(\Omega)\max\left(\widetilde{M}_{out,1}\widetilde{L}_{out,1}\,\mathrm{diam}(\Omega), \Lambda\widetilde{M}_{out,1}^2\right)\right)^{\frac{1}{4}},$$

$$\mathbb{E}_{\mathbb{P}}\left[\sup_{\omega\in\Omega}\partial_h\delta_\omega^{in}\right] \le \lambda^{-\frac{1}{4}}n^{-\frac{1}{2}}\left(c(\Omega)\max\left(\widetilde{M}_{in,1}\widetilde{L}_{in,1}\,\mathrm{diam}(\Omega), \Lambda\widetilde{M}_{in,1}^2\right)\right)^{\frac{1}{4}},$$

$$\mathbb{E}_{\mathbb{P}}\left[\sup_{\omega\in\Omega}\partial_{\omega,h}^2\delta_\omega^{in}\right] \le \lambda^{-\frac{1}{4}}n^{-\frac{1}{2}}d^{\frac{1}{2}}\left(c(\Omega)\max\left(\widetilde{M}_{in,1}\widetilde{L}_{in,1}\,\mathrm{diam}(\Omega), \Lambda\widetilde{M}_{in,1}^2\right)\right)^{\frac{1}{4}},$$

$$\mathbb{E}_{\mathbb{P}}\left[\sup_{\omega\in\Omega}\partial_h^2\delta_\omega^{in}\right] \le \lambda^{-\frac{1}{4}}n^{-\frac{1}{2}}\left(c(\Omega)\max\left(\widetilde{M}_{in,2}^2\widetilde{L}_{in,2}\,\mathrm{diam}(\Omega), \Lambda\widetilde{M}_{in,2}^2\right)\right)^{\frac{1}{4}},$$

*where $c(\Omega)$ is a positive constant greater than 1 that depends only on $\Omega$ and $d$, $\widetilde{L}_{out,1}, \widetilde{L}_{in,1}, \widetilde{L}_{in,2}, \widetilde{M}_{out,1}, \widetilde{M}_{in,1}$, and $\widetilde{M}_{in,2}$ are positive constants defined as:*

$$\widetilde{L}_{out,1} := 2\operatorname{Lip}_{out} M_{out}\kappa, \quad \widetilde{L}_{in,1} := 2\operatorname{Lip}_{in} M_{in}\kappa, \quad \widetilde{L}_{in,2} := 2\operatorname{Lip}_{in} M_{in}\kappa^2,$$

$$\widetilde{M}_{out,1} := M_{out}^2\kappa, \quad \widetilde{M}_{in,1} := M_{in}^2\kappa, \quad \widetilde{M}_{in,2} := M_{in}^2\kappa^2,$$

*and $\operatorname{Lip}_{out}, \operatorname{Lip}_{in}, M_{out}$, and $M_{in}$ are positive constants given in Propositions D.3 and D.4.*

*Proof.* Consider parametric families of real-valued functions indexed by $\Omega$ of the form:

$$\mathcal{T}_{s,a} := \{t_\omega : ((x,y),(x',y')) \mapsto f_s(\omega,x,y)f_s(\omega,x',y')K^a(x,x') \mid \omega \in \Omega\},$$

where $a \in \{1,2\}$, $s$ is an integer satisfying $0 \leq s \leq d+2$, and $f_s(\omega,x,y)$ are real-valued functions given by:

$$f_0 : (\omega,x,y) \mapsto \partial_v\ell_{out}(\omega,h_\omega^\star(x),y), \qquad f_1 : (\omega,x,y) \mapsto \partial_v\ell_{in}(\omega,h_\omega^\star(x),y),$$

$$f_2 : (\omega,x,y) \mapsto \partial_v^2\ell_{in}(\omega,h_\omega^\star(x),y), \qquad f_{2+l} : (\omega,x,y) \mapsto \partial_{\omega_l,v}^2\ell_{in}(\omega,h_\omega^\star(x),y), \quad 1 \leq l \leq d.$$

For any $1 \leq s \leq d+2$, the real-valued functions $f_s$ are uniformly bounded by a positive constant $M_{in}$ thanks to Proposition D.3. Moreover, since the kernel $K$ is bounded by $\kappa$ due to Assumption (B), it follows that all elements $t_\omega$ of $\mathcal{T}_{s,a}$ are uniformly bounded by $\widetilde{M}_{in,a} := M_{in}^2\kappa^a$. Moreover, for $1 \leq s \leq d+2$, the functions $\omega \mapsto f_s(\omega,x,y)$ are $\lambda^{-1}\operatorname{Lip}_{in}$-Lipschitz for any $(x,y) \in \mathcal{X} \times \mathcal{Y}$ by Proposition D.4. Hence, it follows that the map $\omega \mapsto t_\omega((x,y),(x',y'))$ is $\lambda^{-1}\widetilde{L}_{in,a}$-Lipschitz with $\widetilde{L}_{in,a} := 2\operatorname{Lip}_{in} M_{in}\kappa^a$ for any $(x,y)$ and $(x',y')$ in $\mathcal{X} \times \mathcal{Y}$. Similarly, for $s = 0$, we get the same properties, albeit, with different constants, *i.e.*, the family $\mathcal{T}_{0,a}$ is uniformly bounded by a constant $\widetilde{M}_{out,a} := M_{out}^2\kappa^a$ with $M_{out}$ introduced in Proposition D.3, and is $\lambda^{-1}\widetilde{L}_{out,a}$-Lipschitz in its parameter with $\widetilde{L}_{out,a} := 2\operatorname{Lip}_{out} M_{out}\kappa^a$ where $\operatorname{Lip}_{out}$ is given in Proposition D.4. Hence, the maximal inequality in Proposition F.4 is applicable to each of these families with $\mathcal{Z}$ set to $\mathcal{X} \times \mathcal{Y}$, and $\mathbb{D}$ set either to $\mathbb{P}$ for $1 \leq s \leq d+2$, or to $\mathbb{Q}$ for $s = 0$. For conciseness, in all what follows, we will write $z = (x,y)$ and $z_i = (x_i,y_i)$ and $\tilde{z}_j = (\tilde{x}_j,\tilde{y}_j)$ for $1 \leq i \leq n$ and $1 \leq j \leq m$.

**Maximal inequalities for $\partial_h\delta_\omega^{out}$ and $\partial_h\delta_\omega^{in}$.** We control $\partial_h\delta_\omega^{out}$ first as $\partial_h\delta_\omega^{in}$ will be dealt with similarly. Using Cauchy-Schwarz inequality and standard calculus, we have that:

$$\mathbb{E}_\mathbb{Q}\left[\sup_{\omega\in\Omega}\partial_h\delta_\omega^{out}\right]^2$$

$$\leq \mathbb{E}_\mathbb{Q}\left[\sup_{\omega\in\Omega}(\partial_h\delta_\omega^{out})^2\right]$$

$$:= \mathbb{E}_\mathbb{Q}\left[\sup_{\omega\in\Omega}\left\|\mathbb{E}_{(x,y)\sim\mathbb{Q}}\left[\partial_v\ell_{out}(\omega,h_\omega^\star(x),y)K(x,\cdot)\right] - \frac{1}{m}\sum_{j=1}^m\partial_v\ell_{out}(\omega,h_\omega^\star(\tilde{x}_j),\tilde{y}_j)K(\tilde{x}_j,\cdot)\right\|_\mathcal{H}^2\right]$$

$$= \mathbb{E}_\mathbb{Q}\left[\sup_{\omega\in\Omega}\mathbb{E}_{z,z'\sim\mathbb{Q}\otimes\mathbb{Q}}\left[t_\omega(z,z')\right] + \frac{1}{m^2}\sum_{i,j=1}^m t_\omega(z_i,z_j) - \frac{2}{m}\sum_{j=1}^m\mathbb{E}_{z\sim\mathbb{Q}}\left[t_\omega(z,\tilde{z}_j)\right]\right],$$

where $t_\omega(z,z') := \partial_v\ell_{out}(\omega,h_\omega^\star(x),y)\partial_v\ell_{out}(\omega,h_\omega^\star(x'),y')K(x,x') \in \mathcal{T}_{0,1}$. The last term is precisely what Proposition F.4 controls when applying it to the family $\mathcal{T}_{0,1}$ and choosing $\mathbb{D}$ to be $\mathbb{Q}$. Therefore, the following maximal inequality holds by application of Proposition F.4:

$$\mathbb{E}_\mathbb{Q}\left[\sup_{\omega\in\Omega}\partial_h\delta_\omega^{out}\right] \leq \lambda^{-\frac{1}{4}}m^{-\frac{1}{2}}\left(c(\Omega)\max\left(\widetilde{M}_{out,1}\widetilde{L}_{out,1}\operatorname{diam}(\Omega),\Lambda\widetilde{M}_{out,1}^2\right)\right)^{\frac{1}{4}},$$

where $c(\Omega)$ is a positive constant greater than 1 that depends only on $\Omega$ and $d$. We obtain a similar inequality for $\partial_h\delta_\omega^{in}$ by carrying out similar calculations, then applying Proposition F.4 to the family $\mathcal{T}_{1,1}$ and choosing $\mathbb{P}$ for the probability distribution $\mathbb{D}$. The resulting bound is then of the form:

$$\mathbb{E}_\mathbb{P}\left[\sup_{\omega\in\Omega}\partial_h\delta_\omega^{in}\right] \leq \lambda^{-\frac{1}{4}}n^{-\frac{1}{2}}\left(c(\Omega)\max\left(\widetilde{M}_{in,1}\widetilde{L}_{in,1}\operatorname{diam}(\Omega),\Lambda\widetilde{M}_{in,1}^2\right)\right)^{\frac{1}{4}}.$$

**A maximal inequality for $\partial^2_{\omega,h}\delta^{in}_\omega$.** We have:

$$\mathbb{E}_\mathbb{P}\left[\sup_{\omega\in\Omega}\partial^2_{\omega,h}\delta^{in}_\omega\right]^2$$

$$\overset{(a)}{\le}\mathbb{E}_\mathbb{P}\left[\sup_{\omega\in\Omega}(\partial^2_{\omega,h}\delta^{in}_\omega)^2\right]$$

$$\overset{(b)}{:=}\mathbb{E}_\mathbb{P}\left[\sup_{\omega\in\Omega}\left\|\mathbb{E}_{(x,y)\sim\mathbb{P}}\left[\partial^2_{\omega,v}\ell_{in}(\omega,h^\star_\omega(x),y)K(x,\cdot)\right]-\frac{1}{n}\sum_{i=1}^n\partial^2_{\omega,v}\ell_{in}(\omega,h^\star_\omega(x_i),y_i)K(x_i,\cdot)\right\|^2_{\mathrm{op}}\right]$$

$$\overset{(c)}{\le}\mathbb{E}_\mathbb{P}\left[\sup_{\omega\in\Omega}\left\|\mathbb{E}_{(x,y)\sim\mathbb{P}}\left[\partial^2_{\omega,v}\ell_{in}(\omega,h^\star_\omega(x),y)K(x,\cdot)\right]-\frac{1}{n}\sum_{i=1}^n\partial^2_{\omega,v}\ell_{in}(\omega,h^\star_\omega(x_i),y_i)K(x_i,\cdot)\right\|^2_{\mathrm{HS}}\right]$$

$$\overset{(d)}{=}\sum_{l=1}^d\mathbb{E}_\mathbb{P}\left[\sup_{\omega\in\Omega}\left\|\mathbb{E}_{(x,y)\sim\mathbb{P}}\left[\partial^2_{\omega_l,v}\ell_{in}(\omega,h^\star_\omega(x),y)K(x,\cdot)\right]-\frac{1}{n}\sum_{i=1}^n\partial^2_{\omega_l,v}\ell_{in}(\omega,h^\star_\omega(x_i),y_i)K(x_i,\cdot)\right\|^2_\mathcal{H}\right]$$

$$\overset{(e)}{=}\sum_{l=1}^d\mathbb{E}_\mathbb{P}\left[\sup_{\omega\in\Omega}\mathbb{E}_{z,z'\sim\mathbb{P}\otimes\mathbb{P}}\left[t_{\omega,l}(z,z')\right]+\frac{1}{n^2}\sum_{i,j=1}^n t_{\omega,l}(z_i,z_j)-\frac{2}{n}\sum_{i=1}^n\mathbb{E}_{z\sim\mathbb{P}}\left[t_{\omega,l}(z,z_i)\right]\right],$$

where we introduced $t_{\omega,l}(z,z'):=\partial^2_{\omega_l,v}\ell_{in}(\omega,h^\star_\omega(x),y)\partial^2_{\omega_l,v}\ell_{in}(\omega,h^\star_\omega(x'),y')K(x,x')\in\mathcal{T}_{2+l,1}$.
Here, (a) follows from the Cauchy-Schwarz inequality, (b) is obtained by definition of $\partial^2_{\omega,h}\delta^{in}_\omega$, while
(c) uses the general fact that the operator norm of an operator is upper-bounded by its Hilbert-Schmidt
norm which is finite in our case by application of Proposition B.2. Moreover, (d) further uses the
Hilbert-Schmidt norm of an operator in terms of the norm of its rows, while (e) simply expands the
squared RKHS norm and uses the reproducing property in the RKHS $\mathcal{H}$. Each term in the last item
(e) is precisely what Proposition F.4 controls when applying it to the families $\mathcal{T}_{2+l,1}$ for $1\le l\le d$
and choosing $\mathbb{D}$ to be $\mathbb{P}$. Therefore, the following maximal inequality holds by a direct application of
Proposition F.4:

$$\mathbb{E}_\mathbb{P}\left[\sup_{\omega\in\Omega}\partial^2_{\omega,h}\delta^{in}_\omega\right]\le\lambda^{-\frac{1}{4}}n^{-\frac{1}{2}}d^{\frac{1}{2}}\left(c(\Omega)\max\left(\widetilde{M}_{in,1}\widetilde{L}_{in,1}\operatorname{diam}(\Omega),\Lambda\widetilde{M}^2_{in,1}\right)\right)^{\frac{1}{4}},$$

where $c(\Omega)$ is a positive constant greater than 1 that depends only on $\Omega$ and $d$.

**A maximal inequality for $\partial^2_h\delta^{in}_\omega$.** We will use a similar approach as for $\partial^2_{\omega,h}\delta^{in}_\omega$. We have:

$$\mathbb{E}_\mathbb{P}\left[\sup_{\omega\in\Omega}\partial^2_h\delta^{in}_\omega\right]^2$$

$$\overset{(a)}{\le}\mathbb{E}_\mathbb{P}\left[\sup_{\omega\in\Omega}(\partial^2_h\delta^{in}_\omega)^2\right]$$

$$\overset{(b)}{:=}\mathbb{E}_\mathbb{P}\left[\sup_{\omega\in\Omega}\left\|\mathbb{E}_{(x,y)\sim\mathbb{P}}\left[\partial^2_v\ell_{in}(\omega,h^\star_\omega(x),y)K(x,\cdot)\otimes K(x,\cdot)\right]\right.\right.$$
$$\left.\left.-\frac{1}{n}\sum_{i=1}^n\partial^2_v\ell_{in}(\omega,h^\star_\omega(x_i),y_i)K(x_i,\cdot)\otimes K(x_i,\cdot)\right\|^2_{\mathrm{op}}\right]$$

$$\overset{(c)}{\le}\mathbb{E}_\mathbb{P}\left[\sup_{\omega\in\Omega}\left\|\mathbb{E}_{(x,y)\sim\mathbb{P}}\left[\partial^2_v\ell_{in}(\omega,h^\star_\omega(x),y)K(x,\cdot)\otimes K(x,\cdot)\right]\right.\right.$$
$$\left.\left.-\frac{1}{n}\sum_{i=1}^n\partial^2_v\ell_{in}(\omega,h^\star_\omega(x_i),y_i)K(x_i,\cdot)\otimes K(x_i,\cdot)\right\|^2_{\mathrm{HS}}\right]$$

$$\overset{(d)}{=}\mathbb{E}_\mathbb{P}\left[\sup_{\omega\in\Omega}\mathbb{E}_{z,z'\sim\mathbb{P}\otimes\mathbb{P}}\left[t_\omega(z,z')\right]+\frac{1}{n^2}\sum_{i,j=1}^n t_\omega(z_i,z_j)-\frac{2}{n}\sum_{i=1}^n\mathbb{E}_{z\sim\mathbb{P}}\left[t_\omega(z,z_i)\right]\right],$$

where we introduced $t_\omega(z, z') \coloneqq \partial_v^2 \ell_{in}(\omega, x, y)\partial_v^2 \ell_{in}(\omega, x', y')K^2(x, x') \in \mathcal{T}_{2,2}$. Here, (a) follows from the Cauchy-Schwarz inequality, (b) is obtained by definition of $\partial_h^2 \delta_\omega^{in}$, while (c) uses the general fact that the operator norm of an operator is upper-bounded by its Hilbert-Schmidt norm which is finite in our case by application of Proposition B.2. Moreover, (d) further uses the identity in Lemma H.3 for computing the Hilbert-Schmidt norm of sum/expectation of tensor-product operators. The last item (d) is precisely what Proposition F.4 controls when applying it to the family $\mathcal{T}_{2,2}$ and choosing $\mathbb{D}$ to be $\mathbb{P}$. Therefore, the following maximal inequality holds by direct application of Proposition F.4:

$$\mathbb{E}_{\mathbb{P}}\left[\sup_{\omega \in \Omega} \partial_h^2 \delta_\omega^{in}\right] \leq \lambda^{-\frac{1}{4}} n^{-\frac{1}{2}} \left(c(\Omega) \max\left(\widetilde{M}_{in,2}\widetilde{L}_{in,2} \operatorname{diam}(\Omega), \Lambda \widetilde{M}_{in,2}^2\right)\right)^{\frac{1}{4}},$$

where $c(\Omega)$ is a positive constant greater than 1 that depends only on $\Omega$ and $d$. $\qquad\square$

### E.3 Proof of Theorem 4.1

**Theorem E.7** (Generalization bounds). *The following holds under Assumptions (A) to (E):*

$$\mathbb{E}\left[\sup_{\omega \in \Omega}\left|\mathcal{F}(\omega) - \widehat{\mathcal{F}}(\omega)\right|\right] \lesssim \frac{1}{\lambda m^{\frac{1}{2}}} + \frac{C_{out}}{\lambda^{\frac{5}{4}} n^{\frac{1}{2}}},$$

$$\mathbb{E}\left[\sup_{\omega \in \Omega}\left\|\nabla \mathcal{F}(\omega) - \widehat{\nabla \mathcal{F}}(\omega)\right\|\right] \lesssim \frac{1}{\lambda}\left(d^{\frac{1}{2}} + \frac{C_{in}}{\lambda^{\frac{1}{4}}}\right)\frac{1}{m^{\frac{1}{2}}} + \frac{C_{out}}{\lambda^{\frac{5}{4}}}\left(2 + 3\frac{C_{in}}{\lambda} + \frac{C_{in}^2}{\lambda^2}\right)\frac{1}{n^{\frac{1}{2}}},$$

*where the constants $C_{in}$ and $C_{out}$ are given in Proposition E.2.*

*Proof.* Using the point-wise estimates in Proposition E.4 and taking their supremum over $\Omega$ followed by the expectations over data, the following error bounds hold:

$$\mathbb{E}\left[\sup_{\omega \in \Omega}\left|\mathcal{F}(\omega) - \widehat{\mathcal{F}}(\omega)\right|\right] \leq \mathbb{E}_{\mathbb{Q}}\left[\sup_{\omega \in \Omega} \delta_\omega^{out}\right] + \frac{C_{out}}{\lambda}\mathbb{E}_{\mathbb{P}}\left[\sup_{\omega \in \Omega} \partial_h \delta_\omega^{in}\right],$$

$$\mathbb{E}\left[\sup_{\omega \in \Omega}\left\|\nabla \mathcal{F}(\omega) - \widehat{\nabla \mathcal{F}}(\omega)\right\|\right] \leq \mathbb{E}_{\mathbb{Q}}\left[\sup_{\omega \in \Omega} \partial_\omega \delta_\omega^{out}\right] + \frac{C_{in}}{\lambda}\mathbb{E}_{\mathbb{Q}}\left[\sup_{\omega \in \Omega} \partial_h \delta_\omega^{out}\right]$$

$$+ \frac{C_{out}}{\lambda}\left(1 + 2\frac{C_{in}}{\lambda} + \frac{C_{in}^2}{\lambda^2}\right)\mathbb{E}_{\mathbb{P}}\left[\sup_{\omega \in \Omega} \partial_h \delta_\omega^{in}\right]$$

$$+ \frac{C_{out}C_{in}}{\lambda^2}\mathbb{E}_{\mathbb{P}}\left[\sup_{\omega \in \Omega} \partial_h^2 \delta_\omega^{in}\right] + \frac{C_{out}}{\lambda}\mathbb{E}_{\mathbb{P}}\left[\sup_{\omega \in \Omega} \partial_{\omega,h}^2 \delta_\omega^{in}\right].$$

Furthermore, we can use the maximal inequalities in Propositions E.5 and E.6 to control each term appearing in the right-hand side of the above inequalities:

$$\mathbb{E}\left[\sup_{\omega \in \Omega}\left|\mathcal{F}(\omega) - \widehat{\mathcal{F}}(\omega)\right|\right] \leq R\left(m^{-\frac{1}{2}}\lambda^{-1} + C_{out}n^{-\frac{1}{2}}\lambda^{-(1+\frac{1}{4})}\right),$$

$$\mathbb{E}\left[\sup_{\omega \in \Omega}\left\|\nabla \mathcal{F}(\omega) - \widehat{\nabla \mathcal{F}}(\omega)\right\|\right] \leq R\Big(m^{-\frac{1}{2}}\lambda^{-1}d^{\frac{1}{2}} + C_{in}m^{-\frac{1}{2}}\lambda^{-(1+\frac{1}{4})}$$

$$+ C_{out}n^{-\frac{1}{2}}\lambda^{-(1+\frac{1}{4})}\left(1 + 2\frac{C_{in}}{\lambda} + \frac{C_{in}^2}{\lambda^2}\right)$$

$$+ C_{out}C_{in}n^{-\frac{1}{2}}\lambda^{-(2+\frac{1}{4})} + C_{out}n^{-\frac{1}{2}}\lambda^{-(1+\frac{1}{4})}\Big),$$

where the constant $R$ depends only on the Lipschitz constants $\operatorname{Lip}_{in}$ and $\operatorname{Lip}_{out}$, the upper bounds $M_{in}$ and $M_{out}$, the bound $\kappa$ on the kernel, the set $\Omega$, and the dimension $d$. Rearranging the obtained upper bounds concludes the proof. $\qquad\square$

### E.4 Generalization for bilevel gradient methods

*Proof of Corollary 4.2.* Consider that $\inf_{\omega,v,y} \ell_{out}(\omega, v, y) - c\|\omega\|^2 \geq 0$, which entails $\ell_{out}(\omega, v, y) \geq c\|\omega\|^2$ for all $v, y$. Using Proposition D.1 and setting $B = \sup_{y \in \mathcal{Y}} |\partial_v \ell_{in}(\omega_0, 0, y)|$,

we have almost surely that:

$$\widehat{\mathcal{F}}(\omega_0) \leq \max_{|v| \leq \frac{B_\kappa}{\lambda}, y \in \mathcal{Y}} \ell_{out}(\omega_0, v, y) =: \bar{\ell}.$$

Therefore, for any $\omega$ such that $\widehat{\mathcal{F}}(\omega) \leq \widehat{\mathcal{F}}(\omega_0)$, we have $\|\omega\|^2 \leq \bar{\ell}/c$. Define $\Omega$ as the ball of radius $\sqrt{\bar{\ell}/c}$ centered at 0. Using the fact that $\widehat{\nabla \mathcal{F}} = \nabla \widehat{\mathcal{F}}$ in Proposition 3.1 and the representation in Equation (4), it is clear from Proposition D.2 and Assumption (**D**) that $\nabla \widehat{\mathcal{F}}$ is Lipschitz on $\Omega$ with a deterministic constant $L$. It follows from standard results on gradient descent for nonconvex $\widehat{\mathcal{F}}$ with Lipschitz gradient (see, *e.g.*, [9, Theorems 4.25, 4.26]) that if we take $\bar{\eta} = 1/L$, then almost surely:

- $\widehat{\mathcal{F}}(\omega_t) \leq \widehat{\mathcal{F}}(\omega_0)$ and $\omega_t \in \Omega$ for all $t \geq 0$.

- $\nabla \widehat{\mathcal{F}}(\omega_t) \to 0$ as $t \to \infty$.

- $\min_{i=0,\ldots,t} \left\| \nabla \widehat{\mathcal{F}}(\omega_i) \right\| \leq \bar{c}/\sqrt{t+1}$ for all $t \geq 0$, where $\bar{c}$ is a deterministic constant.

The corollary then follows by combining Proposition 3.1 and the uniform bound in Theorem 4.1. $\quad\square$

**Bilevel projected gradient descent.** Considering the constrained (KBO) problem and assuming that $\mathcal{C}$ is convex and compact, the projected gradient descent initialized at $\omega_0 \in \mathcal{C}$ iterates the following recursion $\omega_{t+1} = \Pi_\mathcal{C}(\omega_t - \eta \nabla \widehat{\mathcal{F}}(\omega_t))$ for all $t \geq 0$, where $\Pi_\mathcal{C}$ denotes the orthogonal projection onto $\mathcal{C}$ and $\eta > 0$ is the step size. The algorithmic requirements are the same as the gradient descent algorithm, with the additional cost of computing the projection, which is typically cheap for basic sets such as balls. In the constrained setting, the optimality condition should take the constraints into account. To this end, we consider the *gradient mappings* $\widehat{G}_\eta \colon \omega \mapsto \frac{1}{\eta}(\omega - \Pi_\mathcal{C}(\omega - \eta \nabla \widehat{\mathcal{F}}(\omega)))$ and $G_\eta \colon \omega \mapsto \frac{1}{\eta}(\omega - \Pi_\mathcal{C}(\omega - \eta \nabla \mathcal{F}(\omega)))$ [10, Section 10.3]. This captures the stationarity of the recursion, and any local minimum of $\mathcal{F}$ on $\mathcal{C}$ satisfies $G_\eta = 0$ for all $\eta > 0$.

**Corollary E.8** (Generalization for bilevel projected gradient descent). *Consider Assumptions (**A**) to (**E**) and fix $\lambda > 0$. Assume further that $\mathbf{K}$ in (KBO) is almost surely definite, and that $\mathcal{C}$ is convex and compact. Fix $\omega_0 \in \mathcal{C}$ and let $\omega_{t+1} = \Pi_\mathcal{C}(\omega_t - \eta \nabla \widehat{\mathcal{F}}(\omega_t))$, where $\eta > 0$ is the step size and $t \geq 0$ is the iteration index. Then, there exist constants $\bar{\eta} > 0$ and $\bar{c} > 0$ such that for any $0 < \eta < \bar{\eta}$ and $t > 0$, the following holds:*

$$\mathbb{E}\left[ \min_{i=0,\ldots,t} \|G_\eta(\omega_i)\| \right] \leq \bar{c} \left( \frac{1}{\sqrt{m}} + \frac{1}{\sqrt{n}} + \frac{1}{\sqrt{t+1}} \right), \mathbb{E}\left[ \limsup_{i \to \infty} \|G_\eta(\omega_i)\| \right] \leq \bar{c} \left( \frac{1}{\sqrt{m}} + \frac{1}{\sqrt{n}} \right).$$

*Proof.* We choose $\Omega = \mathcal{C}$. All iterates obviously remain in $\Omega$. Similarly as in the proof of Corollary 4.2, we know that $\nabla \widehat{\mathcal{F}}$ is Lipschitz and that $\mathcal{F}$ is bounded on $\Omega$ with deterministic constants. It then follows from classical analysis on the nonconvex projected gradient algorithm (see, *e.g.*, [10, Theorem 10.15]), that for sufficiently small $\eta$, we have almost surely that $\min_{i=0,\ldots,t} \left\| \widehat{G}_\eta(\omega_i) \right\| \leq \bar{c}/\sqrt{t+1}$ for a deterministic constant $\bar{c} > 0$, and that $\widehat{G}_\eta(\omega_i) \to 0$ as $i \to \infty$. Using the fact that the orthogonal projection is 1-Lipschitz, (see, *e.g.*, [10, Theorem 6.42]), we also have for all $\omega \in \Omega$:

$$\|\widehat{G}_\eta(\omega) - G_\eta(\omega)\| \leq \|\nabla \widehat{\mathcal{F}}(\omega) - \nabla \mathcal{F}(\omega)\|.$$

The result follows by combining Proposition 3.1 and the uniform bound in Theorem 4.1. $\quad\square$

# F   Maximal Inequalities for Bounded and Lipschitz Family of Functions

Let $\mathcal{Z}$ be a subset of a Euclidean space and $\Omega$ be a compact subset of $\mathbb{R}^d$. Denote by $\otimes^k \mathcal{Z}$ the $k$-th tensor power of $\mathcal{Z}$, for any $k \geq 1$. Consider a parametric family $\mathcal{T}$ of real-valued functions defined over $\mathcal{Z}$ and indexed by a parameter $\omega \in \Omega$, *i.e.*,

$$\mathcal{T} := \{ \mathcal{Z} \ni z \mapsto t_\omega(z) \in \mathbb{R} \mid \omega \in \Omega \} . \tag{18}$$

For a given probability measure $\mu$ on $\mathcal{Z}$, denote by $L_2(\mu)$ the space of square $\mu$-integrable real-valued functions. We denote by $\|f\|_{\mathbb{D},2} := \mathbb{E}_{\mathbb{D}}\left[f(z)^2\right]^{\frac{1}{2}}$ the $L_2(\mu)$-norm of any function $f \in L_2(\mu)$. For any $\epsilon > 0$, we denote by $D\left(\epsilon, \mathcal{T}, L_2(\mu)\right)$ the $\epsilon$-packing number of $\mathcal{T}$ w.r.t. $L_2(\mu)$. The next proposition provides a control on such a number under regularity conditions on the family $\mathcal{T}$.

**Proposition F.1** (Control on the packing number). *Assume that $\Omega$ is a compact subset of $\mathbb{R}^d$, that the parametric family $\mathcal{T}$ defined in Equation (18) is uniformly bounded by a positive constant $M$, and that there exists a positive constant $L$ so that, for any probability measure $\mu$ on $\mathcal{Z}$, $\omega \mapsto t_\omega(z)$ is $L$-Lipschitz for any $z \in \mathcal{Z}$. Then, there exists a positive constant $c(\Omega)$ greater than $1$ that depends only on $\Omega$ and $d$ so that, for any probability measure $\mu$ on $\mathcal{Z}$, the following bound holds for any $0 < \epsilon \le M$:*

$$D\left(\epsilon, \mathcal{T}, L_2(\mu)\right) \le c(\Omega) \left(\frac{\max\left(L\operatorname{diam}(\Omega), M\right)}{\epsilon}\right)^d.$$

*Proof.* First using [40, Lemma 9.18] and [40, Paragraph 8.1.2], we know that the $\epsilon$-packing number $D\left(\epsilon, \mathcal{T}, L_2(\mu)\right)$ is smaller than the $\frac{\epsilon}{2}$-bracketing number $N_{[]}\left(\frac{\epsilon}{2}, \mathcal{T}, L_2(\mu)\right)$. Hence, we only need to control the bracketing number. To this end, we recall that the function $\omega \mapsto t_\omega(z)$ is $L$-Lipschitz for any $z \in \mathcal{Z}$, so that [71, Example 19.7] ensures the existence of a positive constant $c(\Omega)$ that depends only on $\Omega$ for which the following inequality holds for any $0 < \epsilon < L\operatorname{diam}(\Omega)$:

$$1 \le N_{[]}\left(\epsilon, \mathcal{T}, L_2(\mu)\right) \le c(\Omega) \left(\frac{L\operatorname{diam}(\Omega)}{\epsilon}\right)^d.$$

Moreover, since the $\epsilon$-bracketing number is decreasing in $\epsilon$, it holds that:

$$N_{[]}\left(\epsilon, \mathcal{T}, L_2(\mu)\right) \le N_{[]}\left(\epsilon_-, \mathcal{T}, L_2(\mu)\right) \le c(\Omega) \left(\frac{L\operatorname{diam}(\Omega)}{\epsilon_-}\right)^d,$$

for any $\epsilon \ge L\operatorname{diam}(\Omega)$ and $\epsilon_- \le L\operatorname{diam}(\Omega)$. Taking the limit when $\epsilon_-$ approaches $L\operatorname{diam}(\Omega)$ yields $N_{[]}\left(\epsilon, \mathcal{T}, L_2(\mu)\right) \le c(\Omega)$ for any $\epsilon \ge L\operatorname{diam}(\Omega)$. Hence, we have shown so far that for any $\epsilon > 0$:

$$N_{[]}\left(\epsilon, \mathcal{T}, L_2(\mu)\right) \le c(\Omega) \max\left(1, \left(\frac{L\operatorname{diam}(\Omega)}{\epsilon}\right)^d\right).$$

Moreover, by noticing that $\max(1, \frac{L\operatorname{diam}(\Omega)}{\epsilon}) \le \frac{\max(M, L\operatorname{diam}(\Omega))}{\epsilon}$ for any $\epsilon \le M$, we further have that:

$$N_{[]}\left(\epsilon, \mathcal{T}, L_2(\mu)\right) \le c(\Omega) \left(\frac{\max\left(M, L\operatorname{diam}(\Omega)\right)}{\epsilon}\right)^d.$$

Finally, recalling that $D\left(\epsilon, \mathcal{T}, L_2(\mu)\right) \le N_{[]}\left(\frac{\epsilon}{2}, \mathcal{T}, L_2(\mu)\right)$, we get that $D\left(\epsilon, \mathcal{T}, L_2(\mu)\right) \le 2^d c(\Omega) \left(\frac{\max(M, L\operatorname{diam}(\Omega))}{\epsilon}\right)^d$. The desired bound follows after redefining $c(\Omega)$ to include the factor $2^d$ (i.e., $c(\Omega) \to 2^d c(\Omega)$). $\qquad\square$

**Theorem F.2** (Maximal inequality for degenerate, bounded, and Lipschitz $U$-processes). *Let $k$ be either $1$ or $2$. Consider a parametric family $\mathcal{T} := \left\{\otimes^k \mathcal{Z} \ni (z_1, \ldots, z_k) \mapsto t_\omega(z_1, \ldots, z_k) \in \mathbb{R} \mid \omega \in \Omega\right\}$ of real-valued functions over $\otimes^k \mathcal{Z}$ indexed by a parameter $\omega \in \Omega$, where $\Omega$ is a compact subset of $\mathbb{R}^d$. For a given probability distribution $\mathbb{D}$ over $\mathcal{Z}$, assume that all elements $t_\omega$ are degenerate w.r.t. $\mathbb{D}$, meaning that:*

$$\begin{cases} \mathbb{E}_{\bar{z}\sim\mathbb{D}}\left[t_\omega(\bar{z})\right] = 0, & \text{if } k = 1 \\ \mathbb{E}_{\bar{z}\sim\mathbb{D}}\left[t_\omega(z, \bar{z})\right] = \mathbb{E}_{\bar{z}\sim\mathbb{D}}\left[t_\omega(\bar{z}, z)\right] = 0, \quad \forall z \in \mathcal{Z}, & \text{if } k = 2. \end{cases}$$

*Furthermore, assume that all functions in $\mathcal{T}$ are uniformly bounded by a positive constant $M$ and that there exists a positive constant $L$ so that $\omega \mapsto t_\omega(z_1, \ldots, z_k)$ is $L$-Lipschitz for any $(z_1, \ldots, z_k) \in \otimes^k \mathcal{Z}$. Given i.i.d. samples $(z_i)_{1 \le i \le n}$ from $\mathbb{D}$, consider the following $U$-statistic $U_n^k$:*

$$U_n^k t_\omega := \begin{cases} \dfrac{1}{n} \displaystyle\sum_{i=1}^n t_\omega(z_i), & \text{if } k = 1 \\ \dfrac{1}{n(n-1)} \displaystyle\sum_{\substack{i,j=1 \\ i \ne j}}^n t_\omega(z_i, z_j), & \text{if } k = 2. \end{cases}$$

*Then, there exists a universal positive constant $c(\Omega)$ greater than $1$ that depends only on $\Omega$ and $d$ such that for any $p \in \{1, 2\}$:*

$$\mathbb{E}_{\mathbb{D}}\left[\sup_{\omega \in \Omega}\left|U_n^k t_\omega\right|^p\right]^{\frac{1}{p}} \leq n^{-\frac{k}{2}} c(\Omega) \max\left(ML\operatorname{diam}(\Omega), M^2\right)^{\frac{1}{2}}.$$

*Proof.* **Maximal inequality for degenerate $U$-processes.** We will first apply the general result in [67, Maximal inequality] which controls $\mathbb{E}_{\mathbb{D}}\left[\sup_{\omega \in \Omega}\left|U_n^k t_\omega\right|\right]$ in terms of the packing number of $\mathcal{T}$. First note, by assumption, that the functions $t_\omega(z_1, \ldots, z_k)$ are uniformly bounded by a positive constant $M$. Therefore, the constant function $T(z_1, \ldots, z_k) \coloneqq M$ is an envelope for $\mathcal{T}$, *i.e.*, $T$ satisfies $T(z_1, \ldots, z_k) \geq \sup_{\omega \in \Omega}|t_\omega(z_1, \ldots, z_k)|$ for any $(z_1, \ldots, z_k) \in \otimes^k \mathcal{Z}$. The envelope $T$ is, a fortiori, square $\mu$-integrable for any probability measure $\mu$ on $\otimes^k \mathcal{Z}$. Hence, we can apply [67, Maximal inequality] with the choice $T$ for the envelope function and set the integer $m$ appearing in the result to $m = d$ to get the following bound:

$$\mathbb{E}_{\mathbb{D}}\left[\sup_{\omega \in \Omega}\left|U_n^k t_\omega\right|^p\right]^{\frac{1}{p}} \leq n^{-\frac{k}{2}}\Gamma\mathbb{E}\left[\|T\|_{\mu_n, 2}\int_0^{\delta_n}\left(D\left(\epsilon\|T\|_{\mu_n, 2}, \mathcal{T}, L_2(\mu_n)\right)\right)^{\frac{1}{2dp}}\mathrm{d}\epsilon\right], \quad (19)$$

where $\Gamma$ is a positive universal constant[3] that depends only on $d$ and that we choose to be greater than $1$, while $\mu_n$ are suitably chosen probability measures on $\otimes^k \mathcal{Z}$ that possibly depend on the samples $z_1, \ldots, z_n$ and other random variables, and $\delta_n\|T\|_{\mu_n, 2} \coloneqq \sup_{\omega \in \Omega}\|t_\omega\|_{\mu_n, 2}$. Here, the expectation symbol in the right-hand side is over all randomness on which $\mu_n$ might depend. Note that the original result in [67, Maximal inequality] is stated using a slightly different definition of the packing number but which is still equivalent to the statement above in our setting[4].

In our setting, the envelope function is constant and equal to $M$, and by definition $\delta_n \leq 1$. Hence, the inequality in Equation (19) further becomes:

$$\mathbb{E}_{\mathbb{D}}\left[\sup_{\omega \in \Omega}\left|U_n^k t_\omega\right|^p\right]^{\frac{1}{p}} \leq n^{-\frac{k}{2}}M\Gamma\mathbb{E}\left[\int_0^1\left(D\left(\epsilon\|T\|_{\mu_n, 2}, \mathcal{T}, L_2(\mu_n)\right)\right)^{\frac{1}{2dp}}\mathrm{d}\epsilon\right]. \quad (20)$$

We simply need to control the packing number $D\left(\epsilon\|T\|_{\mu, 2}, \mathcal{T}, L_2(\mu)\right)$ independently of the probability measure $\mu$.

**Control on the packing number.** We have shown that the constant function $T(z_1, \ldots, z_k) \coloneqq M$ is an envelope for $\mathcal{T}$ which is, a fortiori, square $\mu$-integrable for any probability measure $\mu$ with $\|T\|_{\mu, 2} = M < +\infty$. Moreover, the functions $\omega \mapsto t_\omega(z_1, \ldots, z_k)$ are $L$-Lipschitz for any $(z_1, \ldots, z_k) \in \otimes^k \mathcal{Z}$. We can therefore apply Proposition F.1 which ensures the existence of a positive constant $c(\Omega)$ greater than $1$ and that depends only on $\Omega$ and $d$ so that the following estimate on the $\epsilon$-packing number of the class $\mathcal{T}$ w.r.t. $L_2(\mu)$ holds:

$$D\left(\epsilon\|T\|_{\mu, 2}, \mathcal{T}, L_2(\mu)\right) \leq \underbrace{c(\Omega)\left(\max\left(\frac{L\operatorname{diam}(\Omega)}{M}, 1\right)\right)^d}_{A}\left(\frac{1}{\epsilon}\right)^d, \qquad \forall \epsilon \in (0, 1]. \quad (21)$$

Combining Equation (21) with Equation (20) yields:

$$\mathbb{E}_{\mathbb{D}}\left[\sup_{\omega \in \Omega}\left|U_n^k t_\omega\right|^p\right]^{\frac{1}{p}} \leq n^{-\frac{k}{2}}M\Gamma\mathbb{E}\left[\int_0^1\left(A\epsilon^{-d}\right)^{\frac{1}{2dp}}\mathrm{d}\epsilon\right] = n^{-\frac{k}{2}}M\Gamma A^{\frac{1}{2dp}}\underbrace{\int_0^1\epsilon^{-\frac{1}{2p}}\mathrm{d}\epsilon}_{\leq 2}$$

$$\leq 2n^{-\frac{k}{2}}\Gamma c(\Omega)^{\frac{1}{2d}}\max\left(L\operatorname{diam}(\Omega), M^2\right)^{\frac{1}{2}},$$

---

[3]The constant $\Gamma$ appearing in [67, Maximal inequality] depends only on $k$, $p$ and $m$, *i.e.*, $\Gamma \coloneqq g(k, p, m)$. Since, we are only interested in $k \leq 2$ and $p \leq 2$ and $m$ is fixed to $d$, we choose $\Gamma$ to be $\max_{1 \leq k, p \leq 2} g(k, p, d)^{\frac{1}{p}}$, so that it is the same in all our cases.

[4]In [67, Maximal inequality], the author considers a modified version of the $\epsilon$-packing number (call it $\tilde{D}(\epsilon, \mathcal{T}, L_2(\mu))$) associated to $L_2(\mu)$ but endowed with a normalized version of the standard norm on $L_2(\mu)$: $\|f\|_\mu \coloneqq \frac{\|f\|_{\mu, 2}}{\|T\|_{\mu, 2}}$. Both numbers are related by the following identity: $\tilde{D}(\epsilon, \mathcal{T}, L_2(\mu)) = D(\epsilon\|T\|_{\mu, 2}, \mathcal{T}, L_2(\mu))$, thus making the statement (19) equivalent to the original statement in [67, Maximal inequality].

where, for the last inequality, we used that $A^{\frac{1}{2dp}} \leq A^{\frac{1}{2d}} = c(\Omega)^{\frac{1}{2d}} \max\left(\frac{L\operatorname{diam}(\Omega)}{M}, 1\right)^{\frac{1}{2}}$ since $A$ is greater than 1. The desired result follows after redefining $c(\Omega)$ as $2\Gamma c(\Omega)^{\frac{1}{2d}}$ which is a positive constant that depends only on $\Omega$ and $d$. $\qquad\square$

The following two propositions are particular instances of Theorem F.2 and will be used to obtain the main bounds.

**Proposition F.3** (Maximal inequality for empirical processes)**.** *Consider a parametric family $\mathcal{T} := \{\mathcal{Z} \ni z \mapsto t_\omega(z) \in \mathbb{R} \mid \omega \in \Omega\}$ of real-valued functions defined over a subset $\mathcal{Z}$ of a Euclidean space and indexed by a parameter $\omega \in \Omega$, where $\Omega$ is a compact subset of $\mathbb{R}^d$. Assume that all functions in $\mathcal{T}$ are uniformly bounded by a positive constant $M$ and that there exists a positive constant $L$ so that $\omega \mapsto t_\omega(z)$ is $L$-Lipschitz for any $z \in \mathcal{Z}$. Consider a probability distribution $\mathbb{D}$ over $\mathcal{Z}$ and let $(z_i)_{1 \leq i \leq n}$ be i.i.d. samples drawn from $\mathbb{D}$, then there exists a positive constant $c(\Omega)$ greater than 1 that depends only on $\Omega$ and $d$, such that for any integer $p \in \{1, 2\}$:*

$$\mathbb{E}_\mathbb{D}\left[\sup_{\omega\in\Omega}\left|\mathbb{E}_{z\sim\mathbb{D}}\left[t_\omega(z)\right] - \frac{1}{n}\sum_{i=1}^n t_\omega(z_i)\right|^p\right]^{\frac{1}{p}} \leq \sqrt{\frac{1}{n}}c(\Omega)\max(ML\operatorname{diam}(\Omega), M^2)^{\frac{1}{2}}.$$

*Proof.* The upper bound is a direct consequence of Theorem F.2. Indeed consider the family $\mathcal{S}$ of functions of the form $s_\omega(z) = t_\omega(z) - \mathbb{E}_{\bar{z}\sim\mathbb{D}}\left[t_\omega(\bar{z})\right]$, for any $z \in \mathcal{Z}$. Then clearly, the process $U_n^1 s_\omega := \frac{1}{n}\sum_{i=1}^n s_\omega(z_i)$ is degenerate of order $k = 1$, and the family $\mathcal{S}$ is uniformly bounded by $2M$ and is $2L$-Lipschitz. Hence, by Theorem F.2, the following maximal inequality holds:

$$\mathbb{E}_\mathbb{D}\left[\sup_{\omega\in\Omega}\left|U_n^1 s_\omega\right|^p\right]^{\frac{1}{p}} \leq 2n^{-\frac{1}{2}}c(\Omega)\max\left(ML\operatorname{diam}(\Omega), M^2\right)^{\frac{1}{2}}.$$

We get the desired upper bound by redefining $c(\Omega)$ to contain the factor 2. $\qquad\square$

**Proposition F.4** (Maximal inequality for $U$-processes of order 2)**.** *Consider a parametric family $\mathcal{T} := \{\mathcal{Z} \times \mathcal{Z} \ni (z, z') \mapsto t_\omega(z, z') \in \mathbb{R} \mid \omega \in \Omega\}$ of real-valued functions indexed by a parameter $\omega \in \Omega$, where $\Omega$ is a compact subset of $\mathbb{R}^d$ and $\mathcal{Z}$ is a subset of a Euclidean space. Assume that the functions in $\mathcal{T}$ are symmetric in their arguments, i.e., $t_\omega(z, z') = t_\omega(z', z)$. Additionally, assume that all functions in $\mathcal{T}$ are uniformly bounded by a positive constant $M$ and that there exists a positive constant $L$ so that $\omega \mapsto t_\omega(z, z')$ is $L$-Lipschitz for any $(z, z') \in \mathcal{Z} \times \mathcal{Z}$. Consider a probability distribution $\mathbb{D}$ over $\mathcal{Z}$ and let $(z_i)_{1 \leq i \leq n}$ be i.i.d. samples drawn from $\mathbb{D}$, and define the following statistic:*

$$\tau_\omega := \mathbb{E}_{z,z'\sim\mathbb{D}\otimes\mathbb{D}}\left[t_\omega(z, z')\right] + \frac{1}{n^2}\sum_{i,j=1}^n t_\omega(z_i, z_j) - \frac{2}{n}\sum_{i=1}^n \mathbb{E}_{z\sim\mathbb{D}}\left[t_\omega(z, z_i)\right].$$

*Then there exists a universal positive constant $c(\Omega)$ greater than 1 that depends only on $\Omega$ and $d$ such that:*

$$\mathbb{E}_\mathbb{D}\left[\sup_{\omega\in\Omega}|\tau_\omega|\right] \leq \frac{1}{n}c(\Omega)\max\left(ML\operatorname{diam}(\Omega), M^2\right)^{\frac{1}{2}}.$$

*Proof.* The proof will proceed by first decomposing $\tau_\omega$ into a sum of a degenerate $U$-process and a term of order $\mathcal{O}(\frac{1}{n})$. The maximal inequality for degenerate $U$-processes from [67] will be employed to obtain the desired bound.

**Decomposition of $\tau_\omega$.** Consider the following function defined over $\mathcal{Z} \times \mathcal{Z}$ and indexed by elements $\omega \in \Omega$:

$$s_\omega(z, z') = t_\omega(z, z') - \mathbb{E}_{\bar{z}\sim\mathbb{D}}\left[t_\omega(z, \bar{z})\right] - \mathbb{E}_{\bar{z}\sim\mathbb{D}}\left[t_\omega(\bar{z}, z')\right] + \mathbb{E}_{\bar{z},\underline{z}\sim\mathbb{D}\otimes\mathbb{D}}\left[t_\omega(\bar{z}, \underline{z})\right]. \qquad (22)$$

By direct calculation, we decompose $\tau_\omega$ into two higher order terms and a third term, $U_n^2 s_\omega$, involving $s_\omega$, which happens to be a $U$-statistic:

$$\tau_\omega = \overbrace{\frac{1}{n(n-1)}\sum_{\substack{i,j=1 \\ i\neq j}}^n s_\omega\left(z_i, z_j\right)}^{U_n^2 s_\omega} - \frac{1}{n^2(n-1)}\sum_{\substack{i,j=1 \\ i\neq j}}^n t_\omega(z_i, z_j) + \frac{1}{n^2}\sum_{i=1}^n t_\omega(z_i, z_i).$$

Using the triangle inequality in the above equality and recalling that, by assumption, $t_\omega(z, z')$ is uniformly bounded by a positive constant $M$, it follows that:

$$|\tau_\omega| \leq \left|U_n^2 s_\omega\right| + \frac{1}{n^2(n-1)} \sum_{\substack{i,j=1 \\ i \neq j}}^{n} |t_\omega(z_i, z_j)| + \frac{1}{n^2} \sum_{i=1}^{n} |t_\omega(z_i, z_i)| \leq \left|U_n^2 s_\omega\right| + \frac{2M}{n}.$$

Furthermore, taking the supremum over $\omega$ followed by the expectation over samples yields:

$$\mathbb{E}_\mathbb{D}\left[\sup_{\omega \in \Omega} |\tau_\omega|\right] \leq \mathbb{E}_\mathbb{D}\left[\sup_{\omega \in \Omega} \left|U_n^2 s_\omega\right|\right] + \frac{2M}{n}. \tag{23}$$

Hence, it only remains to control the first term in the above inequality. To this end, we will use a maximal inequality for degenerate $U$-processes due to [67].

**Maximal inequality for degenerate $U$-processes.** We will first check that $U_n^2 s_\omega$ is a degenerate statistic for a given $\omega \in \Omega$. Simple calculations show that for any $z$ in $\mathcal{Z}$:

$$\mathbb{E}_{\bar{z} \sim \mathbb{D}}\left[s_\omega(z, \bar{z})\right] = \mathbb{E}_{\bar{z} \sim \mathbb{D}}\left[s_\omega(\bar{z}, z)\right] = 0.$$

The above equalities precisely ensure that $U_n^2 s_\omega$ is a degenerate $U$-statistic for $\mathbb{D}$. Consider now the family $\mathcal{S} \coloneqq \left\{\mathcal{Z} \times \mathcal{Z} \ni (z, z') \mapsto s_\omega(z, z') \in \mathbb{R} \mid \omega \in \Omega\right\}$. We show that $\mathcal{S}$ is uniformly bounded and Lipschitz which allows to directly apply the result stated in Theorem F.2, which is a special case of the more general result in [67, Maximal inequality]. First note, by assumption, that the functions $t_\omega(z, z')$ are uniformly bounded by a positive constant $M$. Hence, using Equation (22), it follows that $s_\omega(z, z')$ is uniformly bounded by $4M$. Moreover, the functions $\omega \mapsto t_\omega(z, z')$ are $L$-Lipschitz for any $z, z'$ in $\mathcal{Z}$. Hence, from Equation (22), we directly have that $\omega \mapsto s_\omega(z, z')$ is $4L$-Lipschitz for any $z, z' \in \mathcal{Z}$. We can directly apply Theorem F.2 with $k = 2$ and $p = 1$ to $\mathcal{S}$ and get the following maximal inequality:

$$\mathbb{E}_\mathbb{D}\left[\sup_{\omega \in \Omega} \left|U_n^2 s_\omega\right|\right] \leq 4n^{-1} c(\Omega) \max\left(ML \operatorname{diam}(\Omega), M^2\right)^{\frac{1}{2}}.$$

We obtain an upper bound on $\mathbb{E}_\mathbb{D}\left[\sup_{\omega \in \Omega} |\tau_\omega|\right]$ by combining the above inequality with Equation (23), then noticing that $2M \leq 2c(\Omega) \max\left(L \operatorname{diam}(\Omega), M\right)$ so that:

$$\mathbb{E}_\mathbb{D}\left[\sup_{\omega \in \Omega} |\tau_\omega|\right] \leq 6n^{-1} c(\Omega) \max\left(ML \operatorname{diam}(\Omega), M^2\right)^{\frac{1}{2}}.$$

Finally, the desired result follows by redefining $c(\Omega)$ to include the factor 6 in the above inequality. $\square$

# G Differentiability Results

The proofs of Propositions B.1 and B.2 are direct applications of the following more general result.

**Proposition G.1.** *Let $\mathcal{U}$ be an open non-trivial subset of $\mathbb{R}^d$. Consider a real-valued function $\ell : (\omega, v, y) \mapsto \ell(\omega, v, y)$ defined on $\mathcal{U} \times \mathbb{R} \times \mathcal{Y}$ that is of class $C^3$ jointly in $(\omega, v)$ and whose derivatives are jointly continuous in $(\omega, v, y)$. For a given probability distribution $\mathbb{D}$ over $\mathcal{X} \times \mathcal{Y}$, consider the following functional defined over $\mathcal{U} \times \mathcal{H}$:*

$$L(\omega, h) \coloneqq \mathbb{E}_\mathbb{D}\left[\ell(\omega, h(x), y)\right].$$

*Under Assumptions (A) to (C), the following properties hold for $L$:*

- *$L$ admits finite values for any $(\omega, h) \in \mathcal{U} \times \mathcal{H}$.*

- *$(\omega, h) \mapsto L(\omega, h)$ is Fréchet differentiable with partial derivatives $\partial_\omega L(\omega, h)$ and $\partial_h L(\omega, h)$ at any point $(\omega, h) \in \mathcal{U} \times \mathcal{H}$ given by:*

$$\partial_\omega L(\omega, h) = \mathbb{E}_\mathbb{D}\left[\partial_\omega \ell(\omega, h(x), y)\right] \in \mathbb{R}^d,$$
$$\partial_h L(\omega, h) = \mathbb{E}_\mathbb{D}\left[\partial_v \ell(\omega, h(x), y) K(x, \cdot)\right] \in \mathcal{H}.$$

- *The map $(\omega, h) \mapsto \partial_h L(\omega, h)$ is differentiable. Moreover, for any $(\omega, h) \in \mathcal{U} \times \mathcal{H}$, its partial derivatives $\partial^2_{\omega, h} L(\omega, h)$ and $\partial^2_h L(\omega, h)$ at $(\omega, h)$ are Hilbert-Schmidt operators given by:*

$$\partial^2_{\omega, h} L(\omega, h) = \mathbb{E}_{\mathbb{D}} \left[ \partial^2_{\omega, v} \ell(\omega, h(x), y) K(x, \cdot) \right] \in \mathcal{L}(\mathcal{H}, \mathbb{R}^d),$$

$$\partial^2_h L(\omega, h) = \mathbb{E}_{\mathbb{D}} \left[ \partial^2_v \ell(\omega, h(x), y) K(x, \cdot) \otimes K(x, \cdot) \right] \in \mathcal{L}(\mathcal{H}, \mathcal{H}).$$

*Proof.* **Finite values.** Fix $\omega \in \mathcal{U}$ and $h \in \mathcal{H}$. We will first show that $h$ is bounded on $\mathcal{X}$. By the reproducing property, we know that $|h(x)| \leq \|h\|_{\mathcal{H}} \sqrt{K(x, x)}$ for any $x \in \mathcal{X}$. Moreover, the kernel $K$ is bounded by a constant $\kappa$ thanks to Assumption (B). Consequently, $|h(x)|$ is upper-bounded by $\|h\|_{\mathcal{H}} \sqrt{\kappa}$ for any $x \in \mathcal{X}$.

Denote by $\mathcal{I}$ the compact interval defined as $\mathcal{I} = \left[ - \|h\|_{\mathcal{H}} \sqrt{\kappa}, \|h\|_{\mathcal{H}} \sqrt{\kappa} \right]$. By Assumption (C), the set $\mathcal{Y}$ is compact so that $\mathcal{I} \times \mathcal{Y}$ is also compact. Moreover, we know, by assumption on $\ell$, that $(\omega, v, y) \mapsto \ell(\omega, v, y)$ is continuous on $\mathcal{U} \times \mathbb{R} \times \mathcal{Y}$. Therefore, $(v, y) \mapsto \ell(\omega, v, y)$ must be bounded by some finite constant $C$ on the compact set $\mathcal{I} \times \mathcal{Y}$. This allows to deduce that $(x, y) \mapsto \ell(\omega, h(x), y)$ is bounded by $C$ for any $(x, y) \in \mathcal{X} \times \mathcal{Y}$ and a fortiori $\mathbb{D}$-integrable, which shows that $L(\omega, h)$ is finite.

**Fréchet differentiability of** $L$. Let $(\omega, h) \in \mathcal{U} \times \mathcal{H}$. Consider $(\omega_j, h_j)_{j \geq 1}$ a sequence of elements in $\mathcal{U} \times \mathcal{H}$ converging to it, *i.e.*, $(\omega_j, h_j) \to (\omega, h)$ with $(\omega_j, h_j) \neq (\omega, h)$ for any $j \geq 0$. Define the sequence of functions $r_j : \mathcal{X} \times \mathcal{Y} \to \mathbb{R}$ for any $(x, y) \in \mathcal{X} \times \mathcal{Y}$ as follows:

$$r_j(x, y) =$$
$$\frac{\ell(\omega_j, h_j(x), y) - \ell(\omega, h(x), y) - \langle \partial_v \ell(\omega, h(x), y) K(x, \cdot), h_j - h \rangle_{\mathcal{H}} - \langle \partial_\omega \ell(\omega, h(x), y), \omega_j - \omega \rangle}{\|(\omega_j, h_j) - (\omega, h)\|}.$$

$$(24)$$

We will first show that $\mathbb{E}_{\mathbb{D}}\left[ |r_j(x, y)| \right]$ converges to 0 by the dominated convergence theorem [62, Theorem 1.34]. By the reproducing property, note that $\ell(\omega, h(x), y) = \ell(\omega, \langle h, K(x, \cdot) \rangle_{\mathcal{H}}, y)$. Hence, since $\ell$ is jointly differentiable in $(\omega, v)$ for any $y$, it follows that $(\omega, h) \mapsto \ell(\omega, h(x), y)$ is also differentiable for any $(x, y)$ by composition with the evaluation map $(\omega, h) \mapsto (\omega, \langle h, K(x, \cdot) \rangle_{\mathcal{H}}$ which is differentiable. Hence, the sequence $r_j(x, y)$ converges to 0 for any $(x, y) \in \mathcal{X} \times \mathcal{Y}$. Moreover, by the mean-value theorem, there exists $0 \leq c_j \leq 1$ such that:

$$r_j(x, y) =$$
$$\frac{\left\langle \left( \partial_v \ell(\bar{\omega}_j, \bar{h}_j(x), y) - \partial_v \ell(\omega, h(x), y) \right) K(x, \cdot), h_j - h \right\rangle_{\mathcal{H}} - \langle \partial_\omega \ell(\bar{\omega}_j, \bar{h}_j(x), y) - \partial_\omega \ell(\omega, h(x), y), \omega_j - \omega \rangle}{\|(\omega_j, h_j) - (\omega, h)\|},$$

where $(\bar{\omega}_j, \bar{h}_j) := (1 - c_j)(\omega, h) + c_j(\omega_j, h_j)$. We will show that $r_j(x, y)$ is bounded for $j$ large enough. We first construct a compact set that will contain all elements of the form $(\omega_j, h_j(x), y)$ and $(\bar{\omega}_j, \bar{h}_j(x), y)$ for all $j$ large enough. Since $\omega$ is an element in the open set $\mathcal{U}$, there exists a closed ball $\mathcal{B}(\omega, R)$ centered at $\omega$ and with some radius $R$ small enough so that $\mathcal{B}(\omega, R)$ is included in $\mathcal{U}$. For all $j$ large enough, $\omega_j$ and $\bar{\omega}_j$ belong to $\mathcal{B}(\omega, R)$ as these sequences converge to $\omega$. Moreover, $h_j$ and $\bar{h}_j$ are convergent sequences. Consequently, they must be bounded by some constant $B$. By the reproducing property, and recalling that the kernel $K$ is bounded by $\kappa$ by Assumption (B), it follows that $\max(|h_j(x)|, |\bar{h}_j(x)|) \leq B\kappa$. Consider now the set $\mathcal{W} := \mathcal{B}(\omega, R) \times \mathcal{B}_1(0, B\kappa) \times \mathcal{Y}$ which is a product of compact sets (recalling that $\mathcal{Y}$ is compact by Assumption (C)), where $\mathcal{B}_1(0, B\kappa)$ is the closed ball in $\mathbb{R}$ centered at 0 and of radius $B\kappa$. For $j$ large enough, we have established that $(\omega_j, h_j(x), y)$ and $(\bar{\omega}_j, \bar{h}_j(x), y)$ belong to $\mathcal{W}$ for any $(x, y) \in \mathcal{X} \times \mathcal{Y}$. Since, by assumption on $\ell$, $\partial_v \ell(\omega, v, y)$ and $\partial_\omega \ell(\omega, v, y)$ are continuous, they must be bounded on the compact set $\mathcal{W}$ by some constant $C$. This allows to deduce from the expression of $r_j(x, y)$ above that $r_j(x, y)$ is bounded, and a fortiori dominated by an integrable function (a constant function). We then deduce that $\mathbb{E}_{\mathbb{D}}\left[ |r_j(x, y)| \right]$ converges to 0 by application of the dominated convergence theorem [62, Theorem 1.34].

Recalling Equation (24), $\mathbb{E}_{\mathbb{D}}\left[ r_j(x, y) \right]$ admits the following expression:

$$\mathbb{E}_{\mathbb{D}}\left[ r_j(x, y) \right] =$$
$$\frac{L(\omega_j, h_j) - L(\omega, h) - \mathbb{E}_{\mathbb{D}}\left[ \langle \partial_v \ell(\omega, h(x), y) K(x, \cdot), h_j - h \rangle_{\mathcal{H}} \right] - \langle \mathbb{E}_{\mathbb{D}}\left[ \partial_\omega \ell(\omega, h(x), y) \right], \omega_j - \omega \rangle}{\|(\omega_j, h_j) - (\omega, h)\|}.$$

$$(25)$$

The convergence to 0 of the above expression precisely means that $L$ is differentiable at $(\omega, h)$ provided that the linear form $g \mapsto \mathbb{E}_{\mathbb{D}}\left[\langle \partial_v \ell\left(\omega, h(x), y\right) K(x, \cdot), g\rangle_{\mathcal{H}}\right]$ is bounded. To establish this fact, consider the RKHS-valued function $(x, y) \mapsto \partial_v \ell\left(\omega, h(x), y\right) K(x, \cdot)$. This function is Bochner-integrable in the sense that $\mathbb{E}_{\mathbb{D}}\left[\|\partial_v \ell\left(\omega, h(x), y\right) K(x, \cdot)\|_{\mathcal{H}}\right]$ is finite [25, Definition 1, Chapter 2]. Indeed, we have the following:

$$\mathbb{E}_{\mathbb{D}}\left[\|\partial_v \ell\left(\omega, h(x), y\right) K(x, \cdot)\|_{\mathcal{H}}\right] := \mathbb{E}_{\mathbb{D}}\left[|\partial_v \ell\left(\omega, h(x), y\right)| \sqrt{K(x, x)}\right]$$
$$\leq \sqrt{\kappa}\,\mathbb{E}_{\mathbb{D}}\left[|\partial_v \ell\left(\omega, h(x), y\right)|\right] < +\infty,$$

where, for the inequality, we used that $(x, y) \mapsto \partial_v \ell(\omega, h(x), y)$ is bounded as shown previously. Consequently, $\mathbb{E}_{\mathbb{D}}\left[\partial_v \ell\left(\omega, h(x), y\right) K(x, \cdot)\right]$ is an element in $\mathcal{H}$ satisfying:

$$\langle \mathbb{E}_{\mathbb{D}}\left[\partial_v \ell\left(\omega, h(x), y\right) K(x, \cdot)\right], g\rangle_{\mathcal{H}} = \mathbb{E}_{\mathbb{D}}\left[\langle \partial_v \ell\left(\omega, h(x), y\right) K(x, \cdot), g\rangle_{\mathcal{H}}\right], \forall g \in \mathcal{H}.$$

The above property follows from [25, Theorem 6, Chapter 2] for Bochner-integrable functions that allows exchanging the integral and the application of a continuous linear map (here the scalar product with an element $g$). The above identity establishes that $g \mapsto \mathbb{E}_{\mathbb{D}}\left[\langle \partial_v \ell\left(\omega, h(x), y\right) K(x, \cdot), g\rangle_{\mathcal{H}}\right]$ is bounded and provides the desired expression for $\partial_h L(\omega, h)$. The expression for $\partial_\omega L(\omega, h)$ directly follows from the last term in Equation (25).

**Fréchet differentiability of $\partial_h L$.** We use the same proof strategy as for the differentiability of $L$.

Let $(\omega, h) \in \mathcal{U} \times \mathcal{H}$. Consider $(\omega_j, h_j)_{j \geq 1}$ a sequence of elements in $\mathcal{U} \times \mathcal{H}$ converging to it, *i.e.*, $(\omega_j, h_j) \to (\omega, h)$ with $(\omega_j, h_j) \neq (\omega, h)$ for any $j \geq 0$. Define the sequence of functions $s_j : \mathcal{X} \times \mathcal{Y} \to \mathcal{H}$ as follows:

$$\begin{aligned}
\|(\omega_j, h_j) - (\omega, h)\| \, s_j(x, y) = & \left(\partial_v \ell\left(\omega_j, h_j(x), y\right) - \partial_v \ell\left(\omega, h(x), y\right)\right) K(x, \cdot) \\
& - \partial_v^2 \ell\left(\omega, h(x), y\right) K(x, \cdot) \otimes K(x, \cdot)\,(h_j - h) \\
& - (\omega_j - \omega)^\top \partial_{\omega, v}^2 \ell(\omega, h(x), y) K(x, \cdot).
\end{aligned} \tag{26}$$

We will first show that $\mathbb{E}_{\mathbb{D}}\left[\|s_j(x, y)\|_{\mathcal{H}}\right]$ converges to 0 by the dominated convergence theorem for Bochner-integrable functions [25, Theorem 3, Chapter 2]. By the reproducing property, note that $\partial_v \ell(\omega, h(x), y) K(x, \cdot) = \partial_v \ell(\omega, \langle h, K(x, \cdot)\rangle_{\mathcal{H}}, y) K(x, \cdot)$. Hence, since $(\omega, v) \mapsto \partial_v \ell(\omega, v, y)$ is jointly differentiable in $(\omega, v)$ for any $y$, it follows that $(\omega, h) \mapsto \partial_v \ell(\omega, h(x), y) K(x, \cdot)$ is also differentiable for any $(x, y)$ by composition with the evaluation map $(\omega, h) \mapsto (\omega, \langle h, K(x, \cdot)\rangle_{\mathcal{H}}$ which is differentiable. Hence, the sequence $s_j(x, y)$ converges to 0 for any $(x, y) \in \mathcal{X} \times \mathcal{Y}$. Moreover, by the mean-value theorem, there exists $0 \leq c_j \leq 1$ such that:

$$\begin{aligned}
\|(\omega_j, h_j) - (\omega, h)\| \, s_j(x, y) = & \partial_v^2 \ell\left(\bar{\omega}_j, \bar{h}_j(x), y\right) K(x, \cdot) \otimes K(x, \cdot)\,(h_j - h) \\
& + (\omega_j - \omega)^\top \partial_{\omega, v}^2 \ell(\bar{\omega}_j, \bar{h}_j(x), y) K(x, \cdot) \\
& - \partial_v^2 \ell\left(\omega, h(x), y\right) K(x, \cdot) \otimes K(x, \cdot)\,(h_j - h) \\
& - (\omega_j - \omega)^\top \partial_{\omega, v}^2 \ell(\omega, h(x), y) K(x, \cdot),
\end{aligned}$$

where $(\bar{\omega}_j, \bar{h}_j) := (1 - c_j)(\omega, h) + c_j(\omega_j, h_j)$. Using the same construction as for the Fréchet differentiability, we find a compact set $\mathcal{W}$ containing all elements $(\omega_j, h_j(x), y)$ and $(\bar{\omega}_j, \bar{h}_j(x), y)$ for any $(x, y) \in \mathcal{X} \times \mathcal{Y}$ and all $j$ large enough. On such set, $\partial_v^2 \ell(\omega, v, y)$ and $\partial_{\omega, v}^2 \ell(\omega, v, y)$ are bounded by some constant $C$. Consequently, we can write:

$$\begin{aligned}
\|(\omega_j, h_j) - (\omega, h)\| \, \|s_j(x, y)\|_{\mathcal{H}} \leq & 2C \|K(x, \cdot) \otimes K(x, \cdot)\,(h_j - h)\|_{\mathcal{H}} + 2C \|\omega_j - \omega\| \|K(x, \cdot)\|_{\mathcal{H}} \\
\leq & 2C\kappa \|h_j - h\|_{\mathcal{H}} + 2C\sqrt{\kappa} \|\omega_j - \omega\|.
\end{aligned}$$

This already establishes that $s_j(x, y)$ is bounded so that $\mathbb{E}_{\mathbb{D}}\left[\|s_j(x, y)\|_{\mathcal{H}}\right]$ converges to 0 by application of the dominated convergence theorem. Recalling Equation (26), $\mathbb{E}_{\mathbb{D}}\left[s_j(x, y)\right]$ admits the following expression:

$$\begin{aligned}
\|(\omega_j, h_j) - (\omega, h)\| \, \mathbb{E}_{\mathbb{D}}\left[s_j(x, y)\right] = & \partial_h L(\omega_j, h_j) - \partial_h L(\omega, h) \\
& - \mathbb{E}_{\mathbb{D}}\left[\partial_v^2 \ell\left(\omega, h(x), y\right) K(x, \cdot) \otimes K(x, \cdot)\,(h_j - h)\right] \\
& - (\omega_j - \omega)^\top \mathbb{E}_{\mathbb{D}}\left[\partial_{\omega, v}^2 \ell(\omega, h(x), y) K(x, \cdot)\right].
\end{aligned}$$

The convergence to 0 of the above expression precisely means that $L$ is differentiable at $(\omega, h)$ provided that: (1) $\mathbb{E}_{\mathbb{D}}\left[\partial^2_{\omega,v}\ell(\omega, h(x), y)K(x, \cdot)\right]$ is an element in $\mathcal{H}^d$, and (2) the linear map $g \mapsto \mathbb{E}_{\mathbb{D}}\left[\partial^2_v\ell(\omega, h(x), y)(K(x, \cdot) \otimes K(x, \cdot))g\right]$ is bounded. Using the same strategy to establish Bochner's integrability of $(x, y) \mapsto \partial_v\ell(\omega, h(x), y)K(x, \cdot)$, we can show that $(x, y) \mapsto \partial^2_{\omega,v}\ell(\omega, h(x), y)K(x, \cdot)$ is also Bochner-integrable so that $\mathbb{E}_{\mathbb{D}}\left[\partial^2_{\omega,v}\ell(\omega, h(x), y)K(x, \cdot)\right]$ is indeed an element in $\mathcal{H}^d$. This also establishes the expression of $\partial_{\omega,h}L(\omega, h)$. Similarly, we consider the operator-valued function $\xi : (x, y) \mapsto \partial^2_v\ell(\omega, h(x), y)K(x, \cdot) \otimes K(x, \cdot)$ with values in the space of Hilbert-Schmidt operators on $\mathcal{H}$. The Hilbert-Schmidt (HS) norm of such function satisfies the following inequality:

$$\mathbb{E}_{\mathbb{D}}\left[\left\|\partial^2_v\ell(\omega, h(x), y)K(x, \cdot) \otimes K(x, \cdot)\right\|_{\mathrm{HS}}\right] := \mathbb{E}_{\mathbb{D}}\left[\left|\partial^2_v\ell(\omega, h(x), y)\right|K(x, x)\right] \leq \kappa C < +\infty.$$

Therefore, the function $\xi$ is Bochner-integrable, so that $\mathbb{E}_{\mathbb{D}}\left[\partial^2_v\ell(\omega, h(x), y)K(x, \cdot) \otimes K(x, \cdot)\right]$ is a Hilbert-Schmidt operator satisfying:

$$\mathbb{E}_{\mathbb{D}}\left[\partial^2_v\ell(\omega, h(x), y)K(x, \cdot) \otimes K(x, \cdot)\right]g = \mathbb{E}_{\mathbb{D}}\left[\partial^2_v\ell(\omega, h(x), y)(K(x, \cdot) \otimes K(x, \cdot))g\right], \forall g \in \mathcal{H}.$$

The above property follows from [25, Theorem 6, Chapter 2] for Bochner-integrable functions that allows exchanging the integral and the application of a continuous linear map (here the scalar product with an element $g$). Hence, from the above identity, we deduce the desired expression for $\partial^2_h L(\omega, h)$. $\qquad\square$

## H   Auxiliary Technical Lemmas

**Lemma H.1.** *Let $A$ and $A'$ be two bounded operators from $\mathcal{H}$ to $\mathbb{R}^d$, and $B$ and $B'$ be two bounded and invertible operators from $\mathcal{H}$ to itself. Assume that $B \geq \lambda \operatorname{Id}_{\mathcal{H}}$ and $B' \geq \lambda \operatorname{Id}_{\mathcal{H}}$. Then, the following inequalities hold:*

$$\left\|AB^{-1} - A'(B')^{-1}\right\|_{\mathrm{op}} \leq \frac{\|A\|_{\mathrm{op}}}{\lambda^2}\|B - B'\|_{\mathrm{op}} + \frac{1}{\lambda}\|A - A'\|_{\mathrm{op}},$$

$$\left\|AB^{-1}\right\|_{\mathrm{op}} \leq \lambda^{-1}\|A\|_{\mathrm{op}}, \qquad \left\|A'(B')^{-1}\right\|_{\mathrm{op}} \leq \lambda^{-1}\|A'\|_{\mathrm{op}}.$$

*Proof.* By the triangle inequality and the sub-multiplicative property of the operator norm $\|\cdot\|_{\mathrm{op}}$, we have:

$$
\begin{aligned}
\left\|AB^{-1} - A'(B')^{-1}\right\|_{\mathrm{op}} &\leq \left\|AB^{-1} - A(B')^{-1}\right\|_{\mathrm{op}} + \left\|A(B')^{-1} - A'(B')^{-1}\right\|_{\mathrm{op}} \\
&= \left\|A\left(B^{-1} - (B')^{-1}\right)\right\|_{\mathrm{op}} + \left\|(A - A')(B')^{-1}\right\|_{\mathrm{op}} \\
&\leq \|A\|_{\mathrm{op}}\left\|B^{-1} - (B')^{-1}\right\|_{\mathrm{op}} + \|A - A'\|_{\mathrm{op}}\left\|(B')^{-1}\right\|_{\mathrm{op}} \\
&= \|A\|_{\mathrm{op}}\left\|B^{-1}(B' - B)(B')^{-1}\right\|_{\mathrm{op}} + \|A - A'\|_{\mathrm{op}}\left\|(B')^{-1}\right\|_{\mathrm{op}} \\
&\leq \|A\|_{\mathrm{op}}\left\|B^{-1}\right\|_{\mathrm{op}}\|B' - B\|_{\mathrm{op}}\left\|(B')^{-1}\right\|_{\mathrm{op}} + \|A - A'\|_{\mathrm{op}}\left\|(B')^{-1}\right\|_{\mathrm{op}}.
\end{aligned}
\tag{27}
$$

Since $B \geq \lambda \operatorname{Id}_{\mathcal{H}}$ and $B' \geq \lambda \operatorname{Id}_{\mathcal{H}}$, we obtain:

$$\left\|B^{-1}\right\|_{\mathrm{op}} \leq \frac{1}{\lambda} \quad \text{and} \quad \left\|(B')^{-1}\right\|_{\mathrm{op}} \leq \frac{1}{\lambda}.$$

Substituting these into Equation (27), we get:

$$\left\|AB^{-1} - A'(B')^{-1}\right\|_{\mathrm{op}} \leq \frac{\|A\|_{\mathrm{op}}}{\lambda^2}\|B - B'\|_{\mathrm{op}} + \frac{1}{\lambda}\|A - A'\|_{\mathrm{op}}.$$

This proves the first inequality. The remaining two inequalities follow directly from the sub-multiplicative property of the operator norm $\|\cdot\|_{\mathrm{op}}$ and the assumptions $B \geq \lambda \operatorname{Id}_{\mathcal{H}}$ and $B' \geq \lambda \operatorname{Id}_{\mathcal{H}}$. $\qquad\square$

**Lemma H.2.** *Let $f : \mathcal{H} \to \mathbb{R}$ be a $\lambda$-strongly convex and Fréchet differentiable function. Denote by $h^\star \in \mathcal{H}$ its minimizer. Then, for any $h \in \mathcal{H}$, the following holds:*

$$\|h - h^\star\|_{\mathcal{H}} \leq \frac{1}{\lambda}\|\partial_h f(h)\|_{\mathcal{H}}.$$

*Proof.* Let $h \in \mathcal{H}$.

**Case 1:** $h = h^\star$**.** The proof is straightforward.

**Case 2:** $h \neq h^\star$**.** Given that $f$ is $\lambda$-strongly convex, we have:

$$f(h) - f(h^\star) \geq \langle \partial_h f(h^\star), h - h^\star \rangle_{\mathcal{H}} + \frac{\lambda}{2} \|h - h^\star\|_{\mathcal{H}}^2,$$

$$\text{and } f(h^\star) - f(h) \geq \langle \partial_h f(h), h^\star - h \rangle_{\mathcal{H}} + \frac{\lambda}{2} \|h - h^\star\|_{\mathcal{H}}^2.$$

After summing these two inequalities, noticing that $\partial_h f(h^\star) = 0$, and rearranging the terms, we obtain:

$$\langle \partial_h f(h), h - h^\star \rangle_{\mathcal{H}} \geq \lambda \|h - h^\star\|_{\mathcal{H}}^2.$$

After using the Cauchy-Schwarz inequality, we get:

$$\|\partial_h f(h)\|_{\mathcal{H}} \|h - h^\star\|_{\mathcal{H}} \geq \lambda \|h - h^\star\|_{\mathcal{H}}^2.$$

Dividing by $\lambda \|h - h^\star\|_{\mathcal{H}} \neq 0$ concludes the proof. $\qquad\square$

**Lemma H.3.** *Let $\mathcal{X}$ be a subset of $\mathbb{R}^p$, $\mathcal{Y}$ be a subset of $\mathbb{R}^q$, and $\mathbb{D}$ be a probability distribution over $\mathcal{X} \times \mathcal{Y}$. Given i.i.d. samples $(x_i, y_i)_{1 \leq i \leq n}$ drawn from $\mathbb{D}$, consider a function $g : \mathcal{X} \times \mathcal{Y} \to \mathbb{R}$ of class $C^1$ such that the operator $A : \mathcal{H} \to \mathcal{H}$ defined as:*

$$A := \mathbb{E}_{(x,y)\sim\mathbb{D}} [g(x,y)K(x,\cdot) \otimes K(x,\cdot)] - \frac{1}{n} \sum_{i=1}^{n} g(x_i, y_i) K(x_i, \cdot) \otimes K(x_i, \cdot)$$

*is Hilbert-Schmidt. Then, the following holds:*

$$\|A\|_{\mathrm{HS}}^2 = \mathbb{E}_{(x,y),(x',y')\sim\mathbb{D}\otimes\mathbb{D}} \Big[ g(x,y)g(x',y')K^2(x,x') \Big] + \frac{1}{n^2} \sum_{i,j=1}^{n} g(x_i, y_i)g(x_j, y_j)K^2(x_i, x_j)$$

$$- \frac{2}{n} \sum_{i=1}^{n} \mathbb{E}_{(x,y)\sim\mathbb{D}} \Big[ g(x_i, y_i)g(x,y)K^2(x, x_i) \Big].$$

*Proof.* Define $s := \mathbb{E}_{(x,y)\sim\mathbb{D}} [g(x,y)K(x,\cdot) \otimes K(x,\cdot)]$ and $\hat{s} := \frac{1}{n} \sum_{i=1}^{n} g(x_i, y_i)K(x_i,\cdot) \otimes K(x_i,\cdot)$. We have:

$$\|A\|_{\mathrm{HS}}^2 = \|s - \hat{s}\|_{\mathrm{HS}}^2 = \|s\|_{\mathrm{HS}}^2 + \|\hat{s}\|_{\mathrm{HS}}^2 - 2 \langle s, \hat{s} \rangle_{\mathrm{HS}}. \tag{28}$$

Next, we compute each of the following quantities: $\|s\|_{\mathrm{HS}}^2$, $\|\hat{s}\|_{\mathrm{HS}}^2$, and $\langle s, \hat{s} \rangle_{\mathrm{HS}}$, separately. Simple calculations yield:

$$
\begin{aligned}
\|s\|_{\mathrm{HS}}^2 &= \left\| \mathbb{E}_{(x,y)\sim\mathbb{D}} \left[ g(x,y) K(x,\cdot) \otimes K(x,\cdot) \right] \right\|_{\mathrm{HS}}^2 \\
&= \left\langle \mathbb{E}_{(x,y)\sim\mathbb{D}} \left[ g(x,y) K(x,\cdot) \otimes K(x,\cdot) \right], \mathbb{E}_{(x',y')\sim\mathbb{D}} \left[ g(x',y') K(x',\cdot) \otimes K(x',\cdot) \right] \right\rangle_{\mathrm{HS}} \\
&= \mathbb{E}_{(x,y),(x',y')\sim\mathbb{D}\otimes\mathbb{D}} \left[ g(x,y) g(x',y') \left\langle K(x,\cdot) \otimes K(x,\cdot), K(x',\cdot) \otimes K(x',\cdot) \right\rangle_{\mathrm{HS}} \right] \\
&= \mathbb{E}_{(x,y),(x',y')\sim\mathbb{D}\otimes\mathbb{D}} \left[ g(x,y) g(x',y') K^2(x,x') \right],
\end{aligned}
$$

$$
\begin{aligned}
\|\hat{s}\|_{\mathrm{HS}}^2 &= \frac{1}{n^2} \left\| \sum_{i=1}^n g(x_i,y_i) K(x_i,\cdot) \otimes K(x_i,\cdot) \right\|_{\mathrm{HS}}^2 \\
&= \frac{1}{n^2} \left\langle \sum_{i=1}^n g(x_i,y_i) K(x_i,\cdot) \otimes K(x_i,\cdot), \sum_{j=1}^n g(x_j,y_j) K(x_j,\cdot) \otimes K(x_j,\cdot) \right\rangle_{\mathrm{HS}} \\
&= \frac{1}{n^2} \sum_{i,j=1}^n g(x_i,y_i) g(x_j,y_j) \left\langle K(x_i,\cdot) \otimes K(x_i,\cdot), K(x_j,\cdot) \otimes K(x_j,\cdot) \right\rangle_{\mathrm{HS}} \\
&= \frac{1}{n^2} \sum_{i,j=1}^n g(x_i,y_i) g(x_j,y_j) K^2(x_i,x_j),
\end{aligned}
$$

$$
\begin{aligned}
\langle s, \hat{s} \rangle_{\mathrm{HS}} &= \left\langle \mathbb{E}_{(x,y)\sim\mathbb{D}} \left[ g(x,y) K(x,\cdot) \otimes K(x,\cdot) \right], \frac{1}{n} \sum_{i=1}^n g(x_i,y_i) K(x_i,\cdot) \otimes K(x_i,\cdot) \right\rangle_{\mathrm{HS}} \\
&= \frac{1}{n} \sum_{i=1}^n \mathbb{E}_{(x,y)\sim\mathbb{D}} \left[ g(x_i,y_i) g(x,y) \left\langle K(x,\cdot) \otimes K(x,\cdot), K(x_i,\cdot) \otimes K(x_i,\cdot) \right\rangle_{\mathrm{HS}} \right] \\
&= \frac{1}{n} \sum_{i=1}^n \mathbb{E}_{(x,y)\sim\mathbb{D}} \left[ g(x_i,y_i) g(x,y) K^2(x,x_i) \right].
\end{aligned}
$$

After substituting the obtained results into Equation (28) and rearranging, we obtain:

$$
\begin{aligned}
\|A\|_{\mathrm{HS}}^2 =& \mathbb{E}_{(x,y),(x',y')\sim\mathbb{D}\otimes\mathbb{D}} \left[ g(x,y) g(x',y') K^2(x,x') \right] + \frac{1}{n^2} \sum_{i,j=1}^n g(x_i,y_i) g(x_j,y_j) K^2(x_i,x_j) \\
& - \frac{2}{n} \sum_{i=1}^n \mathbb{E}_{(x,y)\sim\mathbb{D}} \left[ g(x_i,y_i) g(x,y) K^2(x,x_i) \right].
\end{aligned}
$$

$\square$

# I   Details on Experiments and Additional Numerical Results

In this section, we provide details on the experimental setting used to obtain Figure 2 and include additional numerical results in Appendix I.5. We recall the formulation of the instrumental variable regression problem introduced in Section 2.2:

$$
\min_{\omega \in \mathbb{R}^d} \mathcal{F}(\omega) := L_{out}(\omega, h_\omega^\star) = \frac{1}{2} \mathbb{E}_{(x,y)\sim\mathbb{Q}} \left[ |h_\omega^\star(x) - y|^2 \right]
$$

$$
\text{s.t.} \quad h_\omega^\star = \underset{h \in \mathcal{H}}{\arg\min}\, L_{in}(\omega, h) = \frac{1}{2} \mathbb{E}_{(x,t)\sim\mathbb{P}} \left[ |h(x) - \omega^\top \phi(t)|^2 \right] + \frac{\lambda}{2} \|h\|_{\mathcal{H}}^2,
$$

where $\phi(t) = (\phi_1(t), \ldots, \phi_d(t))^\top \in \mathbb{R}^d$ is the feature map. We begin by deriving a closed-form expression for $\hat{h}_\omega$ (the empirical counterpart of $h_\omega^\star$), which is key to obtaining closed-form expressions for $\widehat{\mathcal{F}}(\omega)$ and $\widehat{\nabla\mathcal{F}}(\omega)$, and thus accurate approximations of $\mathcal{F}(\omega)$ and $\nabla\mathcal{F}(\omega)$.

## I.1 Closed-form expression for $\hat{h}_\omega$

Let $\omega \in \mathbb{R}^d$. By the first-order optimality condition, the gradient of $\widehat{L}_{in}$ with respect to its second argument must vanish at $\hat{h}_\omega$, i.e., $\partial_h \widehat{L}_{in}(\omega, \hat{h}_\omega) = 0$. Proposition B.1 implies that $\hat{h}_\omega$ satisfies the following equation:

$$\frac{1}{n} \sum_{i=1}^n \left[ (\hat{h}_\omega(x_i) - \omega^\top \phi(t_i)) K(x_i, \cdot) \right] + \lambda \hat{h}_\omega = 0,$$

with $(x_i, t_i)_{1 \le i \le n}$ being $n$ samples drawn from the distribution $\mathbb{P}$. After using the reproducing property of the RKHS $\mathcal{H}$ and rearranging the terms, we arrive at the following closed-form expression for $\hat{h}_\omega$:

$$\hat{h}_\omega = (\widehat{\Sigma}_\lambda^{-1} \widehat{\Phi})^\top \omega \in \mathcal{H},$$

where $\widehat{\Sigma}_\lambda = \widehat{\Sigma} + \lambda \operatorname{Id}_{\mathcal{H}}$ is an operator from $\mathcal{H}$ to $\mathcal{H}$ with $\widehat{\Sigma} = \frac{1}{n} \sum_{i=1}^n K(x_i, \cdot) \otimes K(x_i, \cdot)$ being the empirical covariance operator and $\widehat{\Phi} = \frac{1}{n} \sum_{i=1}^n \phi(t_i) K(x_i, \cdot) = (\widehat{\Phi}_1, \ldots, \widehat{\Phi}_d)^\top \in \mathcal{H}^d$. Next, we compute a closed-form expression for $\widehat{\Sigma}_\lambda^{-1} \widehat{\Phi}$, which can be determined as the solution $\hat{b} = (\hat{b}_1, \ldots, \hat{b}_d)^\top \in \mathcal{H}^d$ of the following minimization problem:

$$\hat{b}_l = \arg\min_{b_l \in \mathcal{H}} \frac{1}{2} b_l^\top \widehat{\Sigma}_\lambda b_l - b_l^\top \widehat{\Phi}_l, \quad \text{for any } 1 \le l \le d.$$

After expanding the terms and rearranging, this minimization problem is equivalent to:

$$\hat{b}_l = \arg\min_{b_l \in \mathcal{H}} \Psi_l(b_l(x_1), \ldots, b_l(x_n), \|b_l\|_{\mathcal{H}}), \quad \text{for any } 1 \le l \le d,$$

where, for any $e_1, \ldots, e_n, e \in \mathbb{R}$, $\Psi_l(e_1, \ldots, e_n, e) = \frac{1}{2n} \sum_{i=1}^n e_i^2 - \frac{1}{n} \sum_{i=1}^n \phi_l(t_i) e_i + \frac{\lambda}{2} e^2$. By the representer theorem, for any $1 \le l \le d$, $\hat{b}_l$ can be expressed as:

$$\hat{b}_l = \sum_{i=1}^n \hat{\mathbf{c}}_{i,l} K(x_i, \cdot),$$

where $\hat{\mathbf{c}}_l = (\hat{\mathbf{c}}_{1,l}, \ldots, \hat{\mathbf{c}}_{n,l})^\top \in \mathbb{R}^n$ satisfies:

$$\hat{\mathbf{c}}_l = \arg\min_{\mathbf{c}_l \in \mathbb{R}^n} \Psi_l \left( [\mathbf{K} \mathbf{c}_l]_1, \ldots, [\mathbf{K} \mathbf{c}_l]_n, \mathbf{c}_l^\top \mathbf{K} \mathbf{c}_l \right) := \frac{1}{2n} \mathbf{c}_l^\top \mathbf{K}^2 \mathbf{c}_l - \frac{1}{n} \mathbf{F}_l^\top \mathbf{K} \mathbf{c}_l + \frac{\lambda}{2} \mathbf{c}_l^\top \mathbf{K} \mathbf{c}_l,$$

where $\mathbf{F}_l = (\phi_l(t_1), \ldots, \phi_l(t_n))^\top \in \mathbb{R}^n$. By the first-order optimality condition, we have:

$$\nabla_{\mathbf{c}_l} \Psi_l \left( [\mathbf{K} \hat{\mathbf{c}}_l]_1, \ldots, [\mathbf{K} \hat{\mathbf{c}}_l]_n, \hat{\mathbf{c}}_l^\top \mathbf{K} \hat{\mathbf{c}}_l \right) = 0, \text{ which results in } \hat{\mathbf{c}}_l = (\mathbf{K} + n\lambda \mathbb{1}_{n \times n})^{-1} \mathbf{F}_l \in \mathbb{R}^n.$$

Using this, we obtain:

$$\hat{b}_l = \hat{\mathbf{c}}_l^\top (K(x_1, \cdot), \ldots, K(x_n, \cdot))^\top, \text{ for any } 1 \le l \le d, \text{ and thus: } \hat{h}_\omega = \hat{b}^\top \omega.$$

Now that we have obtained a closed-form expression for $\hat{h}_\omega$, we can express $\widehat{\mathcal{F}}(\omega)$ and $\widehat{\nabla \mathcal{F}}(\omega)$ in closed-form, as we will see next.

## I.2 Plug-in estimators for $\mathcal{F}(\omega)$ and $\nabla \mathcal{F}(\omega)$

Let $(\tilde{x}_j, \tilde{y}_j)_{1 \le j \le m}$ be $m$ samples drawn from $\mathbb{Q}$ and $\omega \in \mathbb{R}^d$. We have:

$$\widehat{\mathcal{F}}(\omega) = \frac{1}{2m} \sum_{j=1}^m \left( \hat{h}_\omega(\tilde{x}_j) - \tilde{y}_j \right)^2 = \frac{1}{2m} \left\| \widehat{\mathbf{B}} \omega - \tilde{\mathbf{y}} \right\|^2,$$

where $\widehat{\mathbf{B}} = [\hat{b}(\tilde{x}_1), \ldots, \hat{b}(\tilde{x}_m)]^\top \in \mathbb{R}^{m \times d}$ and $\tilde{\mathbf{y}} = (\tilde{y}_1, \ldots, \tilde{y}_m)^\top \in \mathbb{R}^m$. For any $1 \le l \le d$ and $1 \le j \le m$, we have:

$$\hat{b}_l(\tilde{x}_j) = [\mathbf{F}_l]^\top (\mathbf{K} + n\lambda \mathbb{1}_{n \times n})^{-1} \left[ \overline{\mathbf{K}}^\top \right]_j.$$

As a consequence, we obtain:

$$\hat{b}(\tilde{x}_j) = (\hat{b}_1(\tilde{x}_j), \ldots, \hat{b}_d(\tilde{x}_j))^\top = \widehat{\mathbf{C}}^\top \left[ \overline{\mathbf{K}}^\top \right]_j \in \mathbb{R}^d,$$

where $\widehat{\mathbf{C}} = [\hat{\mathbf{c}}_1, \ldots, \hat{\mathbf{c}}_d] = (\mathbf{K} + n\lambda \mathbb{1}_{n\times n})^{-1} \mathbf{F} \in \mathbb{R}^{n \times d}$, with $\mathbf{F} = [\mathbf{F}_1, \ldots, \mathbf{F}_d] \in \mathbb{R}^{n \times d}$. This implies that $\widehat{\mathbf{B}} = \overline{\mathbf{K}} \widehat{\mathbf{C}} \in \mathbb{R}^{m \times d}$, and hence:

$$\widehat{\mathcal{F}}(\omega) = \frac{1}{2m} \left\| \overline{\mathbf{K}} \widehat{\mathbf{C}} \omega - \tilde{\mathbf{y}} \right\|^2 = \frac{1}{2m} \omega^\top (\overline{\mathbf{K}} \widehat{\mathbf{C}})^\top \overline{\mathbf{K}} \widehat{\mathbf{C}} \omega - \frac{1}{m} \tilde{\mathbf{y}}^\top \overline{\mathbf{K}} \widehat{\mathbf{C}} \omega + \frac{1}{2m} \|\tilde{\mathbf{y}}\|^2. \quad (29)$$

On the other hand, using Appendix C, we get:

$$\widehat{\nabla \mathcal{F}}(\omega) = \frac{1}{m} \widehat{\mathbf{C}}^\top \overline{\mathbf{K}}^\top \left( \overline{\mathbf{K}} \widehat{\mathbf{C}} \omega - \tilde{\mathbf{y}} \right) = \frac{1}{m} \left[ (\overline{\mathbf{K}} \widehat{\mathbf{C}})^\top \overline{\mathbf{K}} \widehat{\mathbf{C}} \omega - (\overline{\mathbf{K}} \widehat{\mathbf{C}})^\top \tilde{\mathbf{y}} \right] \in \mathbb{R}^d. \quad (30)$$

The exact expressions of $\mathcal{F}(\omega)$ and $\nabla \mathcal{F}(\omega)$ involve expectations and are therefore intractable to compute analytically. A natural approach is to approximate these quantities using their plug-in estimators $\widehat{\mathcal{F}}(\omega)$ and $\widehat{\nabla \mathcal{F}}(\omega)$, evaluated with a very large number of inner and outer samples, $n$ and $m$. However, this approach quickly becomes computationally and memory-intensive. In particular, storing the kernel matrices $\mathbf{K}$ and $\overline{\mathbf{K}}$ requires $\mathcal{O}(n^2)$ and $\mathcal{O}(nm)$ space, respectively. Moreover, computing the inverse $(\mathbf{K} + n\lambda \mathbb{1}_{n\times n})^{-1}$ incurs a cubic time complexity of $\mathcal{O}(n^3)$, which is prohibitive for large-scale applications. To alleviate these computational bottlenecks, potential strategies rely on classical techniques in kernel methods such as Random Fourier Features (RFF), which approximate kernel functions in a finite-dimensional feature space and enable more efficient gradient computations [60], and Nyström approximations, which mitigate the computational burden of full kernel matrices by using a low-rank approximation of the kernel [75]. In our experiments, we leverage the closed-form expressions of the plug-in estimators, and replace the kernel evaluations with their approximations via RFF. This enables us to construct efficient and scalable approximations of $\mathcal{F}(\omega)$ and $\nabla \mathcal{F}(\omega)$, while significantly reducing both the memory usage and the computational cost. Our approach will be discussed in the following.

### I.3 Scalable approximations for $\mathcal{F}(\omega)$ and $\nabla \mathcal{F}(\omega)$ via random Fourier features

Random Fourier Features (RFF) provide a way to approximate shift-invariant kernels (*i.e.*, kernels that satisfy $K(x, x') = G(x - x')$ for some function $G : \mathcal{X} \to \mathbb{R}$) by mapping the data into a *randomized* feature space. To avoid the high computational burden of building the full kernel matrix from all pairwise kernel evaluations, RFF use a randomized feature map $\psi : \mathcal{X} \to \mathbb{R}^D$, with $D$ being the number of Fourier features, to approximate the kernel as follows:

$$K(x, x') \approx \psi(x)^\top \psi(x'), \quad \text{for any } x, x' \in \mathcal{X}.$$

Now, we derive the expression of the feature map $\psi$. Let $x, x' \in \mathcal{X}$. By Bochner theorem [16], we have that:

$$K(x, x') = \frac{1}{(2\pi)^p} \mathbb{E}_{\mathbf{w} \sim \widehat{G}(\mathbf{w})} \left[ e^{i \mathbf{w}^\top (x - x')} \right] = \frac{1}{(2\pi)^p} \mathbb{E}_{\mathbf{w} \sim \widehat{G}(\mathbf{w})} \left[ \cos(\mathbf{w}^\top (x - x')) \right], \quad (31)$$

where $\widehat{G}$ is the Fourier transform of $G$. For any $b \in \mathbb{R}$, the following product-to-sum identity holds:

$$2\cos(\mathbf{w}^\top x + b) \cos(\mathbf{w}^\top x' + b) = \cos(2b + \mathbf{w}^\top (x + x')) + \cos(\mathbf{w}^\top (x - x')).$$

In particular, when $b \sim \mathcal{U}(0, 2\pi)$ (the uniform distribution over $[0, 2\pi]$), we get:

$$\mathbb{E}_{b \sim \mathcal{U}(0, 2\pi)} \left[ 2\cos(\mathbf{w}^\top x + b) \cos(\mathbf{w}^\top x' + b) \right] = \mathbb{E}_{b \sim \mathcal{U}(0, 2\pi)} \left[ \cos(2b + \mathbf{w}^\top (x + x')) \right] \\ + \cos(\mathbf{w}^\top (x - x')).$$

However, we have:

$$\mathbb{E}_{b \sim \mathcal{U}(0, 2\pi)} \left[ \cos(2b + \mathbf{w}^\top (x + x')) \right] = \frac{1}{2\pi} \int_0^{2\pi} \cos(2b + \mathbf{w}^\top (x + x')) \, \mathrm{d}b$$

$$= \frac{1}{4\pi} [\sin(2b + \mathbf{w}^\top (x + x'))]_{b=0}^{b=2\pi} = 0.$$

Thus:
$$\mathbb{E}_{b\sim\mathcal{U}(0,2\pi)}\left[2\cos(\mathbf{w}^\top x + b)\cos(\mathbf{w}^\top x' + b)\right] = \cos(\mathbf{w}^\top(x - x')).$$
Substituting this back into Equation (31), we arrive at:
$$K(x, x') = \mathbb{E}_{\mathbf{w}\sim\frac{1}{(2\pi)^p}\widehat{G}(\mathbf{w}),\, b\sim\mathcal{U}(0,2\pi)}\left[\sqrt{2}\cos(\mathbf{w}^\top x + b)\sqrt{2}\cos(\mathbf{w}^\top x' + b)\right].$$
Using $D$ samples $\mathbf{w}_1, \ldots, \mathbf{w}_D \sim \frac{1}{(2\pi)^p}\widehat{G}(\mathbf{w})$ and $b_1, \ldots, b_D \sim \mathcal{U}(0, 2\pi)$, we obtain by Monte Carlo estimation:
$$K(x, x') \approx \sum_{i=1}^{D}\left(\sqrt{\frac{2}{D}}\cos(\mathbf{w}_i^\top x + b_i)\right)\left(\sqrt{\frac{2}{D}}\cos(\mathbf{w}_i^\top x' + b_i)\right).$$
This implies that:
$$\psi(x) = \sqrt{\frac{2}{D}}\cos(\mathbf{W}\,x + b), \text{ where } \mathbf{W} = (\mathbf{w}_1, \ldots, \mathbf{w}_D)^\top \in \mathbb{R}^{D\times p} \text{ and } b = (b_1, \ldots, b_D)^\top \in \mathbb{R}^D.$$
In practice, one typically chooses $D \ll n$ and $D \ll m$, which reduces the space complexity of storing $\mathbf{K}$ from $\mathcal{O}(n^2)$ to $\mathcal{O}(nD)$, and that of storing $\overline{\mathbf{K}}$ from $\mathcal{O}(nm)$ to $\mathcal{O}((n+m)D)$. This results in significant computational and memory savings. Using the RFF approach, the two kernel matrices $\mathbf{K}$ and $\overline{\mathbf{K}}$ can then be approximated as:
$$\mathbf{K} \approx \Xi\Xi^\top \text{ and } \overline{\mathbf{K}} \approx \widetilde{\Xi}\Xi^\top,$$
where $\Xi = [\psi(x_1), \ldots, \psi(x_n)]^\top \in \mathbb{R}^{n\times D}$ and $\widetilde{\Xi} = [\psi(\tilde{x}_1), \ldots, \psi(\tilde{x}_m)]^\top \in \mathbb{R}^{m\times D}$. A common term in Equations (29) and (30) is $\overline{\mathbf{K}}\,\widehat{\mathbf{C}}$, which can be approximated using the push-through identity as follows:
$$\overline{\mathbf{K}}\,\widehat{\mathbf{C}} \approx \widetilde{\Xi}\Xi^\top\left(\Xi\Xi^\top + n\lambda\mathbb{1}_{n\times n}\right)^{-1}\mathbf{F} = \widetilde{\Xi}\left(\Xi^\top\Xi + n\lambda\mathbb{1}_{D\times D}\right)^{-1}\Xi^\top\mathbf{F} \in \mathbb{R}^{m\times d}.$$
Here, instead of inverting a matrix of size $n \times n$, we invert a matrix of size $D \times D$, which leads to significant computational savings in time, especially when $D \ll n$. Consequently, using this approximation, we get:
$$\widehat{\mathcal{F}}(\omega) \approx \frac{1}{2m}\omega^\top\mathbf{J}^\top\widetilde{\Xi}^\top\widetilde{\Xi}\,\mathbf{J}\,\omega - \frac{1}{m}\omega^\top\mathbf{J}^\top\widetilde{\Xi}^\top\tilde{\mathbf{y}} + \frac{1}{2m}\|\tilde{\mathbf{y}}\|^2,$$
$$\widehat{\nabla\mathcal{F}}(\omega) \approx \frac{1}{m}\left[\mathbf{J}^\top\widetilde{\Xi}^\top\widetilde{\Xi}\,\mathbf{J}\,\omega - \mathbf{J}^\top\widetilde{\Xi}^\top\tilde{\mathbf{y}}\right],$$
where $\mathbf{J} = \left(\Xi^\top\Xi + n\lambda\mathbb{1}_{D\times D}\right)^{-1}\Xi^\top\mathbf{F} \in \mathbb{R}^{D\times d}$. As mentioned earlier, a very large number of samples $n$ and $m$ is required to obtain accurate approximations of $\mathcal{F}(\omega)$ and $\nabla\mathcal{F}(\omega)$ using the RFF approach. To cope with the issue of storing the two matrices $\Xi \in \mathbb{R}^{n\times D}$ and $\widetilde{\Xi} \in \mathbb{R}^{m\times D}$ in memory, we implement this method in blocks. More precisely, we divide our data $(x_i, t_i)_{1\le i\le n}$ and $(\tilde{x}_j, \tilde{y}_j)_{1\le j\le m}$ into blocks, then compute $\Xi^\top\Xi$, $\widetilde{\Xi}^\top\widetilde{\Xi}$, $\Xi^\top\mathbf{F}$, $\widetilde{\Xi}^\top\tilde{\mathbf{y}}$, and $\|\tilde{\mathbf{y}}\|^2$ as follows:

$$\Xi^\top\Xi = \sum_{i=1}^{n}\psi(x_i)\psi(x_i)^\top = \sum_{B\in\mathcal{B}}\sum_{x\in B}\psi(x)\psi(x)^\top = \sum_{B\in\mathcal{B}}\Xi_B^\top\Xi_B \in \mathbb{R}^{D\times D},$$

$$\widetilde{\Xi}^\top\widetilde{\Xi} = \sum_{j=1}^{m}\psi(\tilde{x}_j)\psi(\tilde{x}_j)^\top = \sum_{B\in\mathcal{B}}\sum_{\tilde{x}\in B}\psi(\tilde{x})\psi(\tilde{x})^\top = \sum_{B\in\mathcal{B}}\widetilde{\Xi}_B^\top\widetilde{\Xi}_B \in \mathbb{R}^{D\times D},$$

$$\Xi^\top\mathbf{F} = \sum_{i=1}^{n}\psi(x_i)\left[\phi_1(t_i), \ldots, \phi_d(t_i)\right] = \sum_{B\in\mathcal{B}}\sum_{(x,t)\in B}\psi(x)\left[\phi_1(t), \ldots, \phi_d(t)\right] = \sum_{B\in\mathcal{B}}\Xi_B^\top\mathbf{F}_B \in \mathbb{R}^{D\times d},$$

$$\widetilde{\Xi}^\top\tilde{\mathbf{y}} = \sum_{j=1}^{m}\psi(\tilde{x}_j)\tilde{y}_j = \sum_{B\in\mathcal{B}}\sum_{(\tilde{x},\tilde{y})\in B}\psi(\tilde{x})\tilde{y} = \sum_{B\in\mathcal{B}}\widetilde{\Xi}_B^\top\tilde{\mathbf{y}}_B \in \mathbb{R}^{D},$$

$$\|\tilde{\mathbf{y}}\|^2 = \sum_{B\in\mathcal{B}}\|\tilde{\mathbf{y}}\|_B^2,$$

where $\mathcal{B}$ denotes the set of blocks, and the subscript $B$ indicates that the corresponding quantity is computed using only the data contained in block $B$. These block-wise computations make it possible to precisely approximate $\widehat{\mathcal{F}}(\omega)$ and $\widehat{\nabla\mathcal{F}}(\omega)$ in a scalable manner. As a result, we can efficiently approximate both $\mathcal{F}(\omega)$ and $\nabla\mathcal{F}(\omega)$ through their plug-in estimators when choosing large sample sizes $n$ and $m$.

## I.4 Additional details on the experimental setup

We use the JAX framework [18] to run our experiments on an NVIDIA RTX 6000 ADA GPU. The experiments take approximately 15 hours to complete.

**Choice of the kernel.** In our experiments, we consider the Gaussian kernel defined, for any $x, x' \in \mathcal{X}$, as $K(x, x') = e^{-\frac{\|x-x'\|^2}{2\sigma^2}}$, where $\sigma > 0$ is the bandwidth parameter controlling the smoothness. Since the Gaussian kernel is translation-invariant, Bochner's theorem is applicable. In this case, using the same notations as in Appendix I.3, we have $G(z) = e^{-\frac{\|z\|^2}{2\sigma^2}}$, for any $z \in \mathcal{X}$. Its Fourier transform $\widehat{G}$ is then given by $\widehat{G}(\mathbf{w}) = (2\pi\sigma^2)^{\frac{p}{2}} e^{-\frac{\sigma^2 \|\mathbf{w}\|^2}{2}}$, for any $\mathbf{w} \in \mathbb{R}^d$. As a consequence, we obtain:

$$\frac{1}{(2\pi)^p}\widehat{G}(\mathbf{w}) = \frac{1}{(2\pi)^p}(2\pi\sigma^2)^{\frac{p}{2}} e^{-\frac{\sigma^2 \|\mathbf{w}\|^2}{2}} = \left(\frac{2\pi}{\sigma^2}\right)^{-\frac{p}{2}} e^{-\frac{\sigma^2 \|\mathbf{w}\|^2}{2}} = \mathcal{N}\left(0, \frac{1}{\sigma^2}\mathbb{1}_{p \times p}\right),$$

which implies that $\mathbf{w}_1, \ldots, \mathbf{w}_D \sim \mathcal{N}(0, \frac{1}{\sigma^2}\mathbb{1}_{p \times p})$.

**Choice of the statistical model and hyperparameters.** We set $p = 3$, $d = 4$, $\lambda = 0.01$, and $\sigma = 0.2$. We generate synthetic data as follows:

$$x \sim P_x, \quad t = 2(\mathbb{1}_p^\top x + \epsilon), \quad y = \omega^{\star\top}\phi(t) + \epsilon,$$

where $\epsilon \sim \mathcal{N}(0, 0.025)$, $\omega^\star \sim \mathcal{U}(0, 1)^d$, and $\phi(t) = (\sin(t+1), \ldots, \sin(t+d))^\top$. We consider two cases for the distribution $P_x$ of the instrumental variable $x$: (i) a $p$-dimensional standard Gaussian, i.e., $P_x = \mathcal{N}(0, \mathbb{1}_{p \times p})$, and (ii) a $p$-dimensional Student's $t$-distribution with degrees of freedom $\nu \in \{2.1, 2.5, 2.9\}$. All random variables are fixed across runs for reproducibility.

## I.5 Additional experimental results

Here, we retain the same experimental setup as in the main paper and extend the analysis by providing additional experimental results in the scenario where both $m$ and $n$ vary simultaneously over the range 100 to 5000. In Figure 3, we visualize the results using heatmaps for four key quantities: $|\mathcal{F}(\omega_0) - \widehat{\mathcal{F}}(\omega_0)|$, $\|\nabla\mathcal{F}(\omega_0) - \widehat{\nabla\mathcal{F}}(\omega_0)\|$, $\|\nabla\mathcal{F}(\omega_T)\|$, and $\min_{i=0,\ldots,T} \|\nabla\mathcal{F}(\omega_i)\|$, with $n$ on the $x$-axis and $m$ on the $y$-axis. We report the results only for the case where the instrumental variable $x$ is sampled from a $p$-dimensional standard Gaussian, since the cases where $x$ is sampled from a $p$-dimensional Student's $t$-distribution with degrees of freedom $\nu \in \{2.1, 2.5, 2.9\}$ exhibit similar trends. From the heatmaps, we observe that the lowest errors across all four metrics occur along the diagonal of the plots, *i.e.*, when $m = n$. This pattern suggests that matching the number of samples in the two dimensions leads to more accurate estimation of both the objective function and its gradient, as well as improved convergence behavior during optimization.

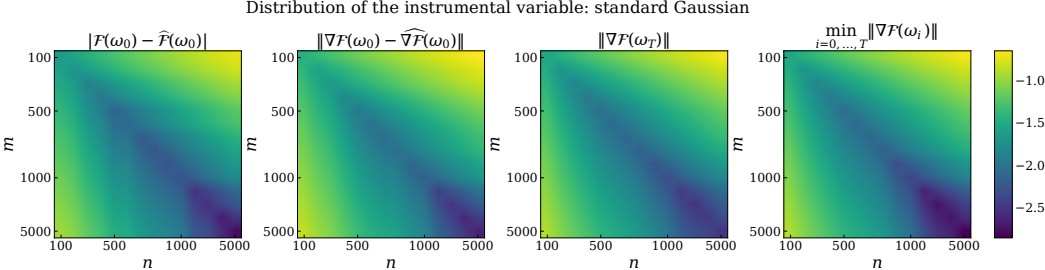

Figure 3: Illustration of gradient descent on (KBO) for the instrumental variable regression task using synthetic data, with an instrumental variable sampled from a standard Gaussian distribution. The logs of the means of the four quantities across 50 runs are displayed.

