# OpenReview forum: "Learning Theory for Kernel Bilevel Optimization"
_NeurIPS.cc/2025/Conference — NeurIPS 2025 poster_

### Official Review · Reviewer_11HJ · 2025-06-08

**Clarity:** 3
**Significance:** 3
**Originality:** 3
**Rating:** 4
**Confidence:** 4

**Summary:**

The paper considers bilevel optimization in the rkhs. The algorithm  involves using grading method like in the parametric case.

**Questions:**

1. what is the essential challeng of doing kernel version?
2. is cobvergence rate optimal?

**Ethical Concerns:**

["NO or VERY MINOR ethics concerns only"]

**Limitations:**

yes

**Quality:**

3

**Strengths And Weaknesses:**

The introduction is well written, showing the importance and challenge of the problem.  However, the analysis is not quite different from the parametric case. One just need to carefully treat the concept of gradient in function space. The empirical process technique used is also just the same as the parametric case.  The result showing that both the function value and its gradient has some uniform convergence property, which is expected. Convergence of the gradient algorithm is also standard. Thus I'm not too enthusiastic about the work. The problem itself is important, and this is probably a reasonable start.

---

> ### Author Rebuttal · Authors · 2025-07-30
>
> Dear Reviewer,
>
> Thank you very much for your review. We are glad to hear that you found the paper clear and that it addresses an important problem. We appreciate your feedback and understand your concerns, which we address in detail below. Please feel free to reach out with any further questions.
>
> ### **1- Main challenge in KBO: Beyond empirical processes**
> Our analysis required considering objects that are not standard empirical processes. The functional setting involves processes that take values in an **infinite-dimensional space** (as opposed to real-valued processes). Hence, we cannot apply classical empirical process theory to those: one could apply the classical result to get an error per dimension, but then summing the errors yields a divergent sum. Instead, we had to use results on $U$-processes (Step 3 in Section 4.2), which, to our knowledge, **have not been used** in the analysis of finite-dimensional problems. Considering these $U$-processes was key for obtaining the $1/\sqrt{n}$ and $1/\sqrt{m}$ rates. As discussed in the last paragraph of Section 4.2, using other "simpler" techniques results in slower rates. We will clarify this technical novelty in the introduction and throughout the text.
>
> ### **2- Convergence rates of (projected) gradient algorithms**
> While our convergence rates resemble those of classical gradient algorithms, their derivation in our setting is far from standard due to the functional nature and statistical dependencies specific to the bilevel kernel framework. In particular, the parameter updates do not rely on the exact gradient, but rather on a **data-dependent** gradient estimator arising from the empirical bilevel problem $\widehat{\text{KBO}}$. This introduces two main sources of errors: (1) a **statistical error**, which stems from approximating the population gradient and depends on the sample sizes, (2) an **optimization error**, which arises from performing a finite number of (projected) gradient descent steps.
>
> ### **3- Optimality of the convergence rates**
> We did not investigate the optimality of the convergence rates, as the primary goal of our paper is to provide the first finite-sample generalization bounds for nonparametric bilevel optimization. We agree that studying the optimality of these rates is an interesting future research direction and plan to include this point in the conclusion of the revised version.
>
> Best regards,
>
> Authors

---

> > ### Comment · Reviewer_11HJ · 2025-08-02
> >
> > Thanks for your response, I have read through your responses and I keep the same opinion about your work.

---

### Official Review · Reviewer_ELKB · 2025-06-20

**Clarity:** 3
**Significance:** 2
**Originality:** 3
**Rating:** 5
**Confidence:** 4

**Summary:**

The paper proposes the formalism of kernel bilevel optimization, a way to frame a specific instance of a recently proposed functional bilevel optimization setting in terms of reproducing kernel Hilbert spaces (RKHSs) - thereby making it nonparametric. The inner optimization problem is given by a classical ridge regression / Tikhonov regularized objective with a generic loss function, which is often found in the kernel literature and allows for the application of a representer theorem.

A gradient-based plugin estimator is investigated. The theoretical core contributions are worst-case inequalities for the distances between the empirical and analytical versions of the objective as well as its gradient in expectation. The regularity assumptions are typical and relatively mild. The bounds exhibit an additive dependence on a split sample with 1/sqrt in both sample sizes each. They also exhibit a generic multiplicative constant which is briefly explained in the main text and in a more detailed form throughout the appendix. For example, a typical diameter-type dependence on the compact parameter domain (outer objective) as well as a regularization parameter dependence (inner objective) are the two most obvious properties reflected in the constant.

The authors propose a gradient scheme and give generalization bounds in expectation, showing a similar sample dependence as before as well as an additional 1/sqrt dependence on the iteration number under additional positive definiteness assumptions. The structure and assumptions of this bound seem to be classical within the context of first-order optimization theory. Finally, a very basic synthetic experiment for the specific instance of kernel instrumental variable regression framework as proposed by Sing et al. (2019) is performed, supposedly confirming the theoretical findings.

**Questions:**

I suggest the authors address the following two points:

1) Using the additional space in order to expand the remarks and discussions around the constants in the main text and add some additional insight about how they depend on parameters related to the problem and the gradient scheme. In particular, I believe the interplay between regularization theory in the inner objective and optimization theory in the outer objective should be interesting to the reader. Furthermore, expanding the discussion of the constants might allow to highlight connections to available results in a more direct way. Note that Corollary 4.2 could specifically benefit from additional remarks.

2) Discussing available results for the applications they list as motivation in order to consider KBO. In particular, how do these results relate to the new findings in the new, more general KBO setting? In particular, for kernel instrumental variable regression as given by [1], the two stage estimator allows the direct application of high-probability lower and optimal upper bounds for kernel ridge regression with vector-valued kernels [2], leading to minimax optimality for the instrumental variable procedure [3]. It seems reasonable to believe that similar arguments can be made for the inner objective of KBO under suitable assumptions about loss and noise, potentially leading to a fundamental lower bound. It might also be insightful to briefly mention relevant classical upper and lower bounds for general gradient schemes in the context of the outer objective in the main text.

A minor technical point:

Measurability of the kernel needs to formally be required in order to ensure that expectations of functions and embedded variables in the RKHS are well-defined (e.g. [4, Chapter 4]). I think this could be added to the formal assumptions, in particular since the rest of the paper is written quite rigorously.

There is another very general question which is outside of the scope of the paper. I wondered the following:

Two-stage kernel instrumental variable regression as given by [1] uses a vector-valued ridge estimate in *both* stages.
The work [3] proposes to use a general spectral algorithm [5] for the *first* stage.
It is known that general spectral algorithms include iterative schemes such as gradient descent. If, instead of in the first stage, one uses a general spectral algorithm in the *second stage*, this might lead to a very similar (or even the equivalent) estimator that the present paper proposes in the special instance of instrumental variable regression and could lead to optimal rates by repeating the arguments by [3]. Incorporating loss calibration results [4] could allow to transfer these rates to more general losses.
Maybe the authors also have some better intuition or opinions about this.

[1] Singh et. al. Kernel instrumental variable regression. NeurIPS 2019.

[2] Li et al. Towards Optimal Sobolev Norm Rates for the Vector-Valued Regularized Least-Squares Algorithm. JMLR 2024.

[3] Meunier et al. Nonparametric Instrumental Regression via Kernel Methods is Minimax Optimal. arXiv:2411.19653 2024.

[4] Steinwart & Christmann. Support Vector Machines. 2008.

[5] Meunier et al. Optimal rates for vector-valued spectral regularization learning algorithms. NeurIPS 2024.

**Ethical Concerns:**

["NO or VERY MINOR ethics concerns only"]

**Final Justification:**

The authors have addressed all points and I maintain my rating.

**Limitations:**

yes

**Paper Formatting Concerns:**

no concerns

**Quality:**

3

**Strengths And Weaknesses:**

I see the main contribution of the paper in the abstract framing of nonparametric bilevel optimization with generic loss functions and the first steps towards analyzing KBO theoretically, which this paper manages to do well.

The derived error bounds are of relatively simple type (analytical regularized loss vs. empirical regularized loss, no ground truth assumptions and corresponding convergence theory for $\lambda \to 0$, only expectation bounds, no high probability bounds under noise assumptions, no fast rates, no lower bounds), but sufficiently outline core properties of the proposed estimator under relatively weak assumptions as a first step towards a theory of KBO. I do think that the main paper could benefit from a bit more insight that the appendix provides, in particular with regard to the structure of the multiplicative constants in the bounds. This could make connections to related theory more clear. This will also open up additional avenues for further investigation, which I believe are plentiful.

Nevertheless, I believe the contribution should be sufficient for the paper to be accepted, as this can be addressed during the rebuttal phase by adding suitable comments and using the additional space.

**Strengths**

To my understanding, this is the first time that kernel-based bilevel optimization is discussed in this form of abstraction.

Generally good presentation and accessibility; newcomers to the field should be able to read and understand the basics of this work without significant issues.

Appropriate degree of mathematical rigor, formalism and attention to detail in technical writing.

The generality of the proposed formalism will allow more targeted investigations based on known theory and techniques from adjacent fields and allow more detailed analyses.

**Weaknesses**

The paper lists two applications as motivation in order to investigate general KBO, in particular distribution shift and instrumental variable regression—the latter is specifically investigated in the experiment. However, the authors do not compare their theoretical results and their proposed gradient scheme to existing results for those applications (to which they are inherently related). It is left unclear when (and how) their results extend these applications and/or give new insights.

I also believe that the main paper hides a lot of theoretical insight in the generic multiplicative constants, which are given in full detail the appendix. In particular, connections to regularization theory and vector-valued regression (inner objective) and general optimization (outer objective) will become more obvious and make the results comparable to related work in those fields; see “Questions” section.

---

> ### Author Rebuttal · Authors · 2025-07-30
>
> Dear Reviewer,
>
> Thank you for your detailed review and thoughtful comments. We are pleased that you found our paper strong and appreciate the constructive suggestions to improve it. Below, we provide detailed responses to your remarks, and we would be happy to address any further questions you may have, should any of our responses require clarification.
>
> ### **1- Discussion of the constants**
> We agree that a more explicit expression would be beneficial. In our case, the most interesting dependence is w.r.t. $\lambda$ as it (1) controls the strong convexity of the inner objective and the smoothness of the outer objective (see Proposition C.4 of Appendix C.2), crucial for optimization, and (2) allows providing generalization for the un-regularized problem ($\lambda= 0$) by controlling the behavior of such constant as $\lambda$ approaches 0, provided additional source assumptions are made (e.g., Smale & Zhou, 2007; Sriperumbudur et al., 2017). Under our current relatively weak assumptions, we are not able to characterize the exact dependence of the constants on $\lambda$, especially to control the behavior of the constant as $\lambda$ decreases to 0: parts of the terms in the constant have only a qualitative dependence on $\lambda$. Providing more quantitative estimates would require stronger regularity assumptions such as Lipschitzness of some of the derivatives. Our choice of assumptions is partly motivated by the simplicity of the resulting proofs; however, it is possible to adapt the proofs to get more quantitative dependence on $\lambda$ and provide generalization errors for un-regularized problems, which would be a great future work direction. We will still add a discussion on the constants and, in particular, discuss how the regularization parameter affects both generalization and optimization: large $\lambda$ yields smoother problems that are faster to optimize using large step sizes, but introduces a large bias in the statistical estimation. When $\lambda$ is a parameter to select, a trade-off that accounts for the bias (un-regularized vs regularized problems), variance (finite sample vs population as done in this study) and optimization (step size) must be found.
>
> ### **2- Remarks on the gradient scheme**
> Corollary 4.2 informs practical algorithm design choices. Indeed, it underscores that the convergence of the bilevel algorithm requires a careful balance between data availability (sample sizes) and computational budget (number of gradient steps).  We will make sure to discuss that right after Corollary 4.2.
>
> ### **3- Connection with existing results in IV**
> Thank you for raising this point, we will include a discussion on that based on the following. Indeed, our problem formulation is related to IV, which is a particular case, with the important difference that our setting currently only considers regularized inner problems (fixed $\lambda$, independent of sample size), while typical studies on IV consider un-regularized population problems. It is more challenging to study the latter, and existing literature on the topic made important effort in that direction, obtaining minimax optimal rates depending on source assumptions, as mentioned in your questions. The challenge in our case, lies in an orthogonal direction: by considering more general inner objectives, even the study of generalization for regularized problems presents additional obstacles that have not been addressed before (see also the answer to the general question). A next natural step would be to study un-regularized problems, after which a more direct comparison with existing results on IV could be made. This is unfortunately beyond the scope of this work.
>
> ### **4- Can existing mimimax results on IV be used for KBO?**
> It would be great to obtain minimax rates for KBO. However, it is unclear to us how the techniques used in the context of IV can be used for the KBO. Results for IV heavily rely on the closed-form expression of the first stage, while KBO assumes general loss structure for the inner problem (no closed-form of the minimizer in general). We believe the non-linear dependence of the inner solution on the outer parameter, in the general case, is precisely what makes it hard to apply existing results on IV and two stage regression (see also the response to the general question).
>
> ### **5- Measurability of the kernel**
> We agree that explicitly including this assumption would make the exposition more rigorous and self-contained. We will add a corresponding measurability condition to our list of formal assumptions in the revised version.
>
> ### **6- General question**
> Thank you for this insightful question. We will add a discussion based on the following points.
>
> - **Spectral techniques.** The idea of spectral filtering is powerful as it allows treating several algorithms in a unified manner as illustrated in (Meunier et al., 2024a). To our knowledge, using the results on spectral filtering for the outer loss suggests that such loss must be a quadratic loss in $\omega$. This is true when using a regression loss and if the inner solution has a linear dependence in the outer parameter. Such linearity arises when the inner objective is also a regression loss, as in KIV, but is no longer true in general: using general losses in the inner level, yields a non-linear dependence on the outer parameter, that is even implicit. Because of this, it is unclear to us at this stage how to directly transfer the minimax results in (Meunier et al., 2024b) to our setting.
>
> - **Surrogate loss and calibration.** The approach based on surrogate loss and calibration, suggests two directions:
>   - Using a regression loss as surrogate for either both or one of the losses. However, this would imply using the estimators arising from the use of such calibrated regression losses. It is unclear to us how to construct computable calibrated losses in the form of a quadratic loss from the general losses we considered. Additional assumptions on the structure of these losses would be needed.
>   - The other direction is to treat the general losses as surrogate for the quadratic losses, but this means one has to derive statistical bounds for these general losses (which is what the current work does), and then transfer those to control quadratic errors.
>
> Overall, it remains unclear to us how to directly leverage the results in the case of  KIV on KBO without restriction on the form of the loss to be essentially quadratic.
>
> Best regards,
>
> Authors
>
> ---
>
> **References**
>
> (Smale & Zhou, 2007) S. Smale and D.-X. Zhou. Learning theory estimates via integral operators and their approximations. Constructive Approximation 2007.
>
> (Sriperumbudur et al., 2017) B. Sriperumbudur, K. Fukumizu, A. Gretton, A. Hyvärinen, and R. Kumar. Density estimation in infinite dimensional exponential families. JMLR 2017.
>
> (Meunier et al., 2024a) D. Meunier, Z. Shen, M. Mollenhauer, A. Gretton, and Z. Li. Optimal rates for vector-valued spectral regularization learning algorithms. NeurIPS 2024.
>
> (Meunier et al., 2024b) D. Meunier, Z. Li, T. Christensen, and A. Gretton. Nonparametric Instrumental Regression via Kernel Methods is Minimax Optimal. arXiv:2411.19653 2024.

---

> > ### Comment · Reviewer_ELKB · 2025-08-02
> >
> > Thank you for your insightful response and clarifications. I will keep my score.

---

### Official Review · Reviewer_cVHc · 2025-07-01

**Clarity:** 3
**Significance:** 4
**Originality:** 4
**Rating:** 5
**Confidence:** 3

**Summary:**

The paper studies generalization properties of Kernel Bilevel Optimization (KBO). This task comes with many additional challenges compared to existing work (e.g. the parameter space growing with sample size for kernel methods), which the authors solve with several technical contributions.

The first contribution is showing that differentiation and plug-in emprical estimation of gradient derived via implicit differentiation commute under specified assumptions. Secondly, based on the commutative property the authors derive generalization bounds in Theorem 4.1 using the maximal inequalities for degenerate U-processes of order 2. These two results form the core of the paper and the intuition behind the proofs is well described. There is an additional result for gradient descent solver of KBO. Numerical experiments match the the theoretical bounds asymptotically when the sample size of the inner and outer loops are the same (m=n), although the scaling in terms of $m\neq n$ is not explored numerically.

**Questions:**

* In the first block equation in 2.2, is the distribution of $x,y$ in the expectation different, i.e., that one of them is on train and the other on the test data? I think it would be useful to explicitly highlight in the equation.
* In the experiments you show an experiment when $m = n$ which does not demonstrate the theoretical scaling of $1/sqrt(n) + 1/sqrt(m)$. Could you vary $m$ and $n$ to show the scaling in both dimensions? This might reveal that the upper bounds are not tight in this regime, but it would be still very important to know.
* L213: Claim that [6, 61] do not hold for parameter dimension growing with $n$, thus not for kernel methods. Is it possible to make some comparison for the asymptotic scaling in m or n?

**Ethical Concerns:**

["NO or VERY MINOR ethics concerns only"]

**Final Justification:**

I am very happy with the paper and the way the authors addressed my questions. I decided to keep the scores.

**Limitations:**

Yes

**Quality:**

4

**Strengths And Weaknesses:**

Overall, the paper in my opinion makes significant contribution and is written with great detail (although I checked the appendices only very quickly).

For weaknesses, I have mostly minor suggestions hoping to help with clarity for other readers and a remark, that I would like to see the experimental results when $m\neq n$, perhaps as a figure in the appendices.

Suggestions for clarity:
This is a long paper and I imagine that it was difficult to fit it into the page requirement.
* If I understand correctly Sec 2.3. is about the difficulty to performing implicit differentiation (which is needed for KBO) for a kernel method. If this is the main point of Sec 2.3. and Prop 2.1, I think the clarity of the text would be improved if this was briefly stated at the beginning to motivate the section. Generally, I would like to better understand the purpose of Prop 2.1 (I guess it is later explained as motivating the plug-in estimate in 3.2), and also to have explained the intuition how it gets around the intractibility of the block equation above.
* L156: Can you specify which of the operator is abstract and intractable? My guess it is the $\nabla_h$, but it would improve clarity to be precise.
* Literature review: I find the literature review very short, which might be also due to the page limit restrictions. Nevertheless, I want to ask the authors to extend this and perhaps include a short section in the appendices that gives:
    * A short paragraph of few sentences on computational aspects of solving KBO and existing approaches.
    * A few sentences about the existing generalization results for Bilevel optimization. There is a paragraph in L207 saying why the existing results don’t hold for kernel bilevel optimization, but it would be useful to have some information about the scaling of the existing bounds for the non-kernel case.

L224: “and show that it recovers the previously introduced estimator $\nabla \hat{F}(\omega)$”, this is technically not accurate as only $\hat{F}(\omega)$ was introduced previously and its gradient is firstly written right below.

---

> ### Author Rebuttal · Authors · 2025-07-30
>
> Dear Reviewer,
>
> Thank you for your thoughtful review of our paper and for your valuable suggestions for improvement. We answer your concerns below and are happy to clarify any further concerns you may have.
>
> ### **1- Clarifying the purpose of Section 2.3**
> Thank you for this helpful comment. As suggested, we will explain at the beginning of the section the following idea: Section 2.3 adresses the challenge of performing implicit differentiation for KBO. This first requires deriving an abstract, a priori intractable, expression for the implicit gradient, from which we derive a more concrete expression in Proposition 2.1. By concrete we mean that it requires solving a regression problem in the RKHS (the adjoint problem), which can be approximately solved using finite samples, and plug it into an expectation (which can be also approximated using finite samples). Hence, Proposition 2.1 lays the ground for defining a plug-in estimate, later used in our statistical analysis, and which allows getting around the intractability of the implicit gradient.
>
> ### **2- Abstract operators in L156**
> Thank you for noticing this point, the phrasing is indeed vague. The operators we meant include $\partial_h$, but also $\partial_h^2$ and $\partial_{\omega, h}^2$, which appear when replacing the Jacobian $\partial_\omega h^\star_\omega$ by its expression in Equation 1. We will make sure to be precise about which operators we had in mind in the revised version.
>
> ### **3- Literature review**
> Thank you for the suggestion. We will add the following in the appendix of the revised version of our paper, which also answers your question about L213.
> - **Computational aspects of KBO.** The computational complexity of KBO algorithms typically scales cubically in sample size as for many kernel methods. For example, in the case of kernel instrumental variable regression (Singh et al., 2019), solving the inner level system costs $\mathcal{O}(n^3)$, and performing vector matrix products operations costs $\mathcal{O}(mnd)$, leading to a total complexity of $\mathcal{O}(n^3+mnd)$. To improve scalability, potential strategies rely on classical techniques in kernel methods such as **Random Fourier Features**, which approximate kernel functions in a finite-dimensional feature space and enable more efficient gradient computations (Rahimi & Recht, 2007), and **Nyström approximations**, which mitigate the computational burden of full kernel matrices by using a low-rank approximation of the kernel (Williams & Seeger, 2001).
>
> - **Existing bounds for the non-kernel case.** Bao et al. (2021) laid foundational work in this direction by analyzing uniform stability in full-batch bilevel optimization. Their criterion for generalization compares the population outer loss evaluated at the output of a randomized algorithm to the empirical outer loss evaluated using the same algorithm. Given a number $\kappa$ between 0 and 1, they obtain a decay at a rate of $\mathcal{O}(T^\kappa / m)$ for unrolled optimization, which decreases as $1/m$ in outer sample size, but increases with the number of outer iterations made. The generalization criterion is unlike the one we consider and which compares the population outer objective at the theoretically optimal inner solution $h^\star_\omega$ to the empirical loss evaluated at the empirical solution $\hat{h}_\omega$. Complementing this upper bound, Wang et al. (2024) established lower bounds on the uniform stability of gradient-based bilevel algorithms, demonstrating a rate of $\Omega(1/m)$. Building on (Bao et al., 2021), Zhang et al. (2024) extended the analysis to stochastic bilevel optimization, establishing on-average stability bounds and deriving a generalization rate of $\mathcal{O}(1/\sqrt{m})$ due to the presence of stochastic gradients. In a related context, Oymak et al. (2021) studied generalization in neural architecture search using a bilevel formulation, showing that approximate inner solutions and Lipschitz continuity of the outer loss yield a generalization bound of $\mathcal{O}(1/\sqrt{n}+1/\sqrt{m})$. Arora et al. (2020) investigated representation learning for imitation learning via bilevel optimization, offering generalization bounds of order $\mathcal{O}(1/\sqrt{m})$ that depend both on the size of the dataset and the stability of learned representations.
>
> ### **4- Clarification on $\nabla \widehat{\mathcal{F}}(\omega)$**
> We apologize for the confusion at L224 and thank you for pointing this out. We considered two estimators: $\nabla \widehat{\mathcal{F}}(\omega)$ (first defined in L199 in the text as the gradient of $\widehat{\mathcal{F}}(\omega)$) and $\widehat{\nabla\mathcal{F}}(\omega)$ which is indeed defined after L224 in Equation 4. In L224, we were referring to the first estimator $\nabla \widehat{\mathcal{F}}(\omega)$. To make this clear in the text, we plan to add in Section 3 a reference to Proposition B.1 in Appendix B, where the detailed expression of the first gradient estimator $\nabla \widehat{\mathcal{F}}(\omega)$ is provided.
>
> ### **5- Inner and outer distributions in Section 2.2**
> Thank you for the suggestion. You are correct: the expectations correspond to different distributions (training for the inner and test for the outer). We will update the equation in the revised version of our paper to explicitly highlight this distinction for clarity.
>
> ### **6- Experiments**
> Thank you for suggesting the addition of experimental results in the case where $m \neq n$: we find this to be a valuable and interesting experiment. To provide these results, we retain the same experimental setup as in the main paper and vary both $m$ and $n$ simultaneously between 100 and 5000. For each distribution (standard Gaussian and Student’s $t$-distribution with degrees of freedom 2.1, 2.5, and 2.9), we plot heatmaps of the quantities of interest, with $n$ on the $x$-axis and $m$ on the $y$-axis. These quantities include $|\mathcal{F}(\omega_0)-\widehat{\mathcal{F}}(\omega_0)|$, $\lVert\nabla\mathcal{F}(\omega_0)-\widehat{\nabla\mathcal{F}}(\omega_0)\rVert$, $\lVert\nabla\mathcal{F}(\omega_T)\rVert$, and $\min_{i=0,\ldots, T}\lVert \nabla\mathcal{F}(\omega_i)\rVert$. We plan to include them in the appendix of the revised version of our paper. Here, since all cases show similar trends, we report only the results for the Gaussian case and focus on the quantity $\lVert\nabla\mathcal{F}(\omega_0)-\widehat{\nabla\mathcal{F}}(\omega_0)\rVert$. The following table summarizes the results for $m, n \in$ {$100, 500, 1000, 5000$}.
> As observed, the quantity decreases as both $m$ and $n$ increase, and so is the upper-bound. However, we are unable to experimentally assess the tightness of the upper-bound at this stage.
>
> |$n \backslash m$ | $\mathbf{100}$ | $\mathbf{500}$ | $\mathbf{1000}$ | $\mathbf{5000}$ |
> |:----------------:|:----------------:|:----------:|:--------------:|:-------------:|
> | $\mathbf{100}$ |0.0340 | 0.0289 | 0.0235 | 0.0200 |
> | $\mathbf{500}$ |0.0265 | 0.0110 | 0.0090 | 0.0076 |
> | $\mathbf{1000}$ |0.0195 | 0.0081 | 0.0066 | 0.0056 |
> | $\mathbf{5000}$ |0.0152 | 0.0063 | 0.0044 | 0.0023 |
>
> Best regards,
>
> Authors
>
> ---
>
> **References**
>
> (Singh et al., 2019) R. Singh, M. Sahani, and A. Gretton. Kernel Instrumental Variable Regression. NeurIPS 2019.
>
> (Rahimi & Recht, 2007) A. Rahimi and B. Recht. Random Features for Large-Scale Kernel Machines. NIPS 2007.
>
> (Williams & Seeger, 2001) C. K. I. Williams and M. Seeger. Using the Nyström Method to Speed Up Kernel Machines. NIPS 2001.
>
> (Bao et al., 2021) F. Bao, G. Wu, C. Li, J. Zhu, and B. Zhang. Stability and generalization of bilevel programming in hyperparameter optimization. NeurIPS 2021.
>
> (Wang et al., 2024) R. Wang, C. Zheng, G. Wu, X. Min, X. Zhang, J. Zhou, and C. Li. Lower bounds of uniform stability in gradient-based bilevel algorithms for hyperparameter optimization. NeurIPS 2024.
>
> (Zhang et al., 2024) X. Zhang, H. Chen, B. Gu, T. Gong, and F. Zheng. Fine-grained analysis of stability and generalization for stochastic bilevel optimization. In Proceedings of the Thirty-Third International Joint Conference on Artificial Intelligence, 2024.
>
> (Oymak et al., 2021) S. Oymak, M. Li, and M. Soltanolkotabi. Generalization guarantees for neural architecture search with train-validation split. ICML 2021.
>
> (Arora et al., 2020) S. Arora, S. Du, S. Kakade, Y. Luo, and N. Saunshi. Provable representation learning for imitation learning via bi-level optimization. ICML 2020.

---

> > ### Comment · Reviewer_cVHc · 2025-08-08
> >
> > I would like to thank the authors for a clarifying reply to my review and I trust that they know how to implement any changes in regards to the clarity of presentation if accepted. I decided to keep my scores.

---

### Official Review · Reviewer_9Xjf · 2025-07-02

**Clarity:** 4
**Significance:** 3
**Originality:** 3
**Rating:** 5
**Confidence:** 2

**Summary:**

This paper develops the first finite-sample generalization bounds for nonparametric bilevel optimization with inner optimization on reproducing kernel Hilbert spaces. Specifically, it gives an explicit gradient expression using implicit differentiation and then derives uniform generalization bounds for both the bilevel value function and its gradient using empirical process and U-process theory.

**Questions:**

- Could the authors comment on the possibility of relaxing Assumptions (C) and (D)?
- Can your framework extend to stochastic bilevel algorithms (e.g., [3,24])?

**Ethical Concerns:**

["NO or VERY MINOR ethics concerns only"]

**Final Justification:**

The authors' response addressed my concern. My recommended score remains unchanged at 5, as it was based on my original recognition of the work, which was not initially affected by these concerns.

**Limitations:**

Yes. The limitations and scope are clearly discussed in Section 6.

**Paper Formatting Concerns:**

None.

**Quality:**

3

**Strengths And Weaknesses:**

**Strengths**
- The work is technically solid, leveraging advanced empirical process methods to handle infinite-dimensional inner spaces for both function and gradient approximations.
- The paper is overall clearly written. The main ideas and proof strategies are well organized, with clear motivation, literature review, and fully discussed theoretical assumptions.
- Bilevel methods underpin many ML tasks (hyperparameter tuning, meta-learning, causal inference), yet lack nonparametric generalization theory. This fills a fundamental gap.
- It introduces a novel perspective to investigate the theoretical properties of bilevel optimization with kernels, which may largely inspire future work to loosen the inner problem's constraints.

**Weaknesses**
- The assumptions (e.g., smoothness and convexity) are relatively strong and may not hold in more complex or practical settings. It would be valuable to validate these assumptions empirically for real-world kernels, though this is admittedly a challenging task.
- Analysis focuses on basic gradient methods.

---

> ### Author Rebuttal · Authors · 2025-07-30
>
> Dear Reviewer,
>
> Thank you for providing feedback and highlighting the strengths of our paper. Below, we address your concerns and welcome any further questions.
>
> ### **1- Assumptions**
> We will add the following discussion on the assumptions.
> - **Assumption (A) – Boundedness of the kernel $K$.** We require $K$ to be bounded, which holds for the most commonly used kernels (Gaussian, Laplacian, Matérn, spline, etc.). It also holds if $K$ is a Mercer kernel, i.e., a continuous, positive‐definite kernel on a compact domain $\mathcal{X}$. For instance, a kernel built using neural network features (e.g., Neural Tangent Kernel) satisfies this assumption on the space of images, which is compact (pixel values have bounded range).
> - **Assumption (B) – Compactness of the set of the labels $\mathcal{Y}$.** In most supervised learning applications, this assumption is immediately met. For example, in regression tasks one often has $y\in [a,b]$ (e.g., probabilities in [0,1] or physical measurements on a known scale), which is compact. Similarly, in classification problems $\mathcal{Y}=${$1,\dots,C$} is compact.
> - **Assumption (C) – Regularity of pointwise losses.** This assumption holds for most loss functions used in practice, including the squared loss, logistic loss, cross-entropy loss, and KL divergence. This assumption no longer holds when considering SVM with the hinge loss. To handle such a case, one would need to extend the analysis to a non-smooth setting, which would be an interesting future work direction.
> - **Assumption (D) – Convexity of the inner loss.** What makes this convexity assumption mild is the fact that it is made on the variable $v$, which corresponds to the value taken by a prediction model $h$. This is weaker than making a convexity assumption with respect to the parameters of such a model. For instance, as discussed in (Petrulionyte et al., 2024), in the case of a neural network model, convexity in the parameters would not hold in general, but it would still hold in the values taken by the model for most commonly used losses in practice, such as the squared loss, logistic loss, cross-entropy loss, and KL divergence.
>
> ### **2- Extension to stochastic bilevel algorithms**
> We thank the reviewer for this insightful question. While our theoretical analysis focuses on a **full-batch** bilevel setting with **exact gradients**, the proposed framework could be extended to stochastic variants. A promising direction would be to consider **approximate kernel representations**, such as **random Fourier features** or **Neural Tangent Kernels**, which enable scalable learning using kernel methods while preserving useful theoretical properties. These extensions lie beyond the current scope of our work and represent an exciting direction for future research. We will make sure to clearly highlight this point in the conclusion of the revised version.
>
> Best regards,
>
> Authors
>
> ---
>
> **Reference**
>
> (Petrulionyte et al., 2024) I. Petrulionyte, J. Mairal, and M. Arbel. Functional Bilevel Optimization for Machine Learning. NeurIPS 2024.

---

> > ### Comment · Reviewer_9Xjf · 2025-08-02
> >
> > Thank you for the detailed response. My concerns are well addressed.

---

### Note · Authors · 2025-08-14

Dear Area Chair,

We thank you for overseeing the review process and for the reviewers’ constructive feedback. We are pleased that the reviews highlighted the strong technical rigor of our work, its clear and accessible presentation, and its novel contributions to kernel bilevel optimization and nonparametric generalization theory, as well as its potential to inspire future research directions. We also appreciate the constructive suggestions provided and will incorporate these improvements in a revised version of our paper, as stated in our rebuttal to each review.

Best regards,

Authors

---

### Decision · Program_Chairs · 2025-09-17

**Decision:**

Accept (poster)

**Comment:**

The paper presents the first finite-sample generalization bounds for nonparametric bilevel optimization, where the inner problem is solved over a reproducing kernel Hilbert space. The work's primary strength is its technical rigor and novelty, leveraging advanced empirical and U-process theory to handle the infinite-dimensional setting of the RKHS. The core contributions include showing that differentiation and empirical estimation commute, and using this property to derive generalization bounds for both the objective function and its gradient. Thus I recommend the paper to be accepted.